# Longer scans boost prediction and cut costs in brain-wide association studies

Leon Qi Rong Ooi[1,2,3,4,5,37], Csaba Orban[2,3,5,37], Shaoshi Zhang[1,2,3,4,5,37], Thomas E. Nichols[6,7], Trevor Wei Kiat Tan[1,2,3,4,5], Ru Kong[2,3,4,5], Scott Marek[8,9], Nico U. F. Dosenbach[8,9,10,11,12,13], Timothy O. Laumann[9,14], Evan M. Gordon[8,9], Kwong Hsia Yap[15,16], Fang Ji[2,3], Joanna Su Xian Chong[2,3], Christopher Chen[15,16], Lijun An[17], Nicolai Franzmeier[18,19,20], Sebastian N. Roemer-Cassiano[18,21], Qingyu Hu[22], Jianxun Ren[22], Hesheng Liu[22,23], Sidhant Chopra[24,25], Carrisa V. Cocuzza[26,27], Justin T. Baker[28,29], Juan Helen Zhou[1,2,3,4], Danilo Bzdok[30,31,32], Simon B. Eickhoff[33,34], Avram J. Holmes[27], B. T. Thomas Yeo[1,2,3,4,5,35✉] & Alzheimer's Disease Neuroimaging Initiative*

A pervasive dilemma in brain-wide association studies[1] (BWAS) is whether to prioritize functional magnetic resonance imaging (fMRI) scan time or sample size. We derive a theoretical model showing that individual-level phenotypic prediction accuracy increases with sample size and total scan duration (sample size × scan time per participant). The model explains empirical prediction accuracies well across 76 phenotypes from nine resting-fMRI and task-fMRI datasets ($R^2 = 0.89$), spanning diverse scanners, acquisitions, racial groups, disorders and ages. For scans of ≤20 min, accuracy increases linearly with the logarithm of the total scan duration, suggesting that sample size and scan time are initially interchangeable. However, sample size is ultimately more important. Nevertheless, when accounting for the overhead costs of each participant (such as recruitment), longer scans can be substantially cheaper than larger sample size for improving prediction performance. To achieve high prediction performance, 10 min scans are cost inefficient. In most scenarios, the optimal scan time is at least 20 min. On average, 30 min scans are the most cost-effective, yielding 22% savings over 10 min scans. Overshooting the optimal scan time is cheaper than undershooting it, so we recommend a scan time of at least 30 min. Compared with resting-state whole-brain BWAS, the most cost-effective scan time is shorter for task-fMRI and longer for subcortical-to-whole-brain BWAS. In contrast to standard power calculations, our results suggest that jointly optimizing sample size and scan time can boost prediction accuracy while cutting costs. Our empirical reference is available online for future study design (https://thomasyeolab.github.io/OptimalScanTimeCalculator/index.html).

A fundamental question in systems neuroscience is how individual differences in brain function are related to common variation in phenotypic traits, such as cognitive ability or physical health. Following recent work[1], we define BWAS as studies of the associations between phenotypic traits and common interindividual variability of the human brain. An important subclass of BWAS seeks to predict individual-level phenotypes using machine learning. Individual-level prediction is important for addressing basic neuroscience questions and is critical for precision medicine[2–7].

Many BWAS are underpowered, leading to low reproducibility and inflated prediction performance[8–13]. Larger sample sizes increase the reliability of brain–behaviour associations[14,15] and individual-level prediction accuracy[16,17]. Indeed, reliable BWAS typically requires thousands of participants[1], although certain multivariate approaches might reduce sample-size requirements[15].

In parallel, other studies have emphasized the importance of a longer fMRI scan time per participant during both resting and task states, which leads to improved data quality and reliability[12,18–23], as well as new insights into the brain[24–27]. When sample size is fixed, increasing resting-state fMRI scan time per participant improves the individual-level prediction accuracy of some cognitive measures[28].

Thus, in a world with infinite resources, fMRI-based BWAS should maximize both sample size and scan time for each participant. However, in reality, BWAS investigators have to decide between scanning more participants (for a shorter duration) or fewer participants (for a longer duration). Furthermore, there is a fundamental asymmetry between sample size and scan time per participant owing to inherent overhead cost associated with each participant that can be quite substantial, for example, when recruiting from a rare population. Notably, the exact trade-off between sample size and scan time per participant

A list of affiliations appears at the end of the paper. *A list of authors and their affiliations appears at the end of the paper.

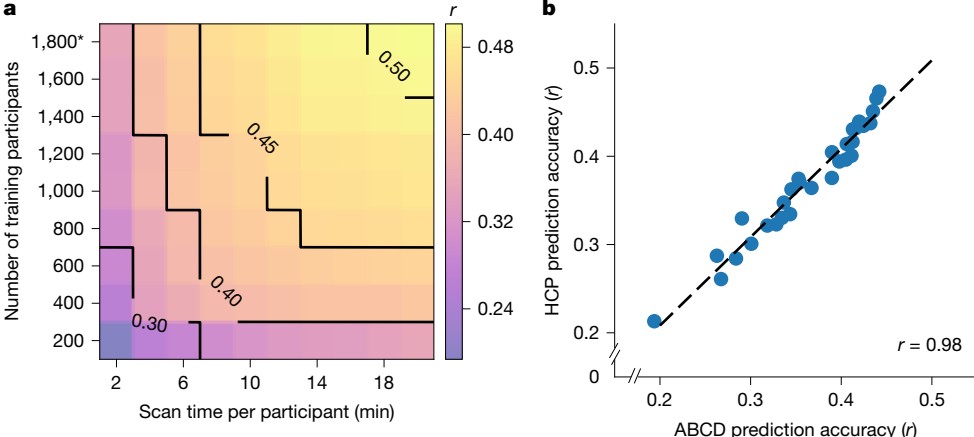

**Fig. 1 | Increasing the number of training participants and the scan time per participant leads to higher phenotypic prediction accuracy. a**, The prediction accuracy (Pearson's correlation) of the cognitive factor score as a function of the scan time *T* used to generate the functional connectivity matrix, and the number of training participants *N* used to train the predictive model in the ABCD dataset. Increasing the number of training participants and scan time both improved the prediction performance. The asterisk indicates that all of the available participants were used and the sample size is therefore close to, but not exactly, the number shown. **b**, The cognitive factor prediction accuracy (Pearson's correlation) in the ABCD and HCP datasets. There are 30 dots in this plot. Each dot represents the prediction accuracy in each dataset for a particular pair of sample size and scan time per participant. The Pearson's correlation between the 30 pairs of dots was 0.98.

has not been comprehensively characterized. This trade-off is not only relevant for small-scale studies, but also important for large-scale data collection, given competing interests among investigators and limited participant availability.

Here we systematically characterize the effects of sample size and scan time of fMRI on BWAS prediction accuracy, using the Adolescent Brain and Cognitive Development (ABCD) study and the Human Connectome Project (HCP). To derive a reference for future study design, we also considered the Transdiagnostic Connectome Project (TCP), Major Depressive Disorder (MDD), Alzheimer's Disease Neuroimaging Initiative (ADNI) and the Singapore Geriatric Intervention Study to Reduce Cognitive Decline and Physical Frailty (SINGER) datasets (Extended Data Table 1; see the 'Datasets, phenotypes and participants' section of the Methods). We find that, to increase prediction power, longer scans and larger sample sizes can yield substantial cost savings compared with increasing only sample size.

## Sample-size and scan-time interchangeability

For each participant in the HCP and ABCD datasets, we calculated a 419 × 419 resting-state functional connectivity (RSFC) matrix using the first *T* minutes of fMRI[29,30] (see the 'Image processing' section of the Methods). *T* was varied from 2 min to the maximum scan time in each dataset in intervals of 2 min. The RSFC matrices (from the first *T* minutes) served as input features to predict a range of phenotypes in each dataset using kernel ridge regression (KRR) through a nested inner-loop cross-validation procedure (see the 'Prediction workflow' section of the Methods). The analyses were repeated with different numbers of training participants (that is, different training sample size *N*). Within each cross-validation loop, the test participants were fixed across different training set sizes, so that the prediction accuracy was comparable across different training set sizes (Extended Data Fig. 1). The whole procedure was repeated multiple times and averaged. The sample sizes and maximum scan times of all datasets are provided in Extended Data Table 1.

We first considered the cognitive factor score because the cognitive factor score was predicted the best across all phenotypes[31]. Figure 1a shows the prediction accuracy (Pearson's correlation) of the ABCD cognitive factor score (HCP results are shown in Supplementary Fig. 1). Along a black iso-contour line, prediction accuracy is

(almost) constant even though scan time and sample size are changing. Consistent with previous literature[16,32], increasing the number of training participants (when scan time per participant is fixed) improved prediction performance. Similarly, increasing the scan time per participant (when sample size is fixed) also improved prediction performance[28].

Although cognitive factor scores are not necessarily comparable across datasets (due to population and phenotypic differences), prediction accuracies were highly similar between the ABCD and HCP datasets (Pearson's *r* = 0.98; Fig. 1b). Similar conclusions were also obtained when we measured the prediction accuracy using the coefficient of determination (COD) instead of Pearson's correlation (Supplementary Fig. 2), computed RSFC using the first *T* minutes of uncensored data (Supplementary Fig. 3), did not perform censoring of high motion frames (Supplementary Fig. 4) or used linear ridge regression (LRR) instead of KRR (Supplementary Figs. 5 and 6).

Notably, the prediction accuracy of the cognitive factor score increased with the total scan duration (number training participants × scan time per participant) in both the ABCD (Spearman's *ρ* = 0.99) and HCP (Spearman's *ρ* = 0.96) datasets (Fig. 2a). In both datasets, there were diminishing returns of sample size and scan time, whereby each unit increase in sample size or scan duration resulted in progressively smaller gains in prediction accuracy (Fig. 2a and Supplementary Table 1).

In the HCP dataset, we also observed diminishing returns of scan time relative to sample size, especially beyond 30 min (Fig. 2a and Supplementary Table 1). For example, starting from an accuracy of 0.33 with 200 participants × 14 min scans, a 3.5× larger sample (*N* = 700) increased the accuracy to 0.45, whereas a 4.1× longer scan (*T* = 58 min) raised it only to 0.40.

Beyond the cognitive factor scores, we focused on 29 (out of 59) HCP phenotypes and 23 (out of 37) ABCD phenotypes with maximum prediction accuracies of *r* > 0.1 (Supplementary Table 2). In total, 90% of HCP phenotypes (that is, 26 out of 29) and 100% of ABCD phenotypes (that is, 23 out of 23) exhibited prediction accuracies that increased with the total scan duration (Spearman's *ρ* = 0.85). Diminishing returns of scan time (relative to the sample size) were observed for many HCP phenotypes, especially beyond 20 min (Supplementary Table 1). This phenomenon was less pronounced for the ABCD phenotypes,

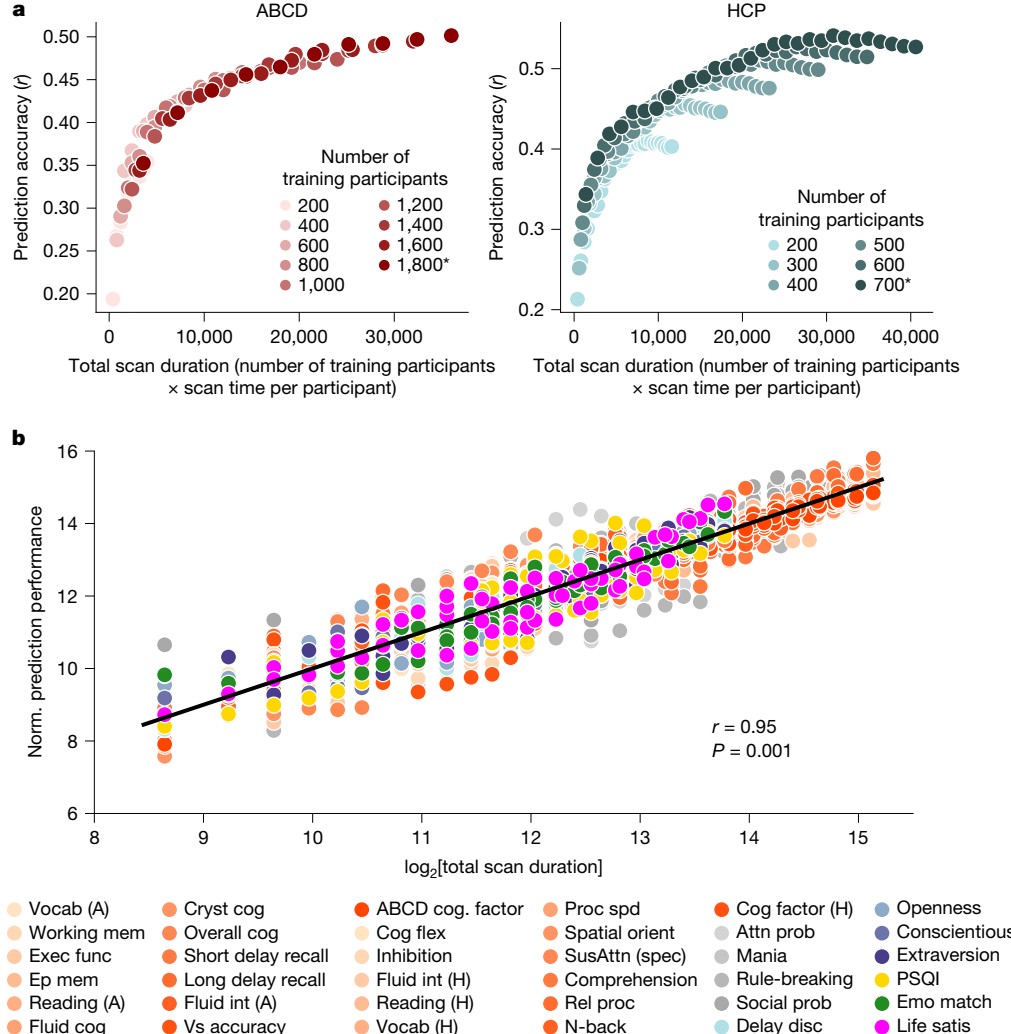

**Fig. 2 | The relationship between prediction accuracy and total scan duration (sample size × scan time per participant). a**, The prediction accuracy (Pearson's correlation) of the cognitive factor score as a function of the total scan duration (defined as the number of training participants × scan time per participant). There are 90 dots in the ABCD plot (left) and 174 dots in the HCP plot (right). Each colour shade represents a different total number of participants used to train the prediction algorithm. The asterisk indicates that all available participants were used and the sample size is therefore close to, but not exactly, the number shown. In both datasets, there were diminishing returns of both sample size and scan time, whereby each unit increase in sample size or scan duration resulted in progressively smaller gains in prediction accuracy. In the HCP dataset, the diminishing returns of scan time were more prominent beyond 30 min (Supplementary Table 1). **b**, The normalized (norm.) prediction accuracy of the two cognitive factor scores and 34 other phenotypes versus $\log_2$[total scan duration], ignoring data beyond 20 min of scan time.

Cognitive, mental health, personality, physicality, emotional and well-being measures are shown in shades of red, grey, blue, yellow, green and pink, respectively. The black line shows that the logarithm of total scan duration explained prediction performance well across phenotypic domains and datasets. The Pearson's correlation was computed between the log of total scan duration and normalized prediction performance based on 2,520 dots in the panel (16 total scan durations × 90 ABCD phenotypes + 18 total scan durations × 60 HCP phenotypes = 2,520). $P$ values were computed using subsampling (to ensure independence) and 1,000 permutations (Supplementary Table 1). Attn prob, attention problems; cog, cognition; cryst, crystalized; disc, discounting; emo match, emotional face matching; ep mem, episodic memory; exec funct, executive function; flex, flexibility; int, intelligence; mem, memory; orient, orientation; proc spd, processing speed; PSQI, Pittsburgh Sleep Quality Index; rel proc, relational processing; satis, satisfaction; SusAttn (spec), sustained attention (specificity); vocab, vocabulary; vs, visuospatial.

potentially because the maximum scan time was only 20 min (Supplementary Table 1).

A logarithmic pattern between prediction accuracy and total scan duration was evident in 73% (19 out of 26) HCP and 74% (17 out of 23) of ABCD phenotypes (Supplementary Table 2 and Supplementary Figs. 7 and 8). To quantify the logarithmic relationship, for each of the 19 HCP and 17 ABCD phenotypes, we fitted a logarithm curve (with two free parameters) between prediction accuracy and total scan duration (ignoring data beyond 20 min per participant; see the 'Fitting the logarithmic model' section of the Methods). Overall, total scan duration explained prediction accuracy across HCP and ABCD phenotypes very well (COD or $R^2$ = 0.88 and 0.89, respectively; Supplementary Table 3).

The logarithm fit allowed phenotypic measures from both datasets to be plotted on the same normalized prediction performance scale (Fig. 2b and Extended Data Fig. 2). The logarithm of the total scan duration explained prediction accuracy very well ($r$ = 0.95; $P$ = 0.001). This suggests that sample size and scan time are broadly interchangeable, in the sense that a larger sample size can compensate for a smaller scan time and vice versa. The exact degree of interchangeability is characterized in the next section.

The logarithm curve was also able to explain prediction accuracy well across different prediction algorithms (KRR and LRR) and different performance metrics (COD and $r$), as illustrated for the cognitive factor scores in Supplementary Fig. 9.

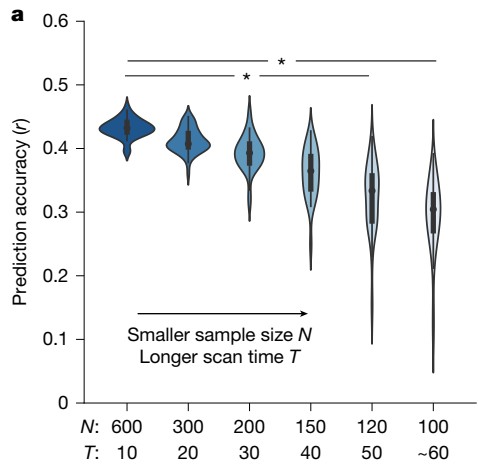

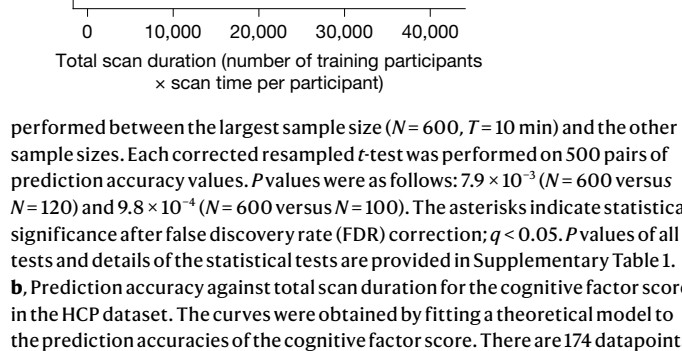

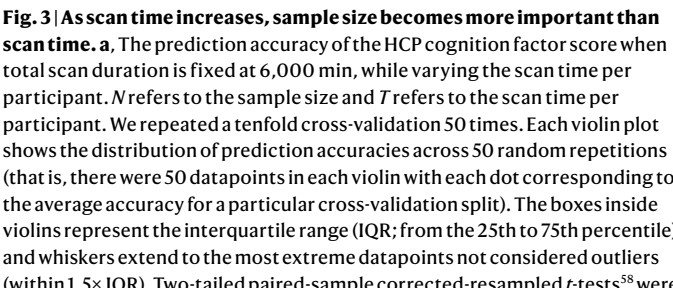

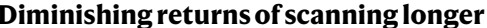

**Fig. 3 | As scan time increases, sample size becomes more important than scan time. a**, The prediction accuracy of the HCP cognition factor score when total scan duration is fixed at 6,000 min, while varying the scan time per participant. $N$ refers to the sample size and $T$ refers to the scan time per participant. We repeated a tenfold cross-validation 50 times. Each violin plot shows the distribution of prediction accuracies across 50 random repetitions (that is, there were 50 datapoints in each violin with each dot corresponding to the average accuracy for a particular cross-validation split). The boxes inside violins represent the interquartile range (IQR; from the 25th to 75th percentile) and whiskers extend to the most extreme datapoints not considered outliers (within 1.5× IQR). Two-tailed paired-sample corrected-resampled $t$-tests[58] were

performed between the largest sample size ($N = 600$, $T = 10$ min) and the other sample sizes. Each corrected resampled $t$-test was performed on 500 pairs of prediction accuracy values. $P$ values were as follows: $7.9 \times 10^{-3}$ ($N = 600$ versus $N = 120$) and $9.8 \times 10^{-4}$ ($N = 600$ versus $N = 100$). The asterisks indicate statistical significance after false discovery rate (FDR) correction; $q < 0.05$. $P$ values of all tests and details of the statistical tests are provided in Supplementary Table 1. **b**, Prediction accuracy against total scan duration for the cognitive factor score in the HCP dataset. The curves were obtained by fitting a theoretical model to the prediction accuracies of the cognitive factor score. There are 174 datapoints in the panel. The theoretical model explains why the sample size is more important than scan time (see the main text).

## Diminishing returns of scanning longer

We have observed diminishing returns of scan time relative to sample size. To examine this phenomenon more closely, we considered the prediction accuracy of the HCP factor score as we progressively increased scan time per participant in 10 min increments while maintaining 6,000 min of total scan duration (Fig. 3a). The prediction accuracy decreased with increasing scan time per participant, despite maintaining 6,000 min of total scan duration (Fig. 3a). However, the accuracy reduction was modest for short scan times (Fig. 3a and Supplementary Table 1). Similar conclusions were obtained for all 19 HCP and 17 ABCD phenotypes that followed a logarithmic fit (Extended Data Fig. 3). These results indicate that, while longer scan times can offset smaller sample sizes, the required increase in scan time becomes progressively larger as scan duration extends.

To gain insights into this phenomenon, we derived a closed-form mathematical relationship relating prediction accuracy (Pearson's correlation) with scan time per participant $T$ and sample size $N$ (see the 'Fitting the theoretical model' section of the Methods). To provide an intuition for the theoretical derivations, we note that phenotypic prediction can be theoretically decomposed into two components: one component relating to an average prediction (common to all participants) and a second component relating to a participant's deviation from this average prediction.

The uncertainty (variance) of the first component scales as $1/N$, like a conventional standard error of the mean. For the second component, we note that the prediction can be written as regression coefficients × functional connectivity (FC) for linear regression. The uncertainty (variance) of the regression coefficient estimates scales with $1/N$. The uncertainty (variance) of the FC estimates scales with $1/T$ (that is, reliability improves with $T$). Thus, the uncertainty of the second component scales with $1/NT$. Overall, our theoretical derivation suggests that prediction accuracy can be expressed as a function of $1/N$ and $1/NT$ with three free parameters.

The theoretical derivations do not tell us the relative importance of the $1/N$ and $1/NT$ terms. We therefore fitted the theoretical model to actual prediction accuracies in the HCP and ABCD datasets. The goal was to determine (1) whether our theoretical model (despite the simplifying assumptions) would still explain the empirical results, and (2) to determine the relative importance of $1/N$ and $1/NT$ (see the 'Fitting the theoretical model' section of the Methods).

We found an excellent fit with actual prediction accuracies for the 19 HCP and 17 ABCD phenotypes that followed a logarithmic fit (Fig. 3b and Supplementary Figs. 10 and 11): $R^2 = 0.89$ for both datasets (Supplementary Table 2). When $T$ was small, the $1/NT$ term dominated the $1/N$ term, which explained the almost one-to-one interchangeability between the scan time and the sample size for shorter scan times. The existence of the $1/N$ term ensured that sample size was still slightly more important than scan time even for small $T$. The FC reliability eventually saturated with increasing $T$. Thus, the $1/N$ term eventually dominated the $1/NT$ term, so the sample size became much more important than the scan time.

For 20-min scans, the logarithmic and theoretical models performed equally well with equivalent goodness of fit ($R^2$) across the 17 ABCD phenotypes ($P = 0.57$; Supplementary Table 1). For longer scan times, the theoretical model exhibited better fit than the logarithmic model across the 19 HCP phenotypes ($P = 0.002$; Supplementary Table 1 and Supplementary Fig. 12). Furthermore, prediction accuracy under the logarithmic model will exceed a correlation of one for sufficiently large $N$ and $T$, which should not be possible. We therefore use the theoretical model in the remaining portions of the study.

## Predictability increases model adherence

To explore the limits of the theoretical model, recall that the 17 ABCD phenotypes and 19 HCP phenotypes were predicted with maximum prediction accuracies of Pearson's $r > 0.1$, and that the theoretical model was able to explain their prediction accuracies with an average COD or $R^2$ of 89% (Supplementary Table 2). If we loosened the prediction

threshold to include phenotypes of which the prediction accuracies (Pearson's $r$) were positive in at least 90% of all combinations of sample size $N$ and scan time $T$ (Supplementary Table 2), the model fit was lower but still relatively high with an average COD or $R^2$ of 76% and 73% in ABCD and HCP datasets, respectively (Supplementary Table 2).

More generally, phenotypes with high overall prediction accuracies adhered to the theoretical model well (an example is shown in Extended Data Fig. 4a), while phenotypes with poor prediction accuracies resulted in poor adherence to the model (an example is shown in Extended Data Fig. 4b). Indeed, the model fit was strongly correlated with prediction accuracy across phenotypes in both datasets (Spearman's $\rho = 0.90$; $P = 0.001$; Extended Data Fig. 4c,d). These findings suggest that the imperfect fit of the theoretical model for some phenotypes may be due to their poor predictability, rather than true variation in prediction accuracy with respect to sample size and scan time.

## Non-stationarity weakens model adherence

As noted above, some phenotypes probably fail to match the theoretical model owing to intrinsically poor predictability. However, there were also phenotypes that were reasonably well predicted, yet still exhibited a poor fit to the theoretical model. For example, 'Anger: Aggression' was reasonably well predicted in the HCP dataset. While the prediction accuracy increased with larger sample sizes (Spearman's $\rho = 1.00$), extending the scan duration did not generate a similarly consistent effect for this phenotype (Spearman's $\rho = 0.21$; Extended Data Fig. 5a).

This suggests that fMRI–phenotype relationships might be non-stationary for certain phenotypes, which violates an assumption in the theoretical model. To put this in more colloquial terms, the assumption is that the FC–phenotype relationship is the same (that is, stationary) regardless of whether FC was computed based on 5 min of fMRI from the beginning, middle or end of the MRI session. We note that, for both HCP and ABCD datasets, fMRI was collected over four runs. To test for non-stationarity, we randomized the fMRI run order independently for each participant and repeated the FC computation (and prediction) using the first $T$ min of resting-state fMRI data under the randomized run order (see the 'Non-stationarity analysis' section of the Methods). The run randomization improved the goodness of fit of the theoretical model ($P < 4 \times 10^{-5}$), suggesting the presence of non-stationarities (Extended Data Fig. 5b,c).

Arousal changes between or during resting-state scans are well established[33–37]; we therefore expect fMRI scans, especially longer-duration scans, to be non-stationary. However, as run randomization affected some phenotypes more than others, this suggests that there is an interaction between fMRI non-stationarity and phenotypes, that is, the fMRI–phenotype relationship is also non-stationary.

## Higher overhead costs favour longer scans

We have shown that investigators have some flexibility in attaining a specified prediction accuracy through different combinations of sample size and scan time per participant (Fig. 1). Furthermore, the theoretical model suggests that the sample size is more important than the scan time (Fig. 3). However, when designing a study, it is important to consider the fundamental asymmetry between sample size and scan time per participant owing to the inherent overhead cost associated with each participant. These overhead costs might include recruitment effort, manpower to perform neuropsychological tests, additional MRI modalities (for example, anatomical T1, diffusion MRI), other biomarkers (for example, positron emission tomography (PET) or blood tests). Thus, the overhead cost can often be higher than the cost of the fMRI scan itself.

To derive a reference for future studies, we considered four additional resting-state datasets (TCP, MDD, ADNI and SINGER; see the 'Datasets,

phenotypes and participants' section of the Methods). In total, 34 phenotypes exhibited good fit to the theoretical model (Supplementary Table 3 and Supplementary Figs. 13–16). We also considered task-FC of the three ABCD tasks, and found that the number of phenotypes with a good fit to the theoretical model ranged from 16 to 19 (Supplementary Table 3 and Supplementary Figs. 17–19).

In total, we considered nine datasets: six resting-fMRI datasets and three ABCD task-fMRI datasets. We fitted the theoretical model to 76 phenotypes in the nine datasets, yielding an average COD or $R^2$ of 89% (Supplementary Table 1). These datasets span multiple fMRI sequences (single-echo single-band, single-echo multiband, multi-echo multiband), coordinate systems (fsLR, fsaverage, MNI152), racial groups (Western and Asian populations), mental health conditions (healthy, neurological and psychiatric) and age groups (children, young adults and older individuals). More dataset characteristics are shown in Supplementary Table 5.

For each phenotype, the fitted model was normalized to the phenotype's maximum achievable accuracy (estimated by the theoretical model), yielding a fraction of maximum achievable prediction accuracy for every combination of sample size and scan time per participant. The fraction of maximum achievable prediction accuracy was then averaged across the phenotypes under a hypothetical tenfold cross-validation scenario (Fig. 4a). Note that the Pearson's correlation between Figs. 4a and 1a across corresponding sample sizes and scan durations was 0.97 (Supplementary Table 1).

Given a scan cost per hour (for example, US$500) and overhead cost per participant (for example, US$500), we can find all pairs of sample sizes and scan times that fit within a particular fMRI budget (for example, US$1 million). We can then use Fig. 4a to find the optimal sample size and scan time leading to the largest fraction of maximum prediction accuracy (see the 'Optimizing within a fixed fMRI budget' section of the Methods). Extended Data Fig. 6 illustrates the prediction accuracy that is achievable with different fMRI budgets, costs per hour of scan time and overhead costs per participant. Extended Data Table 2 shows the optimal scan times for a wider range of fMRI budgets, scan costs per hour and overhead costs per participant.

Larger fMRI budgets, lower scan costs and lower overhead costs enable larger sample sizes and scan times, leading to a greater achievable prediction accuracy (Extended Data Fig. 6). From the curves, we can determine the optimal scan time to achieve the greatest prediction accuracy within a fixed scan budget (Extended Data Fig. 6 (solid circles)). The optimal scan time increases with larger overhead costs, lower fMRI budget and lower scan costs. As the scan time per participant increases, all curves exhibit a steep initial ascent, followed by a gradual decline. The asymmetry of the curves suggests that it is better to overshoot than undershoot optimal scan time (Supplementary Table 1).

For example, consider a US$2.5 million US National Institutes of Health (NIH) R01 grant. Assuming an fMRI budget of US$1 million, a scan cost of US$500 per hour and an overhead cost of US$500 per participant, the optimal scan time would be 34.5 min per participant. Suppose PET data were also collected, then the overhead cost might increase to US$5,000 per participant, resulting in an optimal scan time of 159.3 min per participant.

## 30-min scans are the most cost-effective

Beyond optimizing scan time to maximize prediction accuracy within a fixed scan budget (previous section), the model fits shown in Fig. 4a can also be used to optimize scan time to minimize the study cost to achieve a fixed accuracy target. For example, suppose we want to achieve 90% of the maximum achievable accuracy, we can find all pairs of sample size and scan time per participant along the black contour line corresponding to 0.9 in Fig. 4a. For every pair of sample size and scan time, we can then compute the study cost given a particular scan cost per hour (for

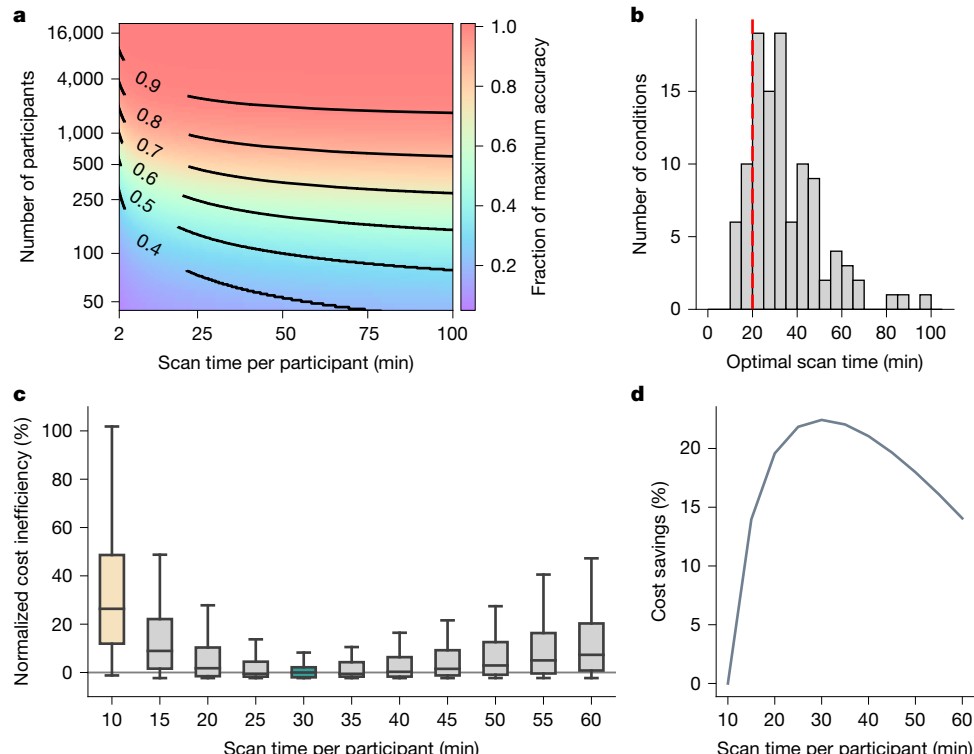

**Fig. 4 | 30-min scans yield considerable cost savings over 10-min scans across nine datasets. a**, The fraction of maximum prediction accuracy as a function of sample size and scan time per participant averaged across 76 phenotypes from nine datasets (six resting-fMRI and three ABCD task-fMRI datasets). We assumed that 90% of participants were used for training the predictive model, and 10% for model evaluation (that is, tenfold cross-validation). **b**, The optimal scan time (minimizing costs) across 108 scenarios. Given three possible accuracy targets (80%, 90% or 95% of the maximum achievable accuracy), 2 possible overhead costs (US$500 or US$1,000 per participant) and 2 possible scan costs per hour (US$500 or US$1,000), there were 3 × 2 × 2 = 12 conditions. In total, we had 9 datasets × 12 conditions = 108 scenarios. For 85% of scenarios, the cost-optimal scan time was ≥20 min (red dashed line). **c**, The

normalized cost inefficiency (across the 108 scenarios) as a function of fixed scan time per participant, relative to the optimal scan time in **b**. In practice, the optimal scan time in **b** is not known in advance, so this plot seeks to derive a fixed optimal scan time generalizable to most situations. Each box plot contains 108 datapoints (corresponding to 108 scenarios). The box limits show the IQR, the horizontal lines show the median values and the whiskers span non-outlier extremes (within 1.5 ×IQR). For visualization, box plots were normalized by subtracting the cost inefficiency of the best possible fixed scan time (30 min in this case), so that the normalized cost inefficiency of the best possible fixed scan time is centred at zero. **d**, The cost savings relative to 10 min of scan time per participant. The greatest cost saving (22%) was achieved at 30 min.

example, US$500) and a particular overhead cost per participant (for example, US$1,000). The optimal scan time (and sample size) with the lowest study cost can then be obtained (see the 'Optimizing to achieve a fixed accuracy' section of the Methods).

Here we considered three possible accuracy targets (80%, 90% or 95% of maximum accuracy), two possible overhead costs (US$500 or US$1,000) and two possible scan costs per hour (US$500 or US$1,000). In total there were 3 × 2 × 2 = 12 conditions. As there were nine datasets, this resulted in 12 × 9 = 108 scenarios. In the vast majority (85%) of these 108 scenarios, the optimal scan time was at least 20 min (Fig. 4b).

However, during study design, the optimal scan time is not known in advance. We therefore also aimed to identify a fixed scan time that is cost-effective in most situations. Figure 4c shows the normalized cost inefficiency of various fixed scan times relative to the optimal scan time for each of 108 scenarios. Many consortium BWAS collect 10 min fMRI scans, which is highly cost inefficient. On average across resting and task states, 30 min scans were the most cost-effective (95% bootstrapped confidence interval (CI) = 25–40; Extended Data Fig. 7 and Supplementary Table 1), yielding 22% cost savings over 10 min scans (Fig. 4d). We again note the asymmetry in the cost curves, so it is cheaper to overshoot than undershoot the most cost-effective scan time. For example, 50-min scans overshoot the optimum by 20 min, but still incur 18% cost savings over 10 min scans (which undershoot the optimum by 20 min).

## Minimizing task-fMRI costs

Across the six resting-state datasets (Fig. 5a), the most cost-effective scan time was the longest for ABCD (60 min; CI = 40–100) and shortest for the TCP and ADNI datasets (20 min; TCP, CI = 10–35; ADNI, CI = 15–35). However, a scan time of 30 min was still relatively cost-effective for all datasets, owing to a flat cost curve near the optimum and the asymmetry of the cost curve. For example, even for the TCP dataset, which had the shortest most cost-effective scan time of 20 min, over-scanning with 30-min scans led to only a 3.7% higher cost relative to 20 min, compared with a 7.3% higher cost for under-scanning with 10 min scans.

Previous studies have shown that task-FC yields better prediction performance for cognitive measures[38,39]. Here we extend previous results, finding that the most cost-effective scan time was shorter for ABCD task-fMRI than ABCD resting-state fMRI (Fig. 5b and Supplementary Table 1). Among the three tasks, the most cost-effective scan time was the shortest for N-back at 25 min (CI = 20–35), but 30 min scans led to only a 0.9% higher cost (relative to 25 min), compared with a 16.1% higher cost for 10 min scans.

These task results suggest that the most cost-effective scan time is sensitive to brain state manipulation. Task-based fMRI may preferentially engage cognitive and physiological mechanisms that are closely tied to the expression of specific phenotypes (for example, processing speed), thereby enhancing the specificity of functional connectivity

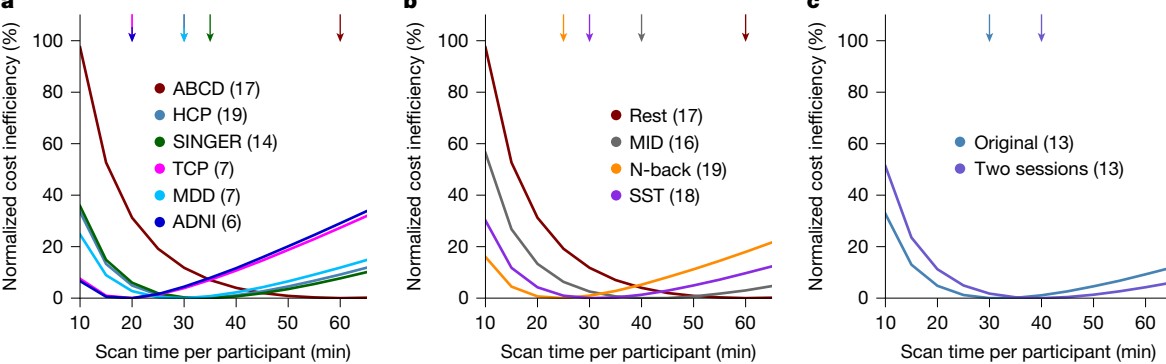

**Fig. 5 | Variation in the most cost-effective scan time across resting-state and task-state fMRI. a,** Cost inefficiency as a function of the scan time per participant for the six resting-state datasets. This plot provides the same information as Fig. 4c, but shown for each dataset separately. **b,** Cost inefficiency for ABCD resting-state and task-state fMRI. **c,** Cost inefficiency when scans collected in two separate sessions versus one session (based on the HCP dataset). Similar to Fig. 4c, for visualization, each curve is normalized by subtracting the cost inefficiency of the best possible fixed scan time (of each curve), so that the normalized cost inefficiency of the best possible fixed scan time is centred at zero. For **a**–**c**, the numbers in brackets indicate the number of phenotypes. The arrows indicate the most cost-effective scan time. 95% bootstrapped CIs are reported in Supplementary Table 1.

estimates for phenotypic prediction. Tasks may also facilitate shorter, more-efficient scan durations by aligning brain states across individuals in a controlled manner, thereby reducing spurious non-stationary influences that could otherwise obscure reliable modelling of inter-individual differences. This alignment might be better achieved in tasks that present stimuli and conditions with identical timing across participants—whether using event-related or block designs.

Non-stationarity may also be potentially increased by distributing resting-state fMRI runs across multiple sessions. As the HCP dataset was collected on two different days (sessions), we were also able to directly compare the effect of a two-session versus a one-session design. The most cost-effective scan time for the two-session design was only slightly longer than for the original HCP analysis (Fig. 5c): 40 min (CI = 30–55) versus 30 min (CI = 25–40).

Overall, these results suggest that state manipulation can influence the most cost-effective scan time, and that a relatively large state manipulation (for example, task fMRI) can significantly influence the cost-effectiveness.

## Variation across phenotypes and scan parameters

There were clear variations across phenotypes. For example, there were phenotypes that could be predicted well and demonstrated prediction gains up to the maximum amount of data per participant (for example, age in the ADNI dataset; Supplementary Fig. 16). However, there were also other phenotypes that were predicted less well (for example, BMI in the SINGER dataset; Supplementary Fig. 13) but showed prediction gains up to the maximum amount of data per participant. As single phenotypes are not easily interpreted, we grouped the phenotypes into seven phenotypic domains to study phenotypic variation in more detail.

For five out of the seven phenotypic domains, the most cost-effective scan times ranged from 25 min to 40 min (Extended Data Fig. 8a). The most cost-effective scan time for the emotion domain was exceptionally long, but this outlier was driven by a single phenotypic measure, so should not be overinterpreted. For the PET phenotypic domain, our original scenarios assumed overhead costs of US$500 or US$1,000 per participant, which was unrealistic. Assuming a more realistic overhead PET cost per participant (US$5,000 or $10,000) yielded 50 min as the most cost-effective scan time.

Although there was a strong relationship between phenotypic prediction accuracy and goodness-of-fit to the theoretical model (Extended Data Fig. 4), we did not find an obvious relationship between phenotypic prediction accuracy and optimal scan time (Extended Data Fig. 8b and Supplementary Table 1). Recent studies have also demonstrated

that phenotypic reliability is important for BWAS power[40,41]. In our theoretical model, phenotypic reliability directly impacts the overall prediction accuracy but does not directly contribute to the trade-off between sample size and scan time. Indeed, there was not an obvious relationship between phenotypic test–retest reliability and optimal scan time (Extended Data Fig. 8c and Supplementary Table 1).

There was also not an obvious relationship between optimal scan time and temporal resolution, voxel resolution or scan sequence (Extended Data Figs. 8d–f and Supplementary Table 1). We emphasize that we are not claiming that scan parameters do not matter, but that other variations between datasets (for example, phenotypes, populations) might exert a greater impact than common variation in scan parameters.

Consistent with the previous sections, we note that, for the vast majority of phenotypes and scan parameters, the optimal scan time was at least 20 min and, on average, the most cost-effective scan time was 30 min (Fig. 4c).

## Minimizing costs of subcortical BWAS

Our main analyses involved a cortical parcellation with 400 regions and 19 subcortical regions, yielding 419 × 419 RSFC matrices. We also repeated the analyses using 19 × 419 subcortical-to-whole-brain RSFC matrices. The most cost-effective scan time for subcortical RSFC was about double that of whole-brain RSFC (Fig. 6a, Supplementary Table 1 and Extended Data Fig. 7). This might arise due to the lower signal-to-noise ratio (SNR) in subcortical regions, resulting in the need for a longer scan time to achieve a better estimate of subcortical FC.

To explore the effects of fMRI SNR, for each parcel time course, we $z$-normalized the fMRI time course, so the resulting s.d. of the time course was equal to one. We then added zero mean Gaussian noise with s.d. of $\sigma$. Even doubling the noise ($\sigma = 1$) had very little impact on the optimal scan time (Fig. 6b and Supplementary Table 1). As a sanity check, we added a large quantity of noise ($\sigma = 3$), which led to a much longer optimal scan time (Fig. 6b and Supplementary Table 1).

Intuitively, this is not surprising because a lower SNR means that a longer scan time is necessary to get an accurate estimate of individual-level FC. It is interesting that a large SNR change is necessary to make a noticeable difference in optimal scan time, which might explain the robustness of optimal scan times across the common scan parameters that we explored in the previous section (Extended Data Fig. 8). Thus, even with small to moderate technological improvements in SNR, the most cost-effective scan time is unlikely to substantially deviate from our estimate. However, a major increase in SNR could shorten the most cost-effective scan time from the current estimates.

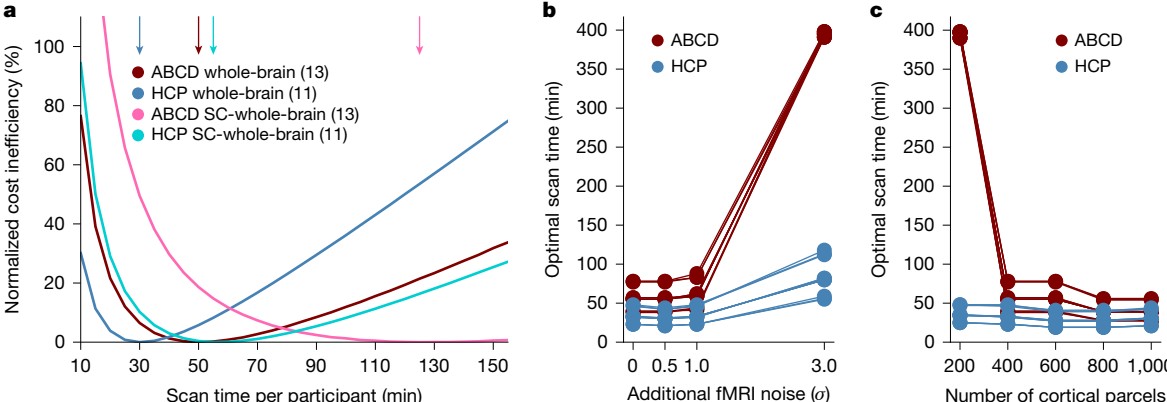

**Fig. 6 | The most cost-effective scan time for subcortical BWAS is longer than for whole-brain BWAS. a**, Cost inefficiency as a function of scan time per participant with subcortical-to-whole-brain FC versus whole-brain FC. For visualization, similar to Fig. 4c, the curves are normalized by subtracting the cost inefficiency of the best possible fixed scan time (of each curve), so that the normalized cost inefficiency of the best possible fixed scan time is centred at zero. The numbers in brackets indicate the number of phenotypes

in each condition. The arrows indicate the most cost-effective scan time. 95% bootstrapped CIs are reported in Supplementary Table 1. **b**, The optimal scan time for predicting the cognitive factor score as a function of simulated Gaussian noise with s.d. of $\sigma$. **c**, The optimal scan time for predicting the cognitive factor score as a function of the cortical parcellation resolution. For **b** and **c**, consistent with Fig. 4c, there were 12 conditions, resulting in 12 curves, but some curves overlap, so are not obvious. SC, subcortical.

We also varied the resolution of the cortical parcellation with 200, 400, 600, 800 or 1,000 parcels for predicting the cognitive factor scores in the HCP and ABCD datasets. There was a weak trend in which higher parcellation resolution led to slightly lower optimal scan time, although there was a big drop in the optimal scan time from 200 parcels to 400 parcels in the ABCD dataset (Fig. 6c and Supplementary Table 1). Given that subcortical-to-whole-brain FC has fewer edges (features) than whole-brain FC, this could be another reason why subcortical-to-whole-brain FC requires longer optimal scan time.

## Accuracy versus reliability

Finally, we examine the effects of sample size and scan time per participant on the reliability of BWAS[1] using a previously established split-half procedure[14,15] (Supplementary Fig. 20; see the 'Brain-wide association reliability' section of the Methods). For both univariate and multivariate BWAS reliability, diminishing returns of scan time (relative to sample size) occurred beyond 10 min per participant (Supplementary Figs. 21–29), instead of 20 min for prediction accuracy (Fig. 2). We note that reliability is necessary but not sufficient for validity[21,42]. For example, hardware artifacts may appear reliably in measurements without having any biological relevance. Thus, reliable BWAS features do not guarantee accurate prediction of phenotypic measures. As such, we recommend that researchers prioritize prediction accuracy.

## Longer scans are more cost-effective

To summarize, 30 min scans are on average the most cost-effective across resting-state and task-state whole-brain BWAS (Fig. 4c). The cost curves are also asymmetric, so it is cheaper to overshoot than undershoot the optimum (Fig. 4d). Thus, even when the most-effective scan time is shorter than 30 min (for example, N-back task or TCP dataset), 30-min scans incur only a small penalty relative to knowing the true optimal scan time a priori. Furthermore, for subcortical BWAS, the most cost-effective scans are much longer than 30 min.

Our results present a compelling case for moving beyond traditional power analyses, of which the only inputs are sample size, to inform BWAS design. Such power analyses can only point towards maximizing the sample size, so the scan time becomes implicitly minimized under budget constraints. Our findings show that we can achieve higher prediction performance by increasing both the sample size and the

scan time, while generating substantial cost-savings compared with increasing the sample size alone.

Our results complement recent advocation for larger sample sizes to increase BWAS reproducibility[1]. Consistent with previous studies[43], when sample size is small, there is a high degree of variability across cross-validation folds (Fig. 3a). Furthermore, large sample sizes are still necessary for high prediction accuracy. To achieve 80% of the maximum prediction accuracy with 30-min scans, a sample size of about 900 is necessary (Fig. 4a), which is much larger than typical BWAS[1]. To achieve 90% of the maximum prediction accuracy with 30-min scans, a sample size of around 2,500 is necessary (Fig. 4a).

In addition to increasing the sample size and scan time, BWAS effect sizes can also be enhanced through innovative study designs. Recent work showed that U-shaped population sampling can enhance the strength of associations between functional connectivity and phenotypic measures[44]. However, more complex screening procedures will increase the overhead costs per participant, which might lengthen optimal scan time.

The current analysis was focused on high target accuracies (80%, 90% or 95%) and relatively low overhead costs (US$500 or US$1,000). Lower target accuracies (in smaller-scale studies) and higher overhead costs (for example, PET, multisite data collection) will lead to longer cost-effective scan time (Extended Data Fig. 6). In practice, scans are also more likely to be spuriously shortened (for example, due to participant discomfort) than to be spuriously extended. We therefore recommend a scan time of at least 30 min.

Overall, 10 min scans are rarely cost-effective, and the optimum scan time is at least 20 min in most BWAS (Fig. 4c). Among the datasets that we analysed, four included scans of at least 20 min, providing robust evidence to support this conclusion across multiple datasets. By contrast, we could identify only one dataset (HCP) with scans exceeding 30 min and a sufficiently large sample size for inclusion in our study. Similarly, although the ABCD task-fMRI scans are among the longest in existing large-scale datasets, the longest scan duration is less than 13 min. This limitation underscores the importance of our findings, emphasizing the need for BWAS to prioritize longer scans.

## Non-economic considerations

Beyond economic considerations, the representativeness of the data sample and the generalizability of predictive models to subpopulations are also important factors when designing a study[45–50]. One approach

would be to aim for a larger sample size (potentially at the expense of scan time) to ensure sufficient sample sizes for subpopulations. Alternatively, one could also make the participant-selection criteria more stringent to maintain the representativeness of a subpopulation. However, this would drive up the recruitment cost for the subpopulation, so our results suggest that it might be more economically efficient to scan harder-to-recruit subpopulations longer. For example, instead of 20 min resting-state scans for all ABCD participants, perhaps subpopulations (for example, Black participants) could be scanned for a longer period of time.

In other situations, the sample size is out of the investigator's control, for example, if the investigator wants to scan an existing cohort. In the case of the SINGER dataset, the sample size was determined by the power calculation of the actual lifestyle intervention[51] with the imaging data included to gain further insights into the intervention. As another example, in large-scale prospective studies (for example, the UK Biobank), the sample size is determined by the fact that only a small proportion of participants will develop a given condition in the future[52]. In these situations, the scan time becomes constrained by the overall budget and fitting all phenotyping efforts within a small number of sessions (to avoid participant fatigue). Nevertheless, even in these situations in which the sample size is predetermined, Fig. 4a can still provide an empirical reference on the marginal gains in prediction accuracy as a function of scan time.

Finally, some studies may necessitate extensive scan time per participant by virtue of the scientific question. For example, when studying sleep stages, it is not easy to predict how long a participant would need to enter a particular sleep stage. Conversely, some phenomena of interest might be inherently short-lived. For example, if the goal is to characterize the effects of a fast-acting drug (for example, nitrous oxide), then it might not make sense to collect long fMRI scans. Furthermore, not all studies are interested in cross-sectional relationships between brain and non-brain-imaging phenotypes. For example, in the case of personalized brain stimulation[53,54] or neurosurgical planning[55], a substantial quantity of resting-state fMRI data might be necessary for accurate individual-level network estimation[24,56,57].

## A web application for study design

Beyond our broad recommendation of scan times of at least 30 min, we recognize that investigators might be interested in achieving the optimal sample size and scan time specific to their study's constraints. We therefore built a web application to help to facilitate flexible study design (https://thomasyeolab.github.io/OptimalScanTimeCalculator/index.html). The web application includes additional constraints that were not analysed in the current study. For example, certain demographic and patient populations might not be able to tolerate longer scans, so an additional factor will be the maximum scan time in each MRI session. Furthermore, our analysis was performed on participants whose data survived quality control. We have therefore also provided an option on the web application to allow researchers to specify their estimate of the percentage of participants whose data might be lost due to poor data quality or participant drop out. Overall, our empirically established guidelines provide actionable insights for significantly reducing costs, while improving BWAS individual-level prediction performance.

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

[1]Integrative Sciences and Engineering Programme (ISEP), National University of Singapore, Singapore, Singapore. [2]Centre for Sleep and Cognition & Centre for Translational MR Research, Yong Loo Lin School of Medicine, National University of Singapore, Singapore, Singapore. [3]Department of Medicine, Healthy Longevity Translational Research Programme, Human Potential Translational Research Programme & Institute for Digital Medicine (WisDM), Yong Loo Lin School of Medicine, National University of Singapore, Singapore, Singapore. [4]Department of Electrical and Computer Engineering, National University of Singapore, Singapore, Singapore. [5]N.1 Institute for Health, National University of Singapore, Singapore, Singapore. [6]Big Data Institute, Li Ka Shing Centre for Health Information and Discovery, Nuffield Department of Population Health, University of Oxford, Oxford, UK. [7]Centre for Integrative Neuroimaging (OxCIN), FMRIB, Nuffield Department of Clinical Neurosciences, University of Oxford, Oxford, UK. [8]Mallinckrodt Institute of Radiology, Washington University School of Medicine, St Louis, MO, USA. [9]Allied Labs for Imaging Guided Neurotherapies (ALIGN), Washington University School of Medicine, St Louis, MO, USA. [10]Department of Neurology, Washington University School of Medicine, St Louis, MO, USA. [11]Department of Paediatrics, Washington University School of Medicine, St Louis, MO, USA. [12]Department of Biomedical Engineering, Washington University, St Louis, MO, USA. [13]Department of Psychological and Brain Sciences, Washington University, St Louis, MO, USA. [14]Department of Psychiatry, Washington University, School of Medicine, St Louis, MO, USA. [15]Memory, Ageing and Cognition Centre, National University Health System, Singapore, Singapore. [16]Department of Pharmacology, Yong Loo Lin School of Medicine, National University of Singapore, Singapore, Singapore. [17]Department of Clinical Sciences, Malmö, SciLifeLab, Lund University, Lund, Sweden. [18]Institute for Stroke and Dementia Research, LMU Munich, Munich, Germany. [19]Munich Cluster for Systems Neurology (SyNergy), Munich, Germany. [20]Department of Psychiatry and Neurochemistry, Institute of Neuroscience and Physiology, University of Gothenburg, The Sahlgrenska Academy, Gothenburg, Sweden. [21]Department of Neurology, LMU Hospital, LMU Munich, Munich, Germany. [22]Changping Laboratory, Beijing, China. [23]Biomedical Pioneering Innovation Center (BIOPIC), Peking University, Beijing, China. [24]Orygen, Melbourne, Victoria, Australia. [25]Center for Youth Mental Health, University of Melbourne, Melbourne, Victoria, Australia. [26]Department of Psychology, Yale University, New Haven, CT, USA. [27]Department of Psychiatry, Brain Health Institute, Rutgers University, Piscataway, NJ, USA. [28]Department of Psychiatry, Harvard Medical School, Boston, MA, USA. [29]Institute for Technology in Psychiatry, McLean Hospital, Boston, MA, USA. [30]The Neuro, McConnell Brain Imaging Centre, Department of Biomedical Engineering, Montreal, Quebec, Canada. [31]Faculty of Medicine, School of Computer Science, McGill University, Montreal, Quebec, Canada. [32]Mila–Quebec Artificial Intelligence Institute, Montreal, Quebec, Canada. [33]Institute of Neuroscience and Medicine, Brain & Behaviour (INM-7), Research Center Jülich, Jülich, Germany. [34]Institute for Systems Neuroscience, Medical Faculty, Heinrich-Heine University Düsseldorf, Düsseldorf, Germany. [35]Martinos Center for Biomedical Imaging, Massachusetts General Hospital, Charlestown, MA, USA. [37]These authors contributed equally: Leon Qi Rong Ooi, Csaba Orban, Shaoshi Zhang. ✉e-mail: thomas.yeo@nus.edu.sg

---

**Alzheimer's Disease Neuroimaging Initative**

**Clifford R. Jack Jr**[36]

[36]Mayo Clinic, Rochester, MN, USA.

# Methods

## Datasets, phenotypes and participants

Following previous studies, we considered 58 HCP phenotypes[59,60] and 36 ABCD phenotypes[15,39]. We also consider a cognition factor score derived from all phenotypes from each dataset[31], yielding a total of 59 HCP and 37 ABCD phenotypes (Supplementary Table 4).

In this study, we used resting-state fMRI from the HCP WU-Minn S1200 release. We filtered participants from a previously reported set of 953 participants[60], excluding participants who did not have at least 40 min of uncensored data (censoring criteria are discussed in the 'Image processing' section) or did not have the full set of the 59 non-brain-imaging phenotypes (hereafter, phenotypes) that we investigated. This resulted in a final set of 792 participants of whom the demographics are described in Supplementary Table 5. The HCP data collection was approved by a consortium of institutional review boards (IRBs) in the USA and Europe, led by Washington University in St Louis and the University of Minnesota (WU-Minn HCP Consortium).

We also considered resting-state fMRI from the ABCD 2.0.1 release. We filtered participants from a previously reported set of 5,260 participants[15]. We excluded participants who did not have at least 15 min of uncensored resting-fMRI data (censoring criteria are discussed in the 'Image processing' section) or did not have the full set of the 37 phenotypes that we investigated. This resulted in a final set of 2,565 participants of whom the demographics are described in Supplementary Table 5. Most ABCD research sites relied on a central IRB at the University of California, San Diego, for the ethical review and approval of the research protocol, while the others obtained local IRB approval.

We also used resting-state fMRI from the SINGER baseline cohort. We filtered participants from an initial set of 759 participants, excluding participants who did not have at least 10 min of resting-fMRI data or did not have the full set of the 19 phenotypes that we investigated (Supplementary Table 4). This resulted in a final set of 642 participants of whom the demographics described in Supplementary Table 5. The SINGER study has been approved by the National Healthcare Group Domain-Specific Review Board and is registered under ClinicalTrials.gov (NCT05007353) with written informed consent obtained from all participants before enrolment into the study.

We used resting-state fMRI from the TCP dataset. We filtered participants from an initial set of 241 participants, excluding participants who did not have at least 26 min of resting-fMRI data or did not have the full set of the 19 phenotypes that we investigated (Supplementary Table 4). This resulted in a final set of 194 participants of whom the demographics are described in Supplementary Table 5. The participants from the TCP study provided written informed consent following guidelines established by the Yale University and McLean Hospital (Partners Healthcare) IRBs.

We used resting-state fMRI from the MDD dataset. We filtered participants from an initial set of 306 participants. We excluded participants who did not have at least 23 min of resting-fMRI data or did not have the full set of the 20 phenotypes that we investigated (Supplementary Table 4). This resulted in a final set of 287 participants of whom the demographics are described in Supplementary Table 5. The MDD dataset was collected from multiple rTMS clinical trials, and all data were obtained at the pretreatment stage. These trials include ChiCTR2300067671 (approved by the Institutional Review Boards of Beijing Anding Hospital, Henan Provincial People's Hospital, and Tianjin Medical University General Hospital); NCT05842278, NCT05842291 and NCT06166082 (all approved by the IRB of Beijing HuiLongGuan Hospital); and NCT06095778 (approved by the IRB of the Affiliated Brain Hospital of Guangzhou Medical University).

We used resting-state fMRI from the ADNI datasets (ADNI 2, ADNI 3 and ADNI GO). We filtered participants from an initial set of 768 participants with both fMRI and PET scans acquired within 1 year of each other. We excluded participants who did not have at least 9 min of resting-fMRI data or did not have the full set of the six phenotypes that we investigated (Supplementary Table 4). This resulted in a final set of 586 participants of whom the demographics are described in Supplementary Table 5. The ADNI study was approved by the IRBs of all participating institutions with informed written consent from all participants at each site.

Moreover, we considered task-fMRI from the ABCD 2.0.1 release. We filtered participants from a previously described set of 5,260 participants[15]. We excluded participants who did not have all three task-fMRI data remaining after quality control, or did not have the full set of the 37 phenotypes that we investigated. This resulted in a final set of 2,262 participants, of whom the demographics are described in Supplementary Table 5.

## Image processing

For the HCP dataset, the MSMAll ICA-FIX resting state scans were used[61]. Global signal regression (GSR) has been shown to improve behavioural prediction[60], so we further applied GSR and censoring, consistent with our previous studies[16,60,62]. The censoring process entailed flagging frames with either FD (framewise displacement) > 0.2 mm or DVARS (differential variance) > 75. The frames immediately before and after flagged frames were marked as censored. Moreover, uncensored segments of data consisting of less than five frames were also censored during downstream processing.

For the ABCD dataset, the minimally processed resting state scans were used[63]. Processing of functional data was performed consistent with our previous study[39]. Specifically, we additionally processed the minimally processed data with the following steps. (1) The functional images were aligned to the T1 images using boundary-based registration[64]. (2) Respiratory pseudomotion filtering was performed by applying a bandstop filter of 0.31–0.43 Hz (ref. 65). (3) Frames with FD > 0.3 mm or DVARS > 50 were flagged. The flagged frame, as well as the frame immediately before and two frames immediately after the marked frame were censored. Furthermore, uncensored segments of data consisting of less than five frames were also censored. (4) Global, white matter and ventricular signals, six motion parameters and their temporal derivatives were regressed from the functional data. Regression coefficients were estimated from uncensored data. (5) Censored frames were interpolated with the Lomb–Scargle periodogram[66]. (6) The data underwent bandpass filtering (0.009–0.08 Hz). (7) Lastly, the data were projected onto FreeSurfer fsaverage6 surface space and smoothed using a 6 mm full-width half-maximum kernel. Task-fMRI data were processed in the same way as the resting-state fMRI data.

For the SINGER dataset, we processed the functional data with the following steps. (1) Removal of the first four frames. (2) Slice time correction. (3) Motion correction and outlier detection: frames with FD > 0.3 mm or DVARS > 60 were flagged as censored frames. 1 frame before and 2 frames after these volumes were flagged as censored frames. Uncensored segments of data lasting fewer than five contiguous frames were also labelled as censored frames. Runs with over half of the frames censored were removed. (4) Correcting for susceptibility-induced spatial distortion. (5) Multi-echo denoising[67]. (6) Alignment with structural image using boundary-based registration[64]. (7) Global, white matter and ventricular signals, six motion parameters and their temporal derivatives were regressed from the functional data. Regression coefficients were estimated from uncensored data. (8) Censored frames were interpolated with the Lomb–Scargle periodogram[66]. (9) The data underwent bandpass filtering (0.009–0.08 Hz). (10) Lastly, the data were then projected onto FreeSurfer fsaverage6 surface space and smoothed using a 6 mm full-width half-maximum kernel.

For the TCP dataset, the details of data processing can be found elsewhere[68]. In brief, the functional data were processed by following the HCP minimal processing pipeline with ICA-FIX, followed by GSR. The processed data were then projected onto MNI space.

For the MDD dataset, we processed the functional data with the following steps. (1) Slice time correction. (2) Motion correction. (3) Normalization for global mean signal intensity. (4) Alignment with structural image using boundary-based registration[64]. (5) Linear detrending and bandpass filtering (0.01–0.08 Hz). (6) Global, white matter and ventricular signals, six motion parameters and their temporal derivatives were regressed from the functional data. (7) Lastly, the data were then projected onto FreeSurfer fsaverage6 surface space and smoothed using a 6 mm full-width half-maximum kernel.

For the ADNI dataset, we processed the functional data with the following steps. (1) Slice time correction. (2) Motion correction. (3) Alignment with structural image using boundary-based registration[64]. (4) Global, white matter and ventricular signals, six motion parameters, and their temporal derivatives were regressed from the functional data. (5) Lastly, the data were then projected onto FreeSurfer fsaverage6 surface space and smoothed using a 6 mm full-width half-maximum kernel.

We derived a 419 × 419 RSFC matrix for each participant of each dataset using the first $T$ minutes of scan time. The 419 regions consisted of 400 parcels from the Schaefer parcellation[30], and 19 subcortical regions of interest[69]. For the HCP, ABCD and TCP datasets, $T$ was varied from 2 to the maximum scan time in intervals of 2 min. This resulted in 29 RSFC matrices per participant in the HCP dataset (generated from using the minimum amount of 2 min to the maximum amount of 58 min in intervals of 2 min), 10 RSFC matrices per participant in the ABCD dataset (generated from using the maximum amount of 2 min to the maximum amount of 20 min in intervals of 2 min) and 13 RSFC matrices per participant in the TCP dataset (generated from using the minimum amount of 2 min to the maximum amount of 26 min in intervals of 2 min).

In the case of the MDD dataset, the total scan time was an odd number (23 min), so $T$ was varied from 3 to the maximum of 23 min in intervals of 2 min, which resulted in 11 RSFC matrices per participant. For SINGER, ADNI and ABCD task-fMRI data, as the scans were relatively short (around 10 min), $T$ was varied from 2 min the maximum scan time in intervals of 1 min. This resulted in 9 RSFC matrices per participant in the SINGER datasets (generated from using the minimum amount of 2 min to the maximum amount of 10 min), 8 RSFC matrices per participant in the ADNI datasets (generated from using the minimum amount of 2 min to the maximum amount of 9 min), 9 RSFC matrices per participant in the ABCD N-back task (from using the minimum amount of 2 min to the maximum amount of 9.65 min), 11 RSFC matrices per participant in the ABCD SST task (from using the minimum amount of 2 min to the maximum amount of 11.65 min) and 10 RSFC matrices per participant in the ABCD MID task (from using the minimum amount of 2 min to the maximum amount of 10.74 min).

We note that the above preprocessed data were collated across multiple laboratories and, even within the same laboratory, datasets were processed by different individuals many years apart. This led to significant preprocessing heterogeneity across datasets. For example, raw FD was used in the HCP dataset because it was processed many years ago, while the more recently processed ABCD dataset used a filtered version of FD, which has been shown to be more effective. Another variation is that some datasets were projected to fsaverage space, while other datasets were projected to MNI152 or fsLR space.

## Prediction workflow

The RSFC generated from the first $T$ minutes was used to predict each phenotypic measure using KRR[16] with an inner-loop (nested) cross-validation procedure.

Let us illustrate the procedure using the HCP dataset (Extended Data Fig. 1). We began with the full set of participants. A tenfold nested cross-validation procedure was used. The participants were divided in ten folds (Extended Data Fig. 1 (first row)). We note that care was taken so siblings were not split across folds, so the ten folds were not exactly the same sizes. For each of ten iterations, one fold was reserved for

testing (that is, test set), and the remainder was used for training (that is, the training set). As there were 792 HCP participants, the training set size was roughly 792 × 0.9 ≈ 700 participants. The KRR hyperparameter was selected through a tenfold cross-validation of the training set. The best hyperparameter was then used to train a final KRR model in the training set and applied to the test set. Prediction accuracy was measured using Pearson's correlation and COD[39].

The above analysis was repeated with different training set sizes achieved by subsampling each training fold (Extended Data Fig. 1 (second and third rows)), while the test set remained identical across different training set sizes, so the results are comparable across different training set sizes. The training set size was subsampled from 200 to 600 (in intervals of 100). Together with the full training set size of approximately 700 participants, there were 6 different training set sizes, corresponding to 200, 300, 400, 500, 600 and 700.

The whole procedure was repeated with different values of $T$. As there were 29 values of $T$, there were in total 29 × 6 sets of prediction accuracies for each phenotypic measure. To ensure robustness, the above procedure was repeated 50 times with different splits of the participants into ten folds to ensure stability (Extended Data Fig. 1). The prediction accuracies were averaged across all test folds and all 50 repetitions.

The procedure for the other datasets followed the same principle as the HCP dataset. However, the ABCD (rest and task) and ADNI datasets comprised participants from multiple sites. Thus, following our previous studies[31,39], we combined ABCD participants across the 22 imaging sites into 10 site-clusters and combined ADNI participants across the 71 imaging sites into 20 site-clusters (Supplementary Table 5). Each site-cluster has at least 227, 156 and 29 participants in the ABCD (rest), ABCD (task) and ADNI datasets respectively.

Instead of the tenfold inner-loop (nested) cross-validation procedure in the HCP dataset, we performed a leave-three-site-clusters-out inner-loop (nested) cross-validation (that is, seven site-clusters are used for training and three site-clusters are used for testing) in the ABCD rest and task datasets. The hyperparameter was again selected using a tenfold CV within the training set. This nested cross-validation procedure was performed for every possible split of the site clusters, resulting in 120 replications. The prediction accuracies were averaged across all 120 replications.

We did not perform a leave-one-site-cluster-out procedure because the site-clusters are 'fixed', so the cross-validation procedure can only be repeated ten times under a leave-one-site-cluster-out scenario (instead of 120 times). Similarly, we did not go for leave-two-site-clusters-out procedure because that will only yield a maximum of 45 repetitions of cross-validation. On the other hand, if we left more than three site clusters out (for example, leave-five-site-clusters-out), we could achieve more cross-validation repetitions, but at the cost of reducing the maximum training set size. We therefore opted for the leave-three-site-clusters-out procedure, consistent with our previous study[39].

To be consistent with the ABCD dataset, for the ADNI dataset, we also performed a leave-three-site-clusters-out inner-loop (nested) cross-validation procedure. This procedure was performed for every possible split of the site clusters, resulting in 1,140 replications. The prediction accuracies were averaged across all 1,140 replications.

We also performed tenfold inner-loop (nested) cross-validation procedure in the TCP, MDD and SINGER datasets. Although the data from the TCP and MDD datasets were acquired from multiple sites, the number of sites was much smaller (2 and 5, respectively) than that of the ABCD and ADNI datasets. We were therefore unable to use the leave-some-site-out cross-validation strategy because that would reduce the training set size by too much. We therefore ran a tenfold nested cross-validation strategy (similar to the HCP). However, we regress sites from the target phenotype in the training set, which were then applied to the test set. In other words, our prediction was

performed on the residuals of phenotypes after site regression. Site regression was unnecessary for the SINGER dataset as the data were collected from only a single site. The rest of the prediction workflow was the same as the HCP dataset, except for the number of repetitions. As TCP, MDD and SINGER datasets had smaller sample sizes than the HCP dataset, the tenfold cross-validation was repeated 350 times. The prediction accuracies were averaged across all test folds and all repetitions.

Similar to the HCP, the analyses were repeated with different numbers of training participants, ranging from 200 to 1,600 ABCD (rest) participants (in intervals of 200). Together with the full training set size of approximately 1,800 participants, there were 9 different training set sizes. The whole procedure was repeated with different values of $T$. As there were 10 values of $T$ in the ABCD (rest) dataset, there were in total $10 \times 9$ values of prediction accuracies for each phenotype. In the case of ABCD (task), the sample size was smaller with maximum training set size of approximately 1,600 participants, so there were only eight different training set sizes.

The ADNI and SINGER datasets had less participants than the HCP dataset, so we decided to sample the training set size more finely. More specifically, we repeated the analyses by varying the number of training participants from the minimum sample size of 100 to the maximum sample size in intervals of 100. For SINGER, the full training set size is around 580 participants, so there were 6 different training set sizes in total (100, 200, 300, 400, 500 and ~580). For ADNI, the full training set size is around 530, so there were also 6 different training set sizes in total (100, 200, 300, 400, 500 and ~530).

Finally, TCP and MDD datasets were the smallest, so the training set size was sampled even more finely. More specifically, we repeated the analyses by varying the number of training participants from the minimum sample size of 50 to the maximum sample size in intervals of 25. For TCP, the full training set size is ~175, so there 6 training set sizes in total (50, 75, 100, 125, 150 and 175). For MDD, the full training set size is ~258, so there 10 training set sizes in total (50, 75, 100, 125, 150, 175, 200, 225, 250 and 258).

Current best MRI practices suggest that the model hyperparameter should be optimized[70], so in the current study, we did not consider the case where the hyperparameter was fixed. As an aside, we note that for all analyses, the best hyperparameter was selected using a tenfold cross-validation within the training set. The best hyperparameter was then used to train the model on the full training set. Thus, the full training set was used for hyperparameter selection and for training the model. Furthermore, we needed to select only one hyperparameter, while training the model required fitting many more parameters. We therefore do not expect the hyperparameter selection to be more dependent on the training set size than training the actual model itself.

We also note that our study focused on out-of-sample prediction within the same dataset, but did not explore cross-dataset prediction[71]. For predictive models to be clinically useful, these models must generalize to completely new datasets. The best way to achieve this goal is by training models from multiple datasets jointly, so as to maximize the diversity of the training data[72,73]. However, we did not consider cross-dataset prediction in the current study because most studies are not designed with the primary aim of combining the collected data with other datasets.

A full table of prediction accuracies for every combination of sample size and scan time per participant is provided in the Supplementary Information.

## Fitting the logarithmic model

By plotting prediction accuracy against total scan duration (number of training participants × scan duration per participant) for each phenotypic measure, we observed diminishing returns of scan time (relative to sample size), especially beyond 20 min per participant.

Furthermore, visual inspection suggests that a logarithmic curve might fit well to each phenotypic measure when scan time per participant is 20 min or less. To explore the universality of a logarithmic relationship between total scan duration and prediction accuracy, for each phenotypic measure $p$, we fitted the function $y_p = z_p \log_2(t_p) + k_p$, where $y_p$ was the prediction accuracy for phenotypic measure $p$, and $t_p$ is the total scan duration. $z_p$ and $k_p$ were estimated from data by minimizing the square error, yielding $\hat{z}_p$ and $\hat{k}_p$.

In addition to fitting the logarithmic curve to different phenotypic measures, the fitting can also be performed with different prediction accuracy measures (Pearson's correlation or COD) and different predictive models (KRR and LRR). Assuming the datapoints are well explained by the logarithmic curve, the normalized accuracies $(y_p - \hat{k}_p)/\hat{z}_p$ should follow a standard $\log_2(t)$ curve across phenotypic measures, prediction accuracies, predictive models and datasets. For example, Supplementary Fig. 9a shows the normalized prediction performance of the cognitive factors for different prediction accuracy measures (Pearson's correlation or COD) and different predictive models (KRR and LRR) across HCP and ABCD datasets.

Here we have chosen to use KRR and linear regression because previous studies have shown that they have comparable prediction performance, and also exhibited similar prediction accuracies as several deep neural networks[16,39]. Indeed, a recent study suggested that linear dynamical models provide a better fit to resting-state brain dynamics (as measured by fMRI and intracranial electroencephalogram) than nonlinear models, suggesting that, due to the challenges of in vivo recordings, linear models might be sufficiently powerful to explain macroscopic brain measurements. However, we note that, in the current study, we are not making a similar claim. Instead, our results suggest that the trade-off between scan time and sample size are similar for different regression models, and phenotypic domains, scanners, acquisition protocols, racial groups, mental disorders, age groups, as well as resting-state and task-state functional connectivity.

## Fitting the theoretical model

We observed that sample size and scan time per participant did not contribute equally to prediction accuracy, with sample size having a more important role than scan time. To explain this observation, we derived a mathematical relationship relating the expected prediction accuracy (Pearson's correlation) between noisy brain measurements and non-brain-imaging phenotype with scan time and sample size.

Based on a linear regression model with no regularization and assumptions including (1) stationarity of fMRI (that is, autocorrelation in fMRI is the same at all timepoints), and (2) prediction errors are uncorrelated with errors in brain measurements, we found that

$$E(\hat{\rho}) \approx K_0 \sqrt{\frac{1}{1 + \frac{K_1}{N} + \frac{K_2}{NT}}},$$

where $E(\hat{\rho})$ is the expected correlation between the predicted phenotype estimated from noisy brain measurements and the observed phenotype. $K_0$ is related to the ideal association between brain measurements and phenotype, attenuated by phenotypic reliability. $K_1$ is related to the noise-free ideal association between brain measurements and phenotype. $K_2$ is related to brain–phenotype prediction errors due to brain measurement inaccuracies. Full derivations are provided in Supplementary Methods 1.1 and 1.2.

On the basis of the above equation, we fitted the following function $y_p = K_{0,p} \sqrt{\frac{1}{1 + K_{1,p}/N + K_{2,p}/(NT)}}$, where $y_p$ is the prediction accuracy for phenotypic measure $p$, $N$ is the sample size and $T$ is the scan time per participant. $K_{0,p}, K_{1,p}$ and $K_{2,p}$ were estimated by minimizing the mean

squared error between the above function and actual observation of $y_p$ using gradient descent.

## Non-stationarity analysis

In the original analysis, FC matrices were generated with increasing time $T$ based on the original run order. To account for the possibility of fMRI-phenotype non-stationarity effects, we randomized the order in which the runs were considered for each participant. As both the HCP and ABCD datasets contained 4 runs of resting-fMRI, we generated FC matrices from all 24 possible permutations of run order. For each cross-validation split, the FC matrix for a given participant was randomly sampled from 1 of the 24 possible permutations. We note that the randomization was independently performed for each participant.

To elaborate further, let us consider an ABCD participant with the original run order (run 1, run 2, run 3, run 4). Each run was 5 min long. In the original analysis, if scan time $T$ was 5 min, then we used all the data from run 1 to compute FC. If scan time $T$ was 10 min, then we used run 1 and run 2 to compute FC. If scan time $T$ was 15 min, then we used runs 1, 2 and 3 to compute FC. Finally, if scan time $T$ was 20 min, we used all 4 runs to compute FC.

On the other hand, after run randomization, for the purpose of this exposition, let us assume that this specific participant's run order had become run 3, run 2, run 4, run 1. In this situation, if the scan time $T$ was 5 min, then we used all data from run 3 to compute FC. If scan time $T$ was 10 min, then we used run 3 and run 2 to compute FC. If scan time $T$ was 15 min, then we used runs 3, 2 and 4 to compute FC. Finally, if $T$ was 20 min, we used all 4 runs to compute FC.

## Optimizing within a fixed fMRI budget

To generate Extended Data Fig. 6, we note that given a particular scan cost per hour $S$ and overhead cost per participant $O$, the total budget for scanning $N$ participants with $T$ min per participant is given by $(T/60 \times S + O) \times N$. Thus, given a fixed fMRI budget (for example, US\$1 million), scan cost per hour (for example, US\$500) and overhead cost per participant (for example, US\$500), we increase scan time $T$ in 1 min intervals from 1 to 200 and, for each value of $T$, we can find the largest sample size $N$, such that the scan costs stayed within the fMRI budget. For each pair of sample size $N$ and scan time $T$, we can then compute the fraction of maximum accuracy based on Fig. 4a.

## Optimizing to achieve a fixed accuracy

To generate Figs. 4b,c, 5 and 6, suppose we want to achieve 90% of maximum achievable accuracy, we can find all pairs of sample size and scan time per participant along the 0.9 black contour line in Fig. 4a. For every pair of sample size $N$ and scan time $T$, we can then compute the study cost given a particular scan cost per hour $S$ (for example, US\$500) and a particular overhead cost per participant $O$ (for example, US\$1,000): $(T/60 \times S + O) \times N$. The optimal scan time (and sample size) with the lowest study cost can then be obtained.

## Brain-wide association reliability

To explore the reliability of univariate brain-wide association analyses (BWAS)[1], we followed a previously established split-half procedure[14,15].

Let us illustrate the procedure using the HCP dataset (Supplementary Fig. 20a). We began with the full set of participants, which were then divided into ten folds (Supplementary Fig. 20a (first row)). We note that care was taken so siblings were not split across folds, so the ten folds were not exactly the same sizes. The ten folds were divided into two non-overlapping sets of five folds. For each set of five folds and each phenotype, we computed the Pearson's correlation between each RSFC edge and phenotype across participants, yielding a $419 \times 419$ correlation matrix, which was then converted into a $419 \times 419$ $t$-statistic matrix. Split-half reliability between the (lower triangular portions of the symmetric) $t$-statistic matrices from the two

sets of five folds was then computed using the intraclass correlation formula[14,15].

The above analysis was repeated with different sample sizes achieved by subsampling each fold (Supplementary Fig. 20a (second and third rows)). The split-half sample sizes were subsampled from 150 to 350 (in intervals of 50). Together with the full sample size of approximately 800 participants (corresponding to a split-half sample size of around 400), there were 6 split-half sample sizes corresponding to 150, 200, 250, 300, 350 and 400 participants.

The whole procedure was also repeated with different values of $T$. As there were 29 values of $T$, there were in total $29 \times 6$ univariate BWAS split-half reliability values for each phenotype. To ensure robustness, the above procedure was repeated 50 times with different split of the participants into 10 folds to ensure stability (Supplementary Fig. 20a). The reliability values were averaged across all 50 repetitions.

The same procedure was followed in the case of the ABCD dataset, except as previously explained, the ABCD participants were divided into ten site-clusters. Thus, the split-half reliability was performed between two sets of five non-overlapping site-clusters. In total, this procedure was repeated 126 times as there were 126 ways to divide 10 site-clusters into two sets of 5 non-overlapping site-clusters.

Similar to the HCP, the analyses were repeated with different numbers of split-half participants, ranging from 200 to 1,000 ABCD participants (in intervals of 200). Together with the full training set size of approximately 2,400 participants (corresponding to a split-half sample size of approximately 1,200 participants, there were 6 split-half sample sizes, corresponding to 200, 400, 600, 800, 1,000, 1,200.

The whole procedure was also repeated with different values of $T$. As there were 10 values of $T$ in the ABCD dataset, there were in total $10 \times 6$ values univariate BWAS split-half reliability values for each phenotype.

Previous studies have suggested the Haufe-transformed coefficients from multivariate prediction are significantly more reliable than univariate BWAS[14,15]. We therefore repeated the above analyses by replacing BWAS with the multivariate Haufe-transform.

A full table of split-half BWAS reliability for each given combination of sample size and scan time per participant is provided in the Supplementary Information.

## Statistical analyses

Supplementary Tables 1–3 summarize all quantifications and statistical analyses performed in this study. When statistical tests were performed, multiple-comparison correction was performed within each result section using Benjamini–Yekutieli FDR correction with $q < 0.05$ (ref. 74).

## Reporting summary

Further information on research design is available in the Nature Portfolio Reporting Summary linked to this article.

## Data availability

The prediction accuracies for each phenotype, sample size $N$ and scan time $T$ in all nine resting and task fMRI datasets are publicly available (https://github.com/ThomasYeoLab/CBIG/tree/master/stable_projects/predict_phenotypes/Ooi2024_ME). The raw data for HCP (https://www.humanconnectome.org/), ABCD (https://abcdstudy.org/), TCP (https://openneuro.org/datasets/ds005237 and https://nda.nih.gov/edit_collection.html?id=3552) and ADNI (https://ida.loni.usc.edu/) are publicly available. ABCD parcellated time courses can be found on NDA (https://doi.org/10.15154/1528763). HCP and TCP parcellated time courses can be found at Zenodo[75] (https://doi.org/10.5281/zenodo.15300607). The ADNI user agreement does not allow us to share the ADNI derivatives. The SINGER dataset can be obtained through a data-transfer agreement (https://medicine.nus.edu.sg/macc-2/projects/singer/). The MDD dataset is available on request from H.L. (hesheng@biopic.pku.edu.cn).

## Code availability

Code for this study is publicly available in the GitHub repository maintained by the Computational Brain Imaging Group (https://github.com/ThomasYeoLab/CBIG). Processing pipelines of the fMRI data are available at GitHub (https://github.com/ThomasYeoLab/CBIG/tree/master/stable_projects/preprocessing/CBIG_fMRI_Preproc2016). Analyses were conducted in MATLAB (2018b) and Python 3.7. Code specific to the analyses is available at GitHub (https://github.com/ThomasYeoLab/CBIG/tree/master/stable_projects/predict_phenotypes/Ooi2024_ME). Code related to this study was reviewed by S.Z., T.W.K.T. and R.K. to reduce the chance of coding errors.

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

**Acknowledgements** The research in this study is supported by the NUS Yong Loo Lin School of Medicine (NUHSRO/2020/124/TMR/LOA), the Singapore National Medical Research Council (NMRC) LCG (OFLCG19May-0035), NMRC CTG-IIT (CTGIIT23jan-0001), NMRC OF-IRG (OFIRG24jan-0006; OFIRG24jul-0049), NMRC STaR (STaR20nov-0003), Singapore Ministry of Health (MOH) Centre Grant (CG21APR1009), the Temasek Foundation (TF2223-IMH-01), as well as National Institutes of Health (NIH) grants, MH120080 (A.J.H., B.T.T.Y.), MH133334 (B.T.T.Y.), NS140256 (E.M.G., N.U.F.D.), MH129616 (T.O.L.), MH121518 (S.M.), MH121276 (N.U.F.D., E.M.G.), MH096773 (N.U.F.D.), MH122066 (E.M.G., N.U.F.D.), NS131131 (N.U.F.D., S.M.), MH124567 (E.M.G., N.U.F.D.), NS129521 (E.M.G., N.U.F.D.); the Allied Labs for Imaging Guided Neurotherapies (N.U.F.D., S.M., T.O.L., E.M.G.), the Taylor Family Institute Fund for Innovative Psychiatric Research (T.O.L.); the McDonnell Center for Systems Neuroscience (N.U.F.D.); the National Spasmodic Dysphonia Association (E.M.G.); the Intellectual and Developmental Disabilities Research Center (N.U.F.D.); the Kiwanis Foundation (N.U.F.D.); the Washington University Hope Center for Neurological Disorders (E.M.G., N.U.F.D.). Any opinions, findings and conclusions or recommendations expressed in this material are those of the authors and do not reflect the views of the funders. Data were provided (in part) by the HCP, WU-Minn Consortium (Principal Investigators: D. Van Essen and K. Ugurbil, 1U54MH091657) funded by the 16 NIH Institutes and Centers that support the NIH Blueprint for Neuroscience Research; and by the McDonnell Center for Systems Neuroscience at Washington University. Data used in the preparation of this Article were obtained from the Adolescent Brain Cognitive DevelopmentSM (ABCD) Study (https://abcdstudy.org), held in the NIMH Data Archive (NDA). This is a multisite, longitudinal study designed to recruit more than 10,000 children aged 9–10 years and follow them over 10 years into early adulthood. The ABCD Study is supported by the National Institutes of Health and additional federal partners under award numbers U01DA041048, U01DA050989, U01DA051016, U01DA041022, U01DA051018, U01DA051037, U01DA050987, U01DA041174, U01DA041106, U01DA041117, U01DA041028, U01DA041134, U01DA050988, U01DA051039, U01DA041156, U01DA041025, U01DA041120, U01DA051038, U01DA041148, U01DA041093, U01DA041089, U24DA041123 and U24DA041147. A full list of supporters is available online (https://abcdstudy.org/federal-partners.html). A listing of participating sites and a complete listing of the study investigators can be found online (https://abcdstudy.org/consortium_members/). ABCD consortium investigators designed and implemented the study and/or provided data but did not necessarily participate in the analysis or writing of this report. This Article reflects the views of the authors and may not reflect the opinions or views of the NIH or ABCD consortium investigators. The ABCD data repository grows and changes over time. The ABCD data used in this report came from https://doi.org/10.15154/1504041. Data collection and sharing for the Alzheimer's Disease Neuroimaging Initiative (ADNI) is funded by the National Institute on Aging (National Institutes of Health Grant U19AG024904). The grantee organization is the Northern California Institute for Research and Education. In the past, ADNI has also received funding from the National Institute of Biomedical Imaging and Bioengineering, the Canadian Institutes of Health Research and private sector contributions through the Foundation for the National Institutes of Health (FNIH), including contributions from the following organizations: AbbVie, Alzheimer's Association; Alzheimer's Drug Discovery Foundation; Araclon Biotech; BioClinica; Biogen; BristolMyers Squibb Company; CereSpir; Cogstate; Eisai; Elan Pharmaceuticals; Eli Lilly and Company; EuroImmun; F. Hoffmann-La Roche and its affiliated company Genentech; Fujirebio; GE Healthcare; IXICO; Janssen Alzheimer Immunotherapy Research & Development; Johnson & Johnson Pharmaceutical Research & Development; Lumosity; Lundbeck; Merck; Meso Scale Diagnostics; NeuroRx Research; Neurotrack Technologies; Novartis Pharmaceuticals; Pfizer; Piramal Imaging; Servier; Takeda Pharmaceutical Company; and Transition Therapeutics. Data used in preparation of this article were obtained from the Alzheimer's Disease Neuroimaging Initiative (ADNI) database (https://adni.loni.usc.edu). As such, the investigators within the ADNI contributed to the design and implementation of ADNI and/or provided data but did not participate in the analysis or writing of this report. A complete listing of ADNI investigators is available online (http://adni.loni.usc.edu/wp-content/uploads/how_to_apply/ADNI_Acknowledgement_List.pdf).

**Author contributions** L.Q.R.O., C.O., S.Z., R.K., N.U.F.D., T.O.L., E.M.G., T.E.N., A.J.H. and B.T.T.Y. conceptualized the study and designed the methodology. L.Q.R.O. and R.K. preprocessed the HCP and ABCD datasets. K.H.Y., F.J. and J.S.X.C. preprocessed the SINGER dataset. S.C. and C.C. preprocessed the TCP dataset. Q.H., J.R. and H.L. preprocessed the MDD dataset. N.F. and S.N.R.-C. preprocessed the ADNI dataset. L.Q.R.O. carried out the analysis in the HCP and ABCD datasets. S.Z. carried out the analysis in the SINGER, TCP and ADNI datasets. Q.H. carried out the analysis in the MDD dataset. T.E.N. derived the theoretical models in the study. L.Q.R.O., C.O., S.Z., T.E.N. and B.T.T.Y. designed and executed statistical analyses across datasets. S.Z., T.W.K.T. and R.K. reviewed the code used in the study. L.Q.R.O., C.O. and B.T.T.Y. wrote the original draft. L.Q.R.O., C.O., S.Z., T.E.N., T.W.K.T., R.K., S.M., N.U.F.D., T.O.L., E.M.G., K.H.Y., F.J., J.S.X.C., C.C., L.A., N.F., S.N.R.-C., Q.H., J.R., H.L., S.C., C.V.C., J.T.B., J.H.Z., D.B., S.B.E., A.J.H. and B.T.T.Y. reviewed and edited the final manuscript. Consortium contributions: data collected by the Alzheimer's Disease Neuroimaging Initiative were used for the analyses in this study. A list of contributing authors to the consortium is provided in the Supplementary Information.

**Competing interests** D.B. is shareholder and advisory board member of MindState Design Labs. L.Q.R.O. and R.K. are co-founders of B1Neuro. B.T.T.Y. is a shareholder of B1Neuro. N.U.F.D. is a co-founder of Turing Medical. H.L. is a co-founder of Neural Galaxy. T.O.L. holds a patent for taskless mapping of brain activity licensed to Sora Neurosciences and a patent for optimizing targets for neuromodulation, implant localization and ablation. E.M.G. and N.U.F.D. may receive royalty income based on technology developed at Washington University School of Medicine and licensed to Turing Medical. T.O.L. and E.M.G. are consultants for Turing Medical. The content in this manuscript is unrelated to the activities of these companies.

**Additional information**
**Correspondence and requests for materials** should be addressed to B. T. Thomas Yeo.

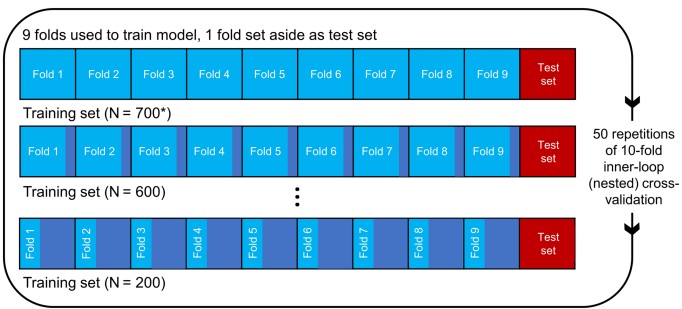

**Extended Data Fig. 1 | Prediction workflow for the HCP dataset.** Participants were split into 10 folds. One fold was set aside to be the test set. The remaining folds comprised the training set. Cross-validation was performed on the training set to select the best hyperparameter. The best hyperparameter was then used to fit a final model from the full training set, which was then used to predict phenotypes in the test set. To vary training set size, each training fold was subsampled and the whole inner-loop nested cross-validation procedure was repeated with the resulting smaller training set. As shown in the panel, the test set remained the same across different training set sizes, so that prediction accuracies were comparable across different sample sizes. Each fold took a turn to be the test set (i.e., 10-fold inner-loop nested cross-validation) and the procedure was repeated with different amounts of fMRI data per participant $T$ (not shown in panel). For stability, the entire procedure was repeated 50 times and averaged. A similar workflow was used in the ABCD dataset (see Methods, "Prediction workflow"). We note that in the case of HCP, care was taken so siblings were not split across folds, while in the case of ABCD, participants from the same site were not split across folds.

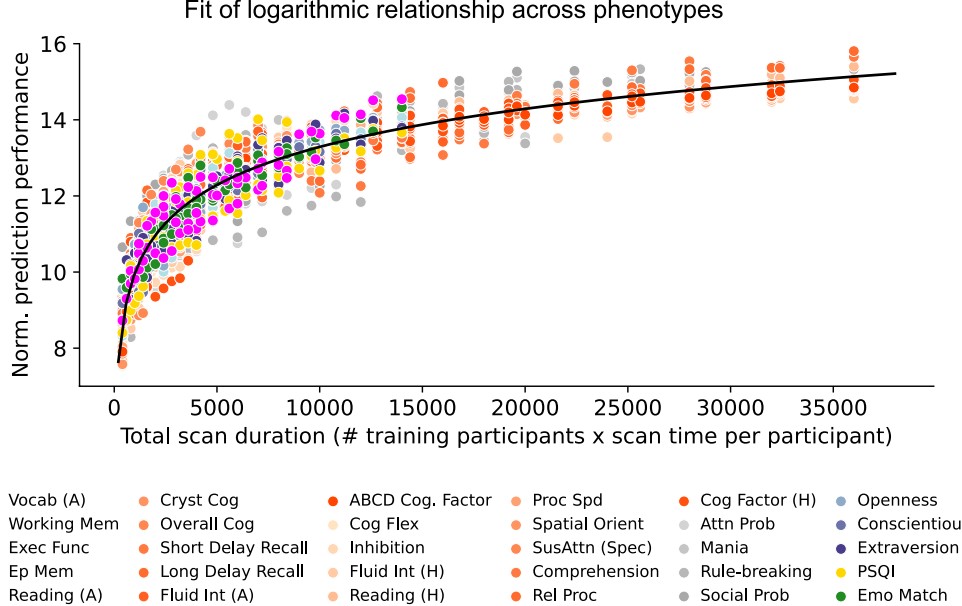

**Extended Data Fig. 2 | Relationship between prediction accuracy and total scan duration (sample size × scan time per participant).** Scatter plot showing normalized prediction accuracy of the two cognitive factor scores and 34 other phenotypes versus total scan duration, ignoring data beyond 20 min of scan time per participant. Black curve shows the logarithmic fit. There are 2280 data points (dots) in this figure.

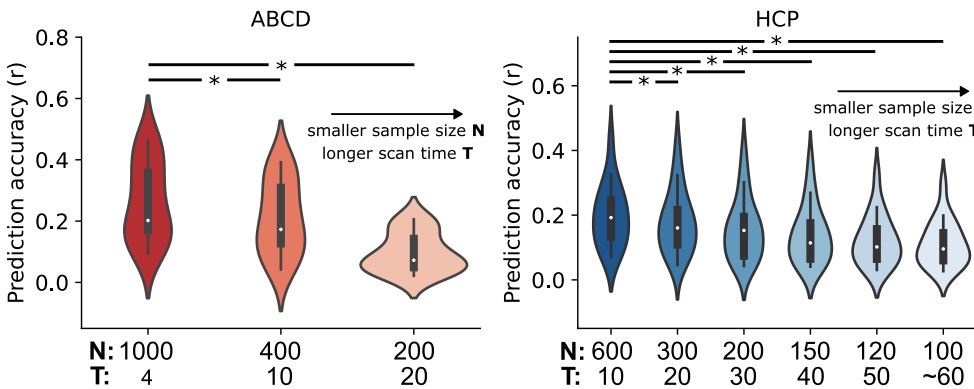

**Extended Data Fig. 3 | As scan time increases, sample size becomes more important than scan time.** Same as Fig. 3a, except in the left panel, each violin plot shows the distribution of average prediction accuracies across 17 ABCD phenotypic measures. Each violin contains 17 data points and has a total scan duration of 4,000 min. In the right panel, each violin shows the distribution of average prediction accuracies across the 19 HCP phenotypic measures. Each violin contains 19 data points and has a total scan duration of 6,000 min. Boxes inside violins represent interquartile range (IQR, from the 25th to 75th percentile), and whiskers extend to the most extreme data points not considered outliers (within 1.5 × IQR). Two-tailed paired-sample t-test was performed between the largest sample size and the other sample sizes. Exact P values and details of statistical test can be found in Supplementary Table 1. The asterisks indicate that the prediction accuracies were significantly different after false discovery rate (FDR) q < 0.05 correction.

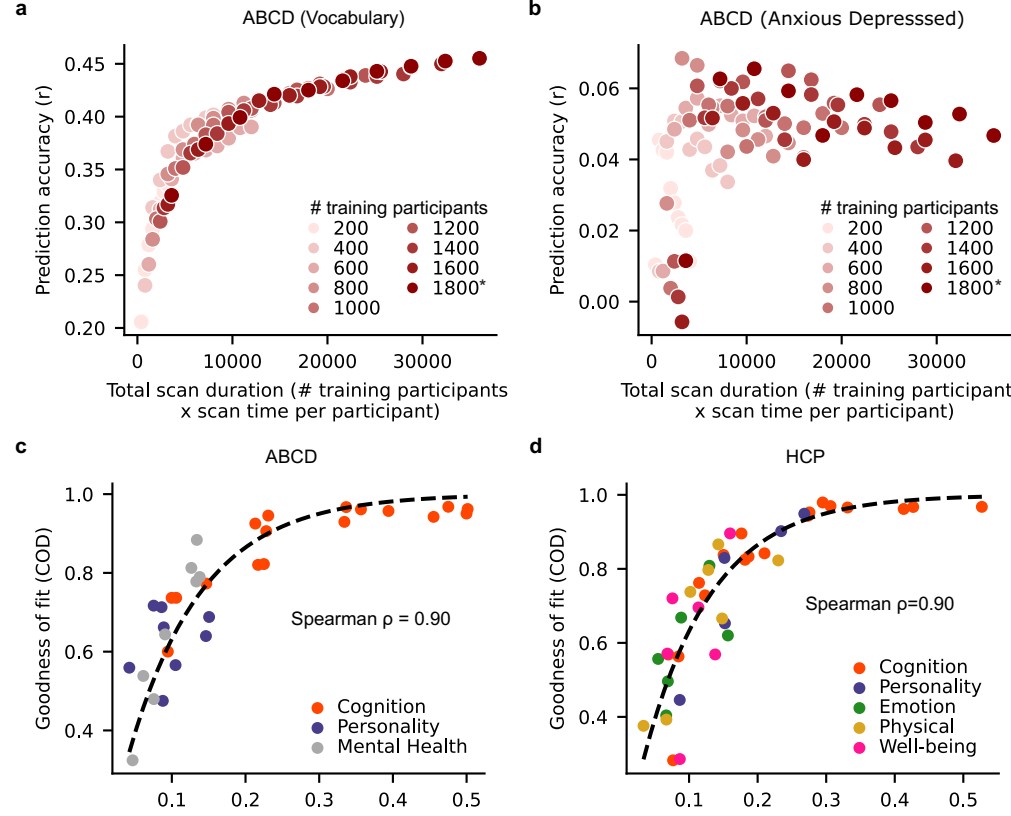

**Extended Data Fig. 4 | Theoretical model works better for well-predicted phenotypes. a**. Scatter plot of prediction accuracy against total scan duration for an exemplary phenotype with high prediction accuracy. There are 90 data points (dots) in this plot. **b**. Scatter plot of prediction accuracy against total scan duration for an exemplary phenotype with low prediction accuracy. There are 90 data points (dots) in this plot. **c**. Scatter plot of theoretical model goodness-of-fit (coefficient of determination or COD) against prediction accuracies of different ABCD phenotypes. COD (also known as $R^2$) is a measure of explained variance. Here, we considered phenotypes whose prediction accuracies

(Pearson's r) were positive in at least 90% of all combinations of sample size $N$ and scan time $T$, yielding 42 HCP phenotypes and 33 ABCD phenotypes. Prediction accuracy (horizontal axis) was based on maximum scan time and sample size. For visualization, we plot a dashed black line by fitting to a monotonically increasing function. **d**. Same as **c** but using HCP (instead of ABCD) phenotypes. For **c** and **d**, Spearman's correlation was computed based on 33 ABCD phenotypes and 42 HCP phenotypes respectively. P values were obtained via a permutation test (see Supplementary Table 1).

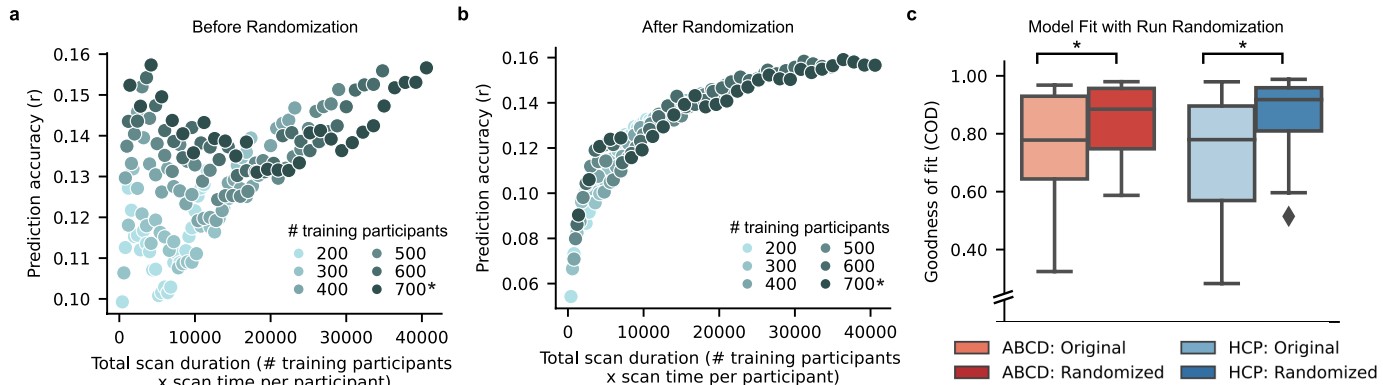

**Extended Data Fig. 5 | Non-stationarity in fMRI-phenotype relationship weakens adherence to theoretical model. a.** Scatter plot of prediction accuracy against total scan duration for the "Anger: Aggression" phenotype in the HCP dataset. There are 174 data points (dots) in this plot. Despite relatively high accuracy, prediction accuracy increases with larger sample sizes (Spearman's ρ = 1.00), while extending scan duration does not generate a similarly consistent effect (Spearman's ρ = 0.21; Supplementary Table 1). **b.** Scatter plot of prediction accuracy against total scan duration for the "Anger: Aggression" phenotype in the HCP dataset after randomizing fMRI run order for each participant. There are 174 data points (dots) in this plot. Observe that the prediction accuracy now adheres strongly to the theoretical model (Supplementary Table 1). **c.** Box plots showing goodness of fit to theoretical model before and after randomizing fMRI run order. Here, we considered all phenotypes whose prediction accuracies

(Pearson's r) were positive in at least 90% of all combinations of $N$ and $T$, so we ended up with 33 ABCD and 42 HCP phenotypes. Therefore, each ABCD boxplot contains 33 data points, while each HCP boxplot contains 42 datapoints. For each boxplot, the horizontal line indicates the median across 33 ABCD phenotypes or 42 HCP phenotypes. The bottom and top edges of the box indicate the 25th and 75th percentiles, respectively. Outliers are defined as data points beyond 1.5 times the interquartile range. The whiskers extend to the most extreme data points not considered outliers. Two-tailed paired-sample t-tests were used to test whether COD was improved after run randomization. P values were 8.80e-5 (ABCD) and 2.23e-7 (HCP). * indicates that goodness-of-fit was significantly different after FDR correction with q < 0.05. More details of statistical tests can be found in Supplementary Table 1.

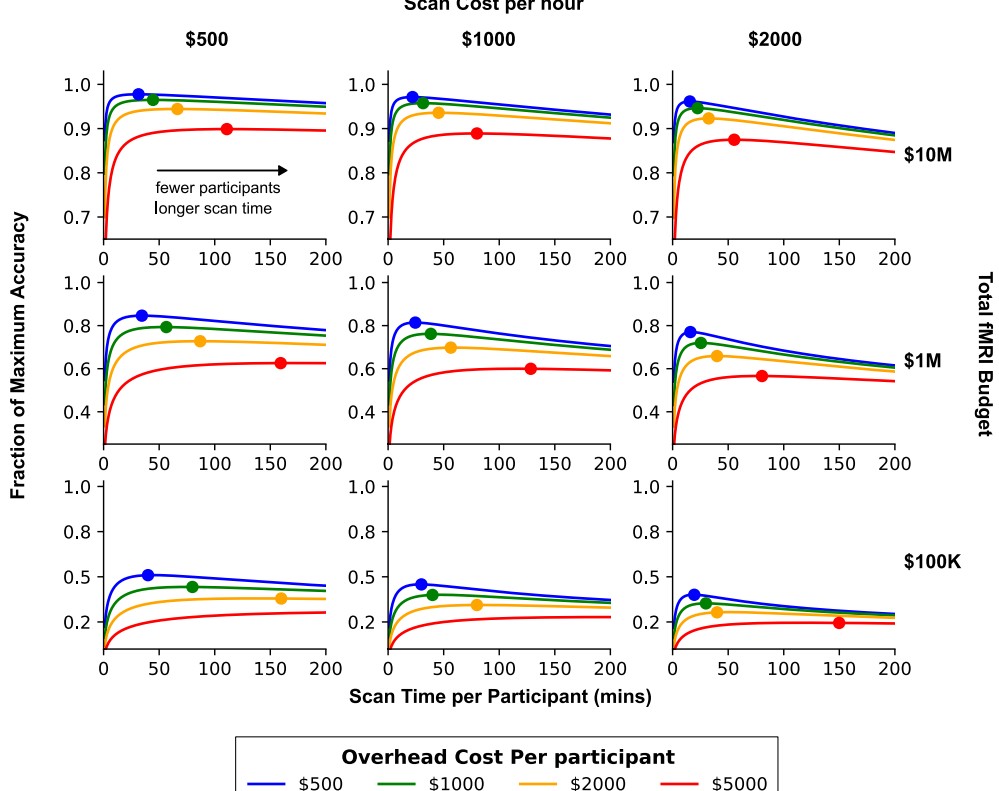

**Extended Data Fig. 6 | Fraction of maximum achievable prediction accuracy as a function of total fMRI budget, scan cost per hour and overhead cost per participant.** The three columns correspond to scan cost per hour of US$500, US$1,000 and US$2,000 respectively. The three rows correspond to total fMRI budget of US$10 million, US$1 million and US$100,000 respectively. The different coloured lines correspond to different overhead cost per participant. Each curve shows the fraction of maximum prediction accuracy as a function of scan time per participant for a given overhead cost per participant and scan cost per hour, while keeping within the total fMRI budget. On each curve, the solid circle indicates the location of maximum prediction accuracy. Circles are not shown if optimal scan time was beyond the edge of the graph (i.e., more than 200 min of scan time).

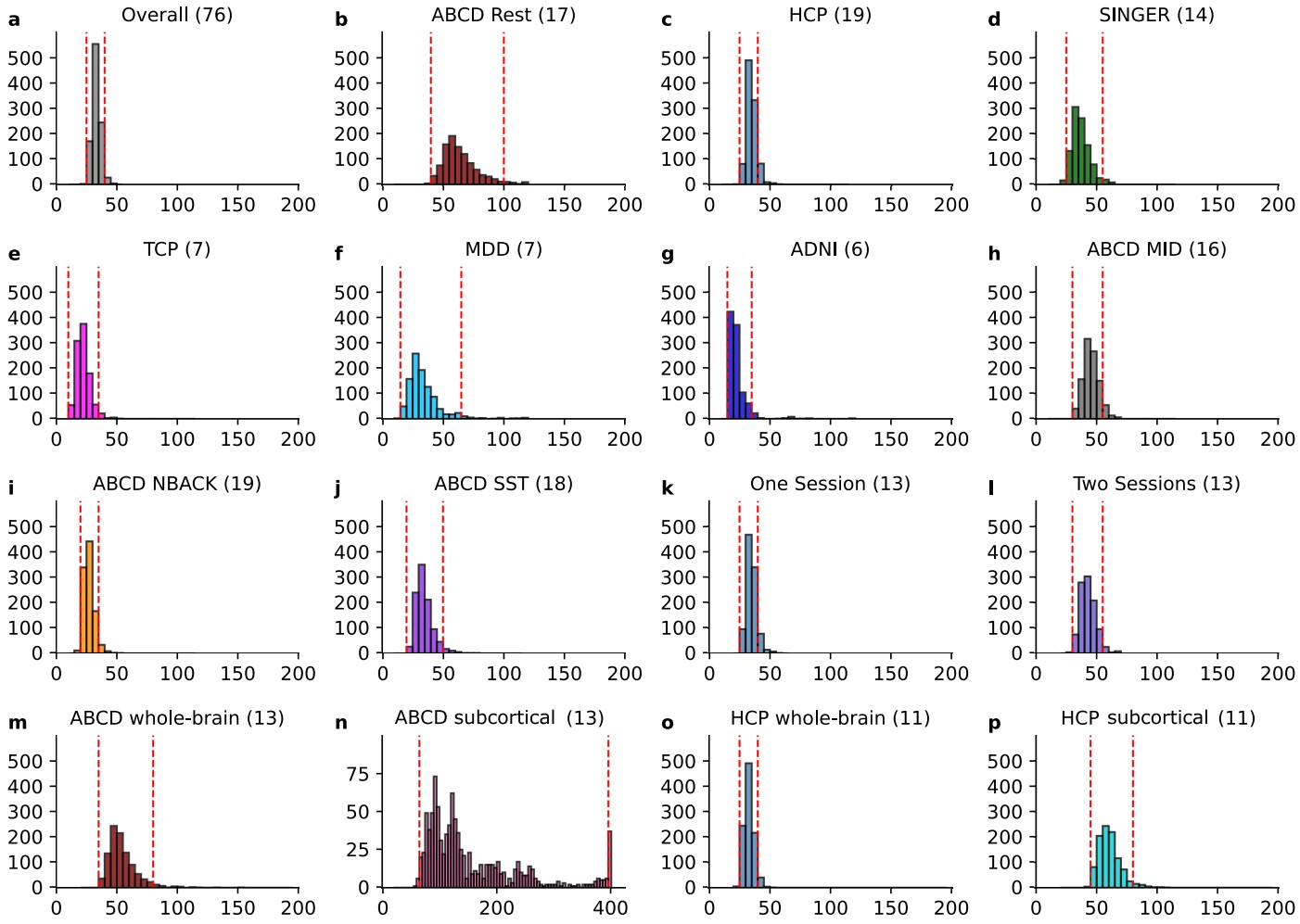

**Extended Data Fig. 7 | Bootstrapped histograms of most cost-effective scan times. a.** Bootstrapped histogram of the most cost-effective scan time averaged across nine datasets. This panel supports the results in Fig. 4c. **b.** Bootstrapped histogram of the most cost-effective scan time for the ABCD resting-fMRI dataset. **c.** Bootstrapped histogram of the most cost-effective scan time for the HCP dataset. **d.** Bootstrapped histogram of the most cost-effective scan time for the SINGER dataset. **e.** Bootstrapped histogram of the most cost-effective scan time for the TCP dataset. **f.** Bootstrapped histogram of the most cost-effective scan time for the MDD dataset. **g.** Bootstrapped histogram of the most cost-effective scan time for the ADNI dataset. **h.** Bootstrapped histogram of the most cost-effective scan time for the ABCD MID task-fMRI dataset. **i.** Bootstrapped histogram of the most cost-effective scan time for the ABCD n-back task-fMRI dataset. **j.** Bootstrapped histogram of the most cost-effective

scan time for the ABCD SST task-fMRI dataset. **k.** Bootstrapped histogram of the most cost-effective scan time for the HCP one-session analysis. **l.** Bootstrapped histogram of the most cost-effective scan time for the HCP two-session analysis. **b** to **l** support the results in Fig. 5. **m.** Bootstrapped histogram of the most cost-effective scan time for the ABCD whole-brain FC analysis. **n.** Bootstrapped histogram of the most cost-effective scan time for the ABCD subcortical-to-whole-brain FC analysis. **o.** Bootstrapped histogram of the most cost-effective scan time for the HCP whole-brain FC analysis. **p.** Bootstrapped histogram of the most cost-effective scan time for the HCP subcortical-to-whole-brain FC analysis. **m** to **p** support the results in Fig. 6a. Each panel contains 1000 bootstrapped samples. The 95% confidence intervals are shown by the red dashed lines. Numbers in brackets indicate the number of phenotypes. Details of bootstrap are found in Supplementary Table 1.

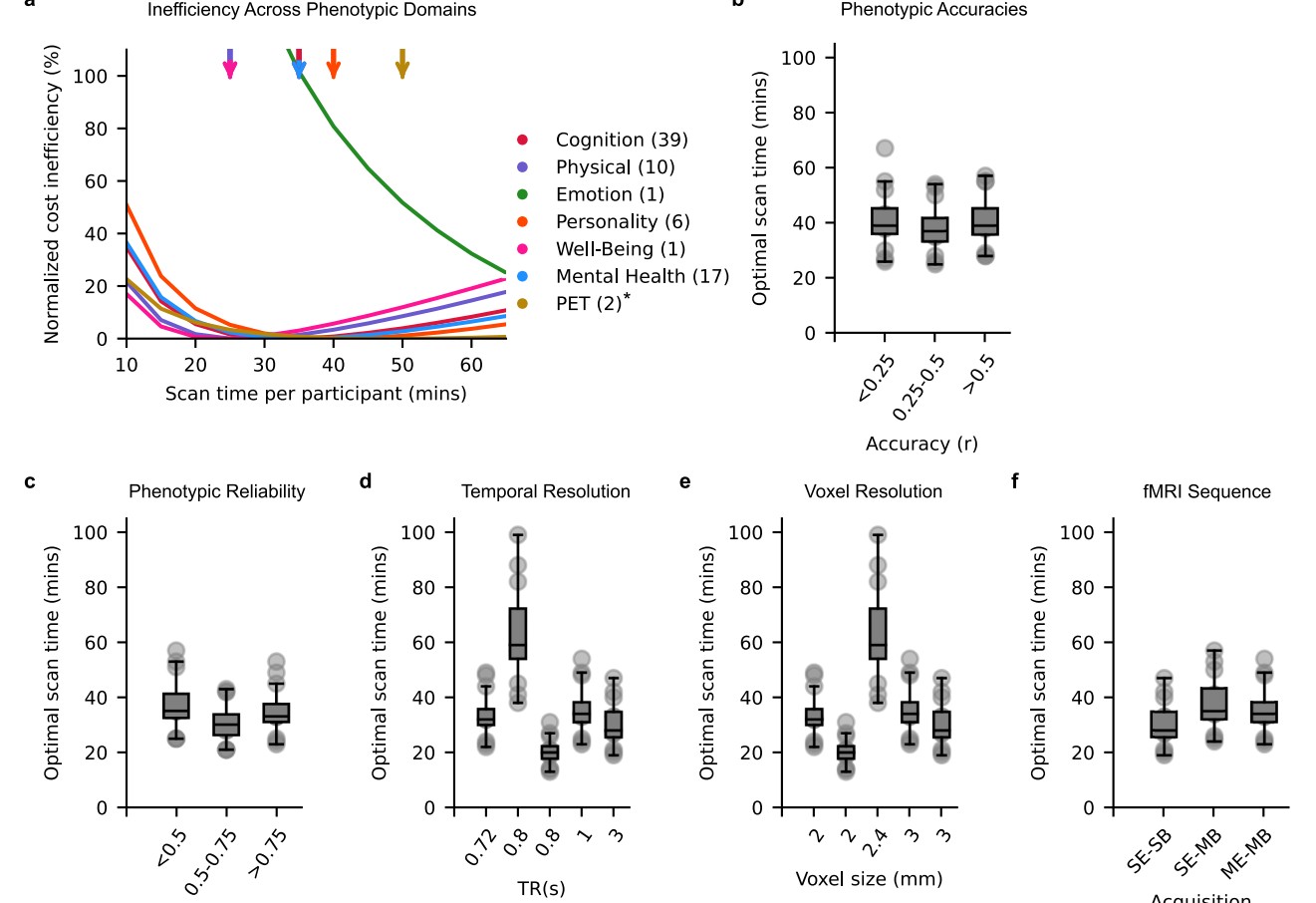

**Extended Data Fig. 8 | Variation across phenotypes & scan parameters.**
**a**. Cost inefficiency as a function of scan time for various phenotypic domains across nine resting-fMRI and task-fMRI datasets. Only for this plot, the positron emission tomography (PET) curve used a more realistic overhead cost of US$5,000 or US$10,000 per participant instead of US$500 or US$1,000 used for other phenotypes. Arrows indicate most cost-effective scan times. Numbers in brackets indicate number of phenotypes. For visualization, curves are normalized by subtracting the cost inefficiency of the best possible fixed scan time (of each curve), so the best possible fixed scan time is centred at zero. **b**. Optimal scan time as a function of phenotypic prediction accuracies. We sorted the maximum prediction accuracies (based on resting-state FC) of 19 HCP and 17 ABCD phenotypes into three bins. **c**. Optimal scan time as a function of phenotypic test-retest reliability. This analysis was obtained by considering

41 HCP participants, where the same phenotypic measures were collected twice (several months apart), allowing us to estimate phenotypic test-retest reliability. **d**. Optimal scan time as a function of repetition time (TR). **e**. Optimal scan time as a function of voxel size. **f**. Optimal scan time as a function of MRI acquisition. SE-SB: single-echo single-band; SE-MB: single-echo multi-band; ME-MB: multi-echo multi-band. **b** to **f** only considered resting-state FC. The ADNI dataset was excluded from **d** to **f** because it included both single-band and multi-band data with different TRs and voxel sizes. Each boxplot in **b** to **f** contains 12 data points corresponding to the 12 conditions we considered. Horizontal lines indicate the medians, boxes represent the interquartile range (IQR, from the 25th to 75th percentile), and whiskers extend to the most extreme data points not considered outliers (within 1.5× IQR).

**Extended Data Table 1 | The sample size and the maximum amount of scan time of each dataset**

| Dataset | Sample Size (N) | Scan Time (T) |
|---|---|---|
| HCP | 792 | 57m 36s |
| ABCD-rest | 2565 | 20m |
| SINGER | 642 | 9m 56s |
| TCP | 194 | 26m 2s |
| MDD | 287 | 23m 12s |
| ADNI | 586 | 9m |
| ABCD-MID | 2262 | 10m 44s |
| ABCD-NBACK | 2262 | 9m 39s |
| ABCD-SST | 2262 | 12m 39s |

**Extended Data Table 2 | Optimal scan time to maximize prediction accuracy given different overhead costs per participant, scan costs per hour and total fMRI budgets**

| Overhead cost per participant | Scan cost per hour | Total fMRI budget | | | | | | |
|---|---|---|---|---|---|---|---|---|
| | | 100K | 250K | 1M | 2.5M | 10M | 25M | 100M |
| 500 | 500 | 40.0 | 40.0 | 34.5 | 32.5 | 31.5 | 30.5 | 30.5 |
| | 1000 | 30.0 | 25.5 | 24.5 | 22.5 | 22.0 | 21.5 | 21.5 |
| | 1500 | 20.0 | 20.0 | 20.0 | 18.5 | 18.0 | 17.5 | 17.5 |
| | 2000 | 19.5 | 18.0 | 16.0 | 16.0 | 15.5 | 15.5 | 15.0 |
| | 2500 | 18.0 | 15.5 | 15.0 | 14.5 | 14.0 | 13.5 | 13.5 |
| 1000 | 500 | 80.0 | 67.4 | 56.4 | 50.5 | 44.5 | 44.0 | 43.0 |
| | 1000 | 40.0 | 40.0 | 38.5 | 35.0 | 31.5 | 31.0 | 30.5 |
| | 1500 | 40.0 | 33.0 | 30.5 | 28.5 | 25.5 | 25.0 | 25.0 |
| | 2000 | 30.0 | 28.0 | 25.5 | 24.0 | 22.5 | 22.0 | 21.5 |
| | 2500 | 24.0 | 26.0 | 24.0 | 21.5 | 20.0 | 19.5 | 19.0 |
| 2000 | 500 | 159.8 | 100.9 | 86.9 | 76.4 | 66.4 | 62.4 | 60.9 |
| | 1000 | 79.9 | 67.4 | 56.4 | 50.5 | 45.5 | 44.5 | 43.0 |
| | 1500 | 52.9 | 62.4 | 49.0 | 40.5 | 37.5 | 36.5 | 35.5 |
| | 2000 | 40.0 | 47.0 | 40.0 | 37.5 | 32.5 | 31.5 | 30.5 |
| | 2500 | 37.5 | 37.5 | 35.0 | 32.0 | 29.0 | 28.0 | 27.5 |
| 5000 | 500 | 322.7 | 233.3 | 159.3 | 133.4 | 110.9 | 102.9 | 96.9 |
| | 1000 | 299.7 | 199.8 | 128.4 | 94.4 | 79.9 | 74.4 | 69.9 |
| | 1500 | 199.8 | 132.9 | 85.4 | 78.4 | 63.4 | 60.0 | 55.9 |
| | 2000 | 149.9 | 99.9 | 80.4 | 65.4 | 55.3 | 51.4 | 49.0 |
| | 2500 | 119.9 | 79.9 | 64.4 | 56.4 | 50.5 | 45.5 | 44.0 |
| 10000 | 500 | 500.0 | 466.5 | 299.7 | 228.3 | 180.8 | 151.3 | 140.9 |
| | 1000 | 399.6 | 282.2 | 189.3 | 149.9 | 112.9 | 110.9 | 100.9 |
| | 1500 | 266.2 | 187.8 | 171.3 | 125.9 | 99.9 | 90.4 | 81.9 |
| | 2000 | 199.8 | 168.3 | 128.4 | 116.4 | 84.4 | 77.4 | 69.9 |
| | 2500 | 159.8 | 134.9 | 112.9 | 94.9 | 80.4 | 69.4 | 63.4 |

This table expands Extended Data Fig. 6 for a wider range of fMRI budgets, scan costs per hour and overhead costs per participant. Entries in the table show the optimal scan time in minutes.

# Reporting Summary

## Statistics

For all statistical analyses, confirm that the following items are present in the figure legend, table legend, main text, or Methods section.

| n/a | Confirmed | |
|---|---|---|
| ☐ | ☒ | The exact sample size (*n*) for each experimental group/condition, given as a discrete number and unit of measurement |
| ☐ | ☒ | A statement on whether measurements were taken from distinct samples or whether the same sample was measured repeatedly |
| ☐ | ☒ | The statistical test(s) used AND whether they are one- or two-sided <br> *Only common tests should be described solely by name; describe more complex techniques in the Methods section.* |
| ☐ | ☒ | A description of all covariates tested |
| ☐ | ☒ | A description of any assumptions or corrections, such as tests of normality and adjustment for multiple comparisons |
| ☐ | ☒ | A full description of the statistical parameters including central tendency (e.g. means) or other basic estimates (e.g. regression coefficient) AND variation (e.g. standard deviation) or associated estimates of uncertainty (e.g. confidence intervals) |
| ☐ | ☒ | For null hypothesis testing, the test statistic (e.g. $F$, $t$, $r$) with confidence intervals, effect sizes, degrees of freedom and $P$ value noted <br> *Give P values as exact values whenever suitable.* |
| ☒ | ☐ | For Bayesian analysis, information on the choice of priors and Markov chain Monte Carlo settings |
| ☒ | ☐ | For hierarchical and complex designs, identification of the appropriate level for tests and full reporting of outcomes |
| ☐ | ☒ | Estimates of effect sizes (e.g. Cohen's *d*, Pearson's *r*), indicating how they were calculated |

*Our web collection on statistics for biologists contains articles on many of the points above.*

## Software and code

Policy information about availability of computer code

| Data collection | No software was used for data collection. |
|---|---|
| Data analysis | FreeSurfer 5.3.0; FSL 5.0.8; MATLAB (2018b); Python 3.7. Code for this study is publicly available in the GitHub repository maintained by the Computational Brain Imaging Group (https://github.com/ThomasYeoLab/CBIG). Processing pipelines of the fMRI data can be found here (https://github.com/ThomasYeoLab/CBIG/tree/master/stable_projects/preprocessing/CBIG_fMRI_Preproc2016). Code specific to the analyses in this study can be found here (https://github.com/ThomasYeoLab/CBIG/tree/master/stable_projects/predict_phenotypes/Ooi2024_ME). |

For manuscripts utilizing custom algorithms or software that are central to the research but not yet described in published literature, software must be made available to editors and reviewers. We strongly encourage code deposition in a community repository (e.g. GitHub). See the Nature Portfolio guidelines for submitting code & software for further information.

## Data

Policy information about availability of data

All manuscripts must include a data availability statement. This statement should provide the following information, where applicable:

- Accession codes, unique identifiers, or web links for publicly available datasets
- A description of any restrictions on data availability
- For clinical datasets or third party data, please ensure that the statement adheres to our policy

The raw data for HCP (https://www.humanconnectome.org/), ABCD (https://abcdstudy.org/), TCP (https://openneuro.org/datasets/ds005237 and https://

# Research involving human participants, their data, or biological material

Policy information about studies with [human participants or human data](). See also policy information about [sex, gender (identity/presentation), and sexual orientation]() and [race, ethnicity and racism]().

| | |
|---|---|
| Reporting on sex and gender | HCP sex distribution: 371 Male / 421 Female<br>ABCD sex distribution: 1251 Male / 1314 Female<br>SINGER sex distribution: 309 Male / 333 Female<br>TCP sex distribution: 81 Male / 110 Female / 3 Self declared<br>MDD sex distribution: 101 Male / 186 Female<br>ADNI sex distribution: 278 Male / 308 Female |
| Reporting on race, ethnicity, or other socially relevant groupings | Information related to race and ethnicity was not used in this study. |
| Population characteristics | HCP: 792 young adult (ages 22-35) participants were recruited from families with twins and non-twin siblings.<br>ABCD: 2565 children (ages 9-10)<br>SINGER: 642 adults aged 60-80 at risk of cognitive impairment and dementia<br>TCP: 194 adults (ages 18-70) meeting diagnostic criteria for a broad range of psychiatric illnesses and a healthy comparison group<br>MDD: 287 participants who meet the diagnostic criteria of DSM-5(Diagnostic and Statistical Manual of Mental Disorders, Fifth Edition) for depression disorder without psychotic symptoms, and currently experiencing a recurrence episode.<br>ADNI: 586 participants aged 55-90 years, consisting of cognitively normal individuals, those with mild cognitive impairment (MCI), and Alzheimer's disease patients, undergoing extensive neuroimaging and cognitive assessments. |
| Recruitment | Recruitment was carried out by the respective studies. |
| Ethics oversight | The HCP data collection was approved by a consortium of institutional review boards (IRBs) in the United States and Europe, led by Washington University in St Louis and the University of Minnesota (WU-Minn HCP Consortium).<br><br>Most ABCD research sites relied on a central IRB at the University of California, San Diego for the ethical review and approval of the research protocol, with a few sites obtaining local IRB approval.<br><br>The SINGER study has been approved by the National Healthcare Group Domain-Specific Review Board and is registered under ClinicalTrials.gov (ID: NCT05007353) with written informed consent obtained from all participants before enrolment into the study.<br><br>Participants from the TCP study were provided written informed consent following guidelines established by the Yale University and McLean Hospital (Partners Healthcare) IRBs.<br><br>The MDD dataset was collected from multiple rTMS clinical trials, and all data were obtained at the pretreatment stage. These trials include ChiCTR2300067671 (approved by the Institutional Review Boards of Beijing Anding Hospital, Henan Provincial People's Hospital, and Tianjin Medical University General Hospital); NCT05842278, NCT05842291, and NCT06166082 (all approved by the IRB of Beijing HuiLongGuan Hospital); and NCT06095778 (approved by the IRB of the Affiliated Brain Hospital of Guangzhou Medical University).<br><br>The ADNI study was approved by the IRBs of all participating institutions with informed written consent from all participants at each site. |

Note that full information on the approval of the study protocol must also be provided in the manuscript.

# Field-specific reporting

Please select the one below that is the best fit for your research. If you are not sure, read the appropriate sections before making your selection.

☐ Life sciences      ☒ Behavioural & social sciences      ☐ Ecological, evolutionary & environmental sciences

For a reference copy of the document with all sections, see [nature.com/documents/nr-reporting-summary-flat.pdf]()

# Behavioural & social sciences study design

All studies must disclose on these points even when the disclosure is negative.

| | |
|---|---|
| Study description | Quantitative cross-sectional study where we train and test predictive models to predict behavioral outcomes from neuroimaging data |
| Research sample | To evaluate the robustness of our theoretical model, we considered a diverse collection of datasets (HCP, ABCD, SINGER, TCP, MDD and ADNI) that span multiple fMRI sequences (single-echo single-band, single-echo multi-band, multi-echo multi-band), coordinate systems (fsLR, fsaverage, MNI152), racial groups (Western and Asian populations), mental disorders (healthy, neurological and psychiatric) and age groups (children, young adults and elderly). |
| Sampling strategy | For each dataset, we exclude participants who meet the exclusion criteria (See 'Data exclusions' section below). As a result, the current sample reflects the maximum available sample size for this study. |
| Data collection | All neuroimaging data were collected through various MRI scanners available at different scanning sites. Behavioral measures were collected based on procedures that were specific to each behavioral test. |
| Timing | HCP: The Human Connectome Project S1200 release started in 2012 and finished collecting data in 2016. <br> ABCD: The data collection for this public dataset is still on-going (however, our lab is not involved in the collection in any form). We first accessed the data in 2018. <br> SINGER: The data collection for this dataset is still on-going (however, our lab is not involved in the collection in any form). We first accessed the data in 2023. <br> TCP: The data collection for this dataset began in 2018 and ended in 2024. <br> MDD: The data collection for this dataset is still on-going (however, our lab is not involved in the collection in any form). We first accessed the data in 2024. <br> ADNI: The data collection for this dataset is still on-going (however, our lab is not involved in the collection in any form). We first accessed the data in 2023. |
| Data exclusions | HCP: We excluded 161 participants who did not have at least 40 minutes of uncensored data or did not have the full set of the 59 non-brain-imaging phenotypes from an initial sample of 953 participants. <br> ABCD: We excluded 2695 participants who did not have at least 15 minutes of uncensored data or did not have the full set of the 37 non-brain-imaging phenotypes from an initial sample of 5260 participants. <br> SINGER: We excluded 117 participants who did not have at least 10 minutes of uncensored data or did not have the full set of the 19 non-brain-imaging phenotypes from an initial sample of 759 participants. <br> TCP: We excluded 47 participants who did not have at least 26 minutes of uncensored data or did not have the full set of the 19 non-brain-imaging phenotypes from an initial sample of 241 participants. <br> MDD: We excluded 19 participants who did not have at least 23 minutes of uncensored data or did not have the full set of the 20 non-brain-imaging phenotypes from an initial sample of 306 participants. <br> ADNI: We excluded 182 participants who did not have at least 9 minutes of uncensored data or did not have the full set of the 6 non-brain-imaging phenotypes from an initial sample of 768 participants. <br><br> More details can be found in the Methods section of the paper (Datasets, phenotypes & participants). |
| Non-participation | We are not aware of how many participants may have dropped out as we are not involved in the data collection process. No data was collected for the current study. |
| Randomization | We did not allocate participants into different experimental groups because they all underwent the same experimental conditions. |

# Reporting for specific materials, systems and methods

We require information from authors about some types of materials, experimental systems and methods used in many studies. Here, indicate whether each material, system or method listed is relevant to your study. If you are not sure if a list item applies to your research, read the appropriate section before selecting a response.

## Materials & experimental systems

| n/a | Involved in the study |
|---|---|
| ☒ | Antibodies |
| ☒ | Eukaryotic cell lines |
| ☒ | Palaeontology and archaeology |
| ☒ | Animals and other organisms |
| ☒ | Clinical data |
| ☒ | Dual use research of concern |
| ☒ | Plants |

## Methods

| n/a | Involved in the study |
|---|---|
| ☒ | ChIP-seq |
| ☒ | Flow cytometry |
| ☐ | ☒ MRI-based neuroimaging |

# Plants

| | |
|---|---|
| Seed stocks | n/a |
| Novel plant genotypes | n/a |
| Authentication | n/a |

# Magnetic resonance imaging

## Experimental design

| | |
|---|---|
| Design type | Resting-state functional imaging |
| Design specifications | HCP: Each participant underwent 4 runs of resting-state functional imaging with a duration of 14.4 minutes.<br>ABCD: Each participant underwent 4 runs of resting-state functional imaging with a duration of 5 minutes.<br>SINGER: Each participant underwent 1 run of resting-state functional imaging with a duration of 10 minutes.<br>TCP: Each participant underwent 4 runs of resting-state functional imaging with a duration of 6.5 minutes.<br>MDD: Each participant underwent 4 runs of resting-state functional imaging with a duration of 6 minutes.<br>ADNI: Each participant underwent 1 run of resting-state functional imaging with a duration of 9 minutes. |
| Behavioral performance measures | No behavioral performance measures are collected during resting-state functional imaging. |

## Acquisition

| | |
|---|---|
| Imaging type(s) | Functional |
| Field strength | 3T |
| Sequence & imaging parameters | HCP: All participants were scanned on a customized Siemens 3T Skyra using a multi-band sequence. Each fMRI run was acquired with a repetition time (TR) of 0.72s at 2mm isotropic resolution and lasted for 14.4 min.<br>ABCD: Multiple scanners of different makes were used, so the sequences were variable, but each fMRI scan was in 2.4 mm isotropic resolution with a TR of 800 ms.<br>SINGER: The following scanning parameters were used:  TR=1000, TE=12/29.75/47.5, voxel size=3×3×3mm.<br>TCP: The following scanning parameters were used: TR = 800 milliseconds, TE = 37 milliseconds, flip angle = 52°, and voxel size =2mm. A multi-band acceleration factor of 8 was applied. An auto-align pulse sequence protocol was used to align the acquisition slices of the functional scans parallel to the anterior commissure-posterior commissure (AC-PC) plane of the MPRAGE and centered on the brain.<br>MDD: Rs-fMRI was acquired with an echo planar imaging (EPI) pulse sequence (TR=3000ms, TE=30ms, flip angle=90°, FOV=240×240, 80×80 matrix, 50 slices, voxel size=3×3×3mm).<br>ADNI: Multiple scanners of different makes were used, so the sequences were variable |
| Area of acquisition | Whole-brain coverage |
| Diffusion MRI | ☐ Used   ☒ Not used |

## Preprocessing

| | |
|---|---|
| Preprocessing software | FreeSurfer 5.3.0; FSL 5.0.8 |
| Normalization | Non-linear volumetric and surface projection |
| Normalization template | HCP: fsLR surface<br>ABCD/SINGER/MDD/ADNI: FreeSurfer fsaverage6 surface<br>TCP: MNI152 |
| Noise and artifact removal | HCP: Denoising was done by ICA-FIX, we additionally regressed out the global signal<br>ABCD: We regressed out the global signal, six motion correction parameters, averaged ventricular signal, averaged white matter signal, and their temporal derivatives (18 regressors in total)<br>SINGER:We regressed out the global signal, six motion correction parameters, averaged ventricular signal, averaged white matter signal, and their temporal derivatives (18 regressors in total)<br>TCP: Denoising was done by ICA-FIX, we additionally regressed out the global signal<br>MDD: We regressed out the global signal, six motion correction parameters, averaged ventricular signal, averaged white matter signal, and their temporal derivatives (18 regressors in total) |

| Volume censoring | ADNI:We regressed out the global signal, six motion correction parameters, averaged ventricular signal, averaged white matter signal, and their temporal derivatives (18 regressors in total) |
|---|---|
| | HCP: Motion outlier frames (FD > 0.2mm, DVARS > 75), along with one volume before and two volumes after, were marked as outliers and subsequently censored.<br>ABCD: Motion outlier frames ( FD > 0.3 mm, DVARS > 50), along with one volume before and two volumes after, were marked as outliers and subsequently censored.<br>SINGER:Motion outlier frames ( FD > 0.3 mm, DVARS > 60), along with one volume before and two volumes after, were marked as outliers and subsequently censored.<br>TCP: No censoring<br>MDD: No censoring<br>ADNI: No censoring |

## Statistical modeling & inference

| Model type and settings | Predictive |
|---|---|
| Effect(s) tested | Prediction of behavioral outcomes from functional connectivity derived from resting-state fMRI. |

Specify type of analysis: ☒ Whole brain ☐ ROI-based ☐ Both

| Statistic type for inference<br><br>(See Eklund et al. 2016) | Cluster-wise statistics are not applicable for our study as our analyses only utilize functional connectivity measures. |
|---|---|

| Correction | Multiple comparisons were corrected using the Benjamini–Yekutieli false discovery rate (FDR) procedure with q < 0.05. |
|---|---|

## Models & analysis

n/a | Involved in the study
☐ ☒ Functional and/or effective connectivity
☒ ☐ Graph analysis
☐ ☒ Multivariate modeling or predictive analysis

| Functional and/or effective connectivity | Pearson's correlation |
|---|---|

| Multivariate modeling and predictive analysis | We used kernel ridge regression and linear ridge regression. Model performance was evaluated with Pearson's correlation and coefficient of determination. |
|---|---|

