## [Peer Review file · Nature]

Longer scans boost prediction and cut costs in brain-wide association studies

Corresponding Author: Dr B.T. Thomas Yeo

Version 0:

Reviewer comments:

Referee #1

(Remarks to the Author)

This manuscript investigates the relative benefits of increasing sample size and scan duration in the prediction of behavioral phenotypes from functional connectivity MRI data. The authors used data from two large public repositories, the ABCD and HCP, to create different versions of 10-fold cross validation that systematically varied the number of participants used in each training fold, and the amount of resting-state fMRI data used from each participant. They find that increases in both scan time and participant number improve prediction levels, with the two factors initially trading off fairly equally, although eventually the benefits of scan duration plateau. The authors develop a statistical model to describe this relationship. They then use this model to create a calculator for the optimal balance between sample size and scan time for different budgets; for common budgets, arguing for larger scan times per participant than have been collected in many studies, but also the need for large sample sizes.

This manuscript adds a practical contribution to the ongoing debate on brain-wide association studies in human neuroimaging, importantly arguing for increased attention to not only sample sizes in these studies, but also the quantity of within subject-data. The manuscript was well written, the authors use techniques that they have previously validated and used extensively, and the general prediction/cross-validation approach is logical and thoughtfully carried out. However, I have some questions regarding the variability in results across datasets/measures, and the extent to which these specific prescriptions are likely to carry forward to other experimental scenarios that dampen my enthusiasm for this work as a “general purpose” solution for fMRI. I detail these issues below.

Major concerns

- My biggest concern with the current manuscript is that, while it thoroughly investigates the question of sample size and scan quantity, it only does so within a relatively constrained set of parameters – it uses two common public datasets (ABCD and HCP) with similar scan parameters (that both use e.g., higher multiband factors, faster temporal resolution, smaller voxel sizes, and single echo data that will limit SNR), it only investigates these questions at a single resolution of fMRI data (large regions), and it only investigates the consequences of participant number and data quantity on a single form of fMRI analysis (BWAS using multivariate prediction). My guess would be that SNR considerations, as well as the resolution of the data, are likely to be major factors that modify the benefit weighting between participant number and scan duration (i.e., presumably examining ever more detailed aspects of the cortex should benefit more from scan duration, rather than participant number, as participants are likely to differ more on these fine scale details; SNR is likely to improve data quality more quickly, and may lead to less of a dependence on scan duration). Thus, it's unclear to what extent the findings from these studies are likely to extend to other, new research designs in fMRI, which as I see it is the putative goal of this paper being. It seems best suited to addressing the potential power in already collected studies, but these can't be adjusted. The authors might address these limitations to some degree by more extensively characterizing the dependence on spatial scale, fMRI SNR, and alternative brain metrics, but the issues of data collection protocols are likely not addressable given current available consortia datasets.
- Relatedly, the current findings are likely driven by cortical functional connectivity results. Many BWAS, especially in the clinical domain, are fundamentally interested in subcortical interactions, which will typically exhibit substantially lower SNR, especially in high MB datasets. I think it is important for the authors to consider whether their proposed recommendations would differ depending on the brain areas under consideration.
- I thought it was great that the authors conducted their analyses in parallel both in the HCP and ABCD. However, it seemed to me that there were substantial differences in the dependency between sample size and scan length across the two

datasets (e.g., Fig. 1B – these panels at face look similar, but use fairly different x and y axis scales across the two figures, making it unclear how truly similar the results are). Given the very specific recommendations that are being made, it seems important to make clear (and resolve) the discrepancies across these two datasets in order to move forward in practice. I think it is also very important to keep the axis scales matched across different figures to prevent the false sense of similarity.

- Relatedly, while the authors propose a model that does a *reasonable* job of reproducing the data, it struck me that Fig. 6A looks fairly different from Fig. 1B, especially in terms of its dependency on scan length. Perhaps this is simply apparent because of differences in color scale and axis – it would help to put this figure on the same color scale/axis as the others to confirm. But if there is a discrepancy, does this point to the model doing a better job at capturing sample size than scanner length dependencies?
- Fig. 3: Why was there such large variation in cross-validation across runs? Does this point to some issues of potential participant selection/overfitting?
- Relatedly, I'm curious if the authors have explored the extent to which different steps in prediction are affected by sample size vs. data quantity. For example, the authors adopt a nested approach, optimizing model parameters for machine learning within each cross validation loop. My hunch is that this step may be relatively more affected by sample size (as the parameters can then be driven by a very small number of participants) ... is this the case? That is, if the optimization step is dropped, is a decreased dependency on participant number found?
- I was a bit confused about the emphasis/interpretation of the scan differences seen in this work. The authors appear to state in the text that the diminishing returns in scan duration beyond 20-30 minutes is due to state effects; however, this profile also matches well to the profile of reliability of cortical FC (Laumann et al., 2015, Neuron), and the later results on reliability seem (to me) to argue this is really the major driver of the observed relationship. On the other hand, the putative state level differences also match relatively well with the amount of data collected on a single day in the HCP (so the differences could be due to e.g., sleep/daily rhythms/etc. over longer time windows, rather than states). In general, if the scan level effects are truly due to more rapid "states", it seems like the most likely major state effect that could be impacting these scans systematically across people is arousal (presumably other states would differ across individuals). I would recommend the authors adjust their relative emphasis on the import of reliability vs. state level effects (or justify the claims about state level effects further). I also suggest that the authors consider this potential explanation of arousal or daily differences as driving state level effects, as well as the practical conundrum that experimenters can't typically chose to randomize their data collection protocols.
- While some measures show diminishing returns for scan duration after 20 minutes, others shown in the supplement continue to improve throughout the testable scan duration, even doubling. In general, there was a lot of variability in the relationships between scan duration and participant number across measures. Understanding what might cause this variation seems important if practical suggestions are likely to be made from this work. I would appreciate the authors digging into this issue in a bit more detail, and providing more nuance in their text to characterize this variation.
- Why does Fig. 3B show spatial orientation rather than the cognition factor score shown elsewhere? I think it would be more fair to carry the same measure forward, rather to swap to a new measure (out of all of the measures, this shows perhaps the least increase with increasing scan amount, so it doesn't feel representative of the full group)

Minor concerns

- Similar to above, can Fig. 2A results be shown on the same axes for better comparability? In general, I think it's important that equivalent measures in the ABCD and HCP be shown on the same axes. Similarly, can the reliability figures in S18 be shown on the same axes.
- There are measures that cannot be made without more extensive data per participant (and vice versa). That is, the current work only characterizes sample size/scan duration dependency in a very specific set of (albeit popular) analyses. I think it would be useful for the discussion to grapple with these more complex issues and to provide some context of how these results are relevant (or not) to these other scenarios.
- I was surprised that regular FD was used for the HCP data, given evidence (similar to in the ABCD) that a filtered version of FD might be more appropriate.
- In the derivations shown in the supplement, I didn't understand the S8 extension – could some more text be added here? What does the limitation of $t = 0.5$ indicate? What is t ?
- In association with Figure 4, the authors say that the model does poorly in some cases due to "intrinsic predictability" of measures. What does that mean?
- Fig. 5: are C and D for all measures, or just anger/aggression?

Referee #2

(Remarks to the Author)

This study investigates the impact of sample size and fMRI scan duration on phenotypic predictions using resting-state fMRI (rs-fMRI) data from two large datasets, ABCD and HCP. It reveals that while sample size and scan duration are largely interchangeable in enhancing prediction accuracy, the benefits of extended scan duration diminish beyond 30 minutes, making sample size the more critical factor. Additionally, the study introduces a theoretical model capable of estimating prediction accuracy based on these variables, accompanied by a web application designed to guide future research planning.

The research underscores the importance of considering scan duration in study design and power calculations, a point well-acknowledged in previous studies or discussions on reliability or replicability (Birn et al., 2013; Nee, 2019; Gordon et al.,

2017) but less so in the context of brain-wide association studies (BWAS).

However, the study could benefit from a more detailed exploration of why scan time versus sample size is important for phenotypic prediction. Although the authors concluded these factors are broadly interchangeable, this assertion may only hold superficially, as their beneficial effects likely stem from different mechanisms. For example, longer scan durations might enhance the reliability of brain measures, whereas larger sample sizes could improve model robustness by increasing data variability and thus reducing the risk of overfitting.

Their model, primarily descriptive, appears suited only to the specific context of rsfMRI-based BWAS prediction for cognitive phenotypes. While there is a growing body of literature that highlights the advantages of task-based data for phenotypic prediction (Zhao et al., 2023; Greene et al., 2018), the generalizability of the study's findings and models to task fMRI-based BWAS remains uncertain. Similarly, it is ambiguous whether the model is applicable to clinical data, although the authors imply that their results have clinical implications (e.g., for children with ADHD, as discussed in the Discussion section). A deeper mechanistic understanding of how scan time and sample size enhance phenotypic prediction could facilitate the application of these findings to broader contexts.

The analyses and interpretations of the state effects were intriguing; however, these remain speculative due to the lack of direct measures of states in the data. More direct evidence of state-related measures is required to substantiate these interpretations, thus necessitating caution.

However, their interpretation of state effects indeed raises an important point—even with increasing data for an individual, the benefits of such data could stem from multiple factors. For instance, longer scan times might enhance the reliability of measures, but they could also introduce greater within-individual variability (i.e., state-related variability) in the training data. In this context, "longer scan time" encompasses various meanings and can be implemented in several ways. For example, to target reliability, a single-session long scan (e.g., a 30-minute resting scan in one session) may be effective, while to maximize within-individual variability, the scan might be segmented into multiple short scan sessions (e.g., three 10-minute resting scans across different days). Alternatively, simple state manipulation procedures could be used to increase within-individual variability. Therefore, it is conceivable that extensive scan times may not be necessary if sufficient within-individual variability can be induced. This suggests that a broad assertion such as "many studies would benefit from longer scan time per participant" could be misleading without a more detailed and nuanced understanding of the specific advantages conferred by longer scans.

Their main findings derive from the averaged prediction accuracies across 10-fold cross-validation and 50 repetitions with random participant splits. While average is a key metric of prediction performance, variance represents another significant aspect. Figure 3A clearly shows the differential impacts of sample size and scan time on both the variance and the mean of prediction accuracy—specifically, a smaller sample size results in higher variance in test outcomes. Considering that clinical applications or independent testing of models in prospective studies typically do not involve multiple repetitions, this variance becomes an essential factor to consider.

I wonder if their theoretical model works better even after considering the increased number of free parameters.

The abstract says prediction accuracy "might benefit from much longer scan durations (>50 min)," but it appears somewhat contradictory to the observed diminishing returns on prediction accuracy when extending beyond 30 minutes. The study indicates that sample size and scan duration are interchangeable for up to 30 minutes in the HCP dataset. This discrepancy suggests a need for clarity or revision.

For the ABCD data, the researchers employed a leave-3-site-clusters-out nested cross-validation approach, i.e., 7 vs. 3 splits, whereas for the HCP data, they used a 10-fold nested cross-validation strategy, i.e., 9 vs. 1 splits. Clarification about this analysis decision would be helpful.

Birn, R. M., Molloy, E. K., Patriat, R., Parker, T., Meier, T. B., Kirk, G. R., Nair, V. A., Meyerand, M. E., & Prabhakaran, V. (2013). The effect of scan length on the reliability of resting-state fMRI connectivity estimates. **NeuroImage**, *83**, 550–558.

Nee, D. E. (2019). fMRI replicability depends upon sufficient individual-level data. **Communications Biology**, *2**, 130.

Gordon, E. M., Laumann, T. O., Gilmore, A. W., Newbold, D. J., Greene, D. J., Berg, J. J., Ortega, M., Hoyt-Drazen, C., Gratton, C., Sun, H., Hampton, J. M., Coalson, R. S., Nguyen, A. L., McDermott, K. B., Shimony, J. S., Snyder, A. Z., Schlaggar, B. L., Petersen, S. E., Nelson, S. M., & Dosenbach, N. U. F. (2017). Precision Functional Mapping of Individual Human Brains. **Neuron**, *95*(4)*, 791–807.e7.

Zhao, W., Makowski, C., Hagler, D. J., Garavan, H. P., Thompson, W. K., Greene, D. J., Jernigan, T. L., & Dale, A. M. (2023). Task fMRI paradigms may capture more behaviorally relevant information than resting-state functional connectivity. **NeuroImage**, *270**, 119946.

Greene, A. S., Gao, S., Scheinost, D., & Constable, R. T. (2018). Task-induced brain state manipulation improves prediction of individual traits. **Nature Communications**, *9*(1)*, 2807.

Referee #3

(Remarks to the Author)

In this study, Ooi et al. examined the issues of sample size and scan duration selection in brain-phenotype prediction modeling and association analysis. The topic is important and practical, and the conclusion will be very helpful for many studies in relevant fields. Most of the analyses are appropriate and sound. The manuscript is well written and very informative. However, I have some concerns that should be adequately addressed before it is considered for publication. I have described my points below and hope that the authors will find them helpful in improving this manuscript.

1. My main concerns are from the randomization analysis regarding brain “states”. Fig 5CD visualizes the results from the maximum scan time. I am sorry if I misunderstood something, but functional connectivity by correlation does not depend on the order of the time points. Therefore, it is unclear why the original and randomized results differ in the figure panels.

The authors mentioned “state” effects during the resting scan and used randomization to address this issue. However, brain state within the resting scan was never directly defined, examined, and demonstrated in this study. The authors did not provide any practical or theoretical evidence for different states within the resting state fMRI. I would like to ask for a clear definition of states and direct evidence supporting different brain states within the resting period.

Together, I found the randomization analysis and results needs clarification.

2. One important finding of this study is the fit of logarithmic and theoretical models explaining the relationship between prediction accuracy and total scan time. The manuscript argues that the theoretical models fit better. However, the authors did not perform a statistical analysis comparing the fit of two models. Although two measures are numerically very different, the authors would want to add statistical results.

3. One of the target phenotypes is the cognition factor. If the factor was defined and estimated by the HCP and ABCD samples used in this manuscript, this is data leakage that inflates the prediction accuracy. If the factor was defined by an independent set of small samples, I would like to ask how stable and reliable the factor estimation is.

4. The authors distinguish between BWA reliability and prediction accuracy. Predictive modeling requires external validation beyond internal CV for better demonstration. Although the authors replicate similar patterns of internal CV results using two datasets, HCP and ABCD, it does not necessarily mean the reliability and validity of the prediction results. Therefore, I would like to see how well the prediction models would work in external independent datasets.

5. The authors found that KRR and LRR lead to similar conclusions across phenotype domains. I would like to see a discussion of this finding. What does it tell us about the underlying brain-phenotype associations? Are the associations always linear? Or does it mean that we need a lot more data to uncover true brain-phenotype associations?

6. ABCD has different imaging sites. I wonder if the site effect, which would add more inter-individual variance, compensates for the “state effect”, if it exists, which was stronger in HCP.

Minor

-Fig 5 and 6 need more explanation about the graph; error bar, shaded area, etc.

- In the first line and paragraph, the manuscript starts by mentioning brain structure and function. But this study (scan duration) is only about functional data. It might be helpful to explicitly clarify this first.

- The authors reported that the censoring did not substantially change their conclusion. I would like to ask for discussion about this observation.

Version 1:

Reviewer comments:

Referee #1

(Remarks to the Author)

I appreciate the added work the authors have put into this revision. I'm still of the opinion that this work is interesting, and I found that the analyses on the new datasets are an important addition. However, I'm still unconvinced about the overall message and its impact, as explained further below:

- In general, I found the new dependence on 1% error bars to largely muddy the underlying issues at play in this work, leading to a lack of clarity on the findings. While I understand the desire to provide them, in practice, they lead to very large discrepancies in the “amount of data needed” (e.g., in Figure S24.1, anywhere from 10 to 120 minutes in the ADNI dataset). This is even worse in several other comparisons. If this is truly the best guess provided by the current approach, this severely weakens the impact of the current manuscript, as it provides little practical guidance on the optimal amount of data to collect for future researchers. Moreover, these large error bars allow the authors to ignore the differences that cause

certain measures/datasets to be acceptable with relatively less (10 minutes) vs. more (30 minutes) data per person. Understanding the differences leading to these variations in the “minimum acceptable data” per participant seems critical to extrapolating conclusions and next directions from the current work, and to making it more useful for the broader community.

- With this in mind, while the authors have done a lot for this response in terms of adding new datasets, many of the specific questions and suggestions I (and the other reviewers made) made were not very well addressed (e.g., the specific dependency on SNR, resolution, association with test-retest reliability, hyperparameter selection). While new datasets were added, it wasn't clear to me that there was a substantial investigation of how scan parameters affected the results, aside from the description that the error bars overlapped (and, given the size of the error bars as noted above, this seems like a not very useful conclusion). When differences were found, they were relatively skirted over (e.g., the differences between subcortical and cortical connectivity, which were never returned to in the discussion section).
- I was also surprised, given the focus of the article, that with the addition of many new datasets, very little attention was given to how much data was present per person in the new datasets. This seems like information that should be prominently included in the primary text figures. Were any able to sufficiently test the maximal scan time per participant to validate the findings from the model shown in Fig. 6? If no, this seems like an important limitation that should be noted in the paper, and weakens the conclusions that can be made.
- I still think it's notable that many of the measures continue to show more gains with more data per participant, for the full extent of data collected (Supp. Figs), especially for high prediction measures like Age. At the least, this point seems like one that merits additional discussion within the manuscript. Could it be that other limitations (e.g., the accuracy of behavioral metrics) limit the potential per participant gains seen with other measures?
- The x-axis scale chosen for these model-based scan figures makes it very difficult to identify the small amounts of scan time (<10 min.) used in many large consortia datasets (UK Biobank, ADNI). How then can researchers practically use the information from this paper to interpret past results? This seems like a missed opportunity to clarify the current literature. Ultimately, as I read through the manuscript, I asked myself: what is the goal of this manuscript? If the goal is to critique past consortia studies for not collecting sufficient per participant data, then this point is not made very clearly, and not shown in the figures. If it is to provide direction for future consortia studies, this guidance seems relatively weak, given the large error bars and lack of engagement with specific parameters likely to affect future BWAS. At best, it provides murky findings. At worst, I think it has the risk of promoting a take-away message that is not sufficiently nuanced (i.e., “participant number and data quantity trade off”), which can lead to the impression that these two quantities don't matter individually and have the same influence. But, as R2 pointed out (and is also embedded in the modeling framework), different mechanisms cause these parameters to influence BWAS, which will likely lead their outcome to differ based on the specific brain areas, resolution, and dataset characteristics of the project in question. While the model may be able to capture these differences, the authors do not do much to engage with them within the primary text or explain them to readers.

Referee #2

(Remarks to the Author)

The revision has successfully addressed all my comments. I appreciate their thoughtful and thorough responses. In particular, I commend the authors for their efforts in developing a detailed theoretical interpretation of a potential mechanism in R2C1, along with the corresponding revisions in the Results section. Additionally, the inclusion of further tests using task-based data, clinical data, and two-session resting-state data significantly strengthens the study. Overall, I believe the authors have done an excellent job with these revisions. Congratulations!

Referee #3

(Remarks to the Author)

In this revision, the authors have added a number of new analyses, including statistics and generalizability tests to task fMRI data, clinical data, different phenotypes, and different MR acquisition and processing protocols. The authors have provided substantial results, addressing the previous review comments from myself and other reviewers. This revision has greatly improved the manuscript, and I commend the authors for their responsiveness and extensive work.

I have one specific follow-up question regarding the task fMRI analyses. The revision discusses "non-stationarity" in rest fMRI data. I wonder whether the differences between task and rest fMRI data (Fig. S26) can be at least partially explained by differences in stationarity between fMRI types (i.e. mental/cognitive states, randomness, etc.), and if so, to what extent. I guess that the experimental designs (block vs. event and their similarity across participants) may influence such an observation. It would be helpful to further discuss the experimental choices (sample size and scan length) with differences between fMRI data types.

Version 2:

Reviewer comments:

Referee #1

(Remarks to the Author)

I have read the authors second revision on this manuscript and am very impressed. The authors have directly addressed my

core concerns with the previous work, and in doing so, I believe they have significantly strengthened the message of this work and the impact it will have on the field. I congratulate them for their careful and extensive work.

Referee #3

(Remarks to the Author)

It seems to me that the authors have adequately addressed my last comment and R1's concerns in this re-revised submission.

However, I'd like to raise an additional concern regarding the authors' third key message. They suggest that 30-minute scans are cost-effective even for task-related data, as demonstrated by their analysis. My concern is that this conclusion about cost-effectiveness for task fMRI is not directly supported by the empirical data, since all task fMRI datasets used in this study are shorter than 15 minutes (Table 1).

This limitation should be clearly discussed in the manuscript. I am sorry if the authors have already clarified this somewhere in the manuscript that I may have missed.

Response to reviewers 2023-06-10250A-Z “MRI economics: Balancing sample size and scan duration in brain wide association studies”

We thank the reviewers and editor for their close read of this manuscript and insightful comments. Several important suggestions were made for improvement, and we have considered each carefully and revised accordingly. Our responses are in blue, while reviewer and editor comments are in italics. For convenience, changes to the manuscript are quoted verbatim (normal font) when appropriate. We believe the manuscript is much improved and hope it is now suitable for publication.

Editor:

(EIC1) [...] while the referees find your work of potential interest, they have raised important concerns about the generalizability, that, in our view, need to be addressed before we can consider publication in Nature. More specifically, you will see that both R1 and R2 note the constrained set of parameters you use in your exploration of the trade off between sample size and scan quality. Without getting into the impact of factors that will likely require other datasets (e.g. differences in MRI data collection protocols), and ideally also a demonstration that this holds (or doesn't) for task and clinical data (see R2's comments), we are not persuaded that the current version represents a development of sufficiently broad and striking impact for further consideration in Nature.

We thank the editor for the careful consideration of our manuscript and helpful suggestions throughout the review process.

Before providing point-by-point responses to the reviewers, we will first summarize major changes in the manuscript. The original manuscript utilized HCP-YA and ABCD datasets. There were concerns that our reference for selecting sample size and scan time in future studies might not be broadly applicable. Therefore, we have added four more datasets: Transdiagnostic Connectome Project (TCP), Major Depressive Disorder (MDD), Alzheimer’s Disease Neuroimaging Initiative (ADNI) and the Singapore geriatric intervention study to reduce cognitive decline and physical frailty (SINGER) datasets.

In the revision, we demonstrate the following:

- Our study design reference generalizes beyond healthy participants to participants with psychiatric (MDD and TCP) and neurological (ADNI) disorders (Figure S24).
- Our study design reference generalizes beyond predominantly Western populations to Asian populations (MDD and SINGER) (Figure S24).
- Our study design reference generalizes beyond single-echo multi-band acquisition to multi-echo multi-band (SINGER) and single-echo single-band (MDD) (Figure S24).
- Our study design reference generalizes beyond children and young adults to older adults in the SINGER and ADNI datasets (Figure S24).
- Our study design reference generalizes beyond our original parcellation (419 parcels) to a higher resolution parcellation (1019 parcels) (Figures S28 and S29).
- We show that acquiring resting-state fMRI in two separate sessions yields similar study design reference to our original analyses (Figure S29).

- Our study design reference generalizes beyond resting-state FC to task-state FC in the ABCD dataset (Figure S26).
- Our study design reference generalizes across phenotypic domains (Figure S25).
- Our study design reference generalizes beyond whole-brain FC to cortical-subcortical FC (Figures S28 and S29).

Another important change we have made in the manuscript is to allow for increased flexibility in study design. More specifically, instead of providing the exact sample size and scan time to achieve maximum prediction accuracy, we now provide a range of scan time over which studies can achieve within 1% of the maximum prediction accuracy (updated Figure 6B). Observe that all curves rise steeply with increasing scan time per participant, and then declines slowly with increasing scan time. Therefore, the 1% optimal scan time range (error bars in Figure 6B) is asymmetric with a long tail towards longer scan time.

With the revised Figure 6B, we modified our results to focus on the *minimum* scan time necessary to achieve within 1% of the maximum prediction accuracy (left side of error bars in Figure 6B) instead of the optimum scan time (solid circles in Figure 6B). The manuscript has been updated as follows.

pg 3 (Abstract)

When accounting for fixed overhead costs associated with each participant (e.g., recruitment, non-imaging measures), prediction accuracy in many small-scale and some large-scale BWAS might benefit from longer scan time than typically assumed. These results generalize across phenotypic domains, scanners, acquisition protocols, racial groups, mental disorders, age groups, as well as resting-state and task-state functional connectivity.

pg 4 (Introduction)

Here, we systematically characterize the effects of sample size and scan time of resting-state fMRI on BWAS prediction accuracy, using the Adolescent Brain and Cognitive Development (ABCD) study and the Human Connectome Project (HCP). To explore how overhead cost per participant impacts the trade-off between sample size and scan time in maximizing prediction accuracy within a fixed scan budget, we then expanded the datasets to include the Transdiagnostic Connectome Project (TCP), Major Depressive Disorder (MDD), Alzheimer's Disease Neuroimaging Initiative (ADNI) and the Singapore geriatric intervention study to reduce cognitive decline and physical frailty (SINGER) datasets. Overall, our results provide an empirical reference for future study design.

pg 15 (Results)

A reference for future study design

We have shown that investigators have the flexibility of attaining a specified prediction accuracy through different combinations of sample size and scan time per participant. To derive a reference for future studies, we considered four additional datasets – the TCP, MDD, ADNI and SINGER datasets. 34 phenotypes exhibited good fit to the theoretical

model (Table S2; Figures S17 to S20). We also considered task-FC of the three ABCD tasks, and found that the number of phenotypes with good fit to the theoretical model ranged from 16 to 19 (Table S2; Figures S21 to S23).

In total, we fitted the theoretical model to 76 phenotypes in the HCP, ABCD, TCP, MDD, ADNI and SINGER datasets, yielding 89% average explained variance (Table S2). For each phenotype, the model was normalized by its maximum achievable accuracy (estimated by the theoretical model), yielding a fraction of maximum achievable prediction accuracy for every combination of sample size and scan time per participant. The fraction of maximum achievable prediction accuracy was then averaged across the phenotypes under a hypothetical 10-fold cross-validation scenario (Figure 6A). We note that the correlations between Figure 6A and Figure 1B across corresponding sample sizes and scan durations were 0.99 for both HCP and ABCD datasets.

For the purpose of study design, we also need to consider the fundamental asymmetry between sample size and scan time per participant because of inherent overhead cost associated with each participant. These overhead costs might include recruitment effort, manpower to perform neuropsychological tests, additional MRI modalities (e.g., anatomical T1, diffusion MRI), other biomarkers (e.g., position emission tomography or blood tests). Therefore, the overhead cost can be higher than the cost of fMRI itself. Figure 6B illustrates the prediction accuracy achievable with different total fMRI budgets, costs per hour of scan time and overhead cost per participant.

There are three observations. First, larger fMRI budgets, lower scan cost and lower overhead cost facilitates larger sample size and scan time, leading to greater achievable prediction accuracy. Second, optimal scan time (solid circles in Figure 6B) increases with larger overhead cost, lower fMRI budget and lower scan cost. Third, all curves rise steeply with increasing scan time per participant, and then declines slowly with increasing scan time. Therefore, the optimal scan time range (error bars in Figure 6B) needed to achieve within 1% of the maximum prediction accuracy is asymmetric with a longer tail towards longer scan time. This long tail becomes longer when overhead cost is high. The reason is that even though sample size is more important than scan time in the theoretical model, higher overhead cost makes sample size and scan time more interchangeable.

(A)

(B)

Figure 6. Empirical reference for balancing sample size and scan duration while accounting for fixed costs per participant to optimally design BWAS. (A) Fraction of maximum achievable prediction accuracy as a function of sample size and scan time per participant. Here, we assume a hypothetical 10-fold cross-validation scenario, where 90% of the sample size is used for training a model. The theoretical model was fitted to 77 phenotypes from the HCP, ABCD (rest and task), SINGER, TCP, MDD and ADNI datasets, yielding 89% average explained variance (Table S2). For each phenotype, the model was normalized by its maximum achievable accuracy (based on the theoretical model), yielding a fraction of maximum achievable prediction accuracy for every combination of sample size and scan time per participant. The fraction of maximum achievable prediction accuracy was then averaged across the phenotypes yielding the plot. (B) Fraction of maximum achievable prediction accuracy as a function of total fMRI budget, scan cost per hour and overhead cost per participant. The solid circles indicate location of the maximum prediction accuracy. Circles are not shown if optimal combination of sample size and scan time was beyond the edge of the graph (i.e., more than 200 minutes of scan time). The error bars indicate the optimal range of scan time that can achieve within 1% of the maximum prediction accuracy.

As an example, when total fMRI budget was \$100K, scan cost was \$500 per hour and overhead cost was \$500 for each participant, the optimal prediction accuracy was achieved by scanning 119 participants for 40.8 min per participant. To achieve within 1% of the maximum prediction accuracy, the *minimum* scan time was 25.5 min, which jumped to 67.5 min when overhead cost was \$2000 per participant. As another example, when total fMRI budget was \$10M and scan cost was \$500 per hour, to achieve within 1% of the maximum prediction accuracy, the *minimum* scan time per participant were 8 min, 21.9 min and 41.6 min, corresponding to overhead costs of \$500, \$2000 and \$5000 respectively.

Among the six datasets, the optimal scan time was the highest for ABCD and lowest for ADNI, but overall, the 1% optimum scan time ranges were highly overlapping across the six datasets (Figure S24). The consistency across datasets was remarkable given variations across datasets (Table S5), including different phenotypes, acquisitions (single band, multiband, multiecho multiband), coordinate systems (fsLR, fsaverage, MNI152), racial groups (Western and Asian populations), mental disorders (healthy, neurological and psychiatric) and age groups (children, young adults and elderly).

Different phenotypic domains also yielded highly overlapping 1% optimum scan time ranges (Figure S25). Furthermore, replicating previous studies (Greene et al., 2018; Chen et al., 2022; Zhao et al., 2023), n-back task-FC yielded better prediction performance for cognitive measures compared with RSFC in the ABCD dataset. However, although the optimal scan time was slightly higher for resting-state and the SST-task, the 1% optimum

scan time ranges were highly overlapping across resting-state and the three tasks (Figure S26).

These results (Figure 6) do not account for second-order effects. For example, certain populations (e.g., children) might not be able to handle more than 1 hour of MRI scanning at a time, so longer scans would need to be broken up into multiple sessions, yielding an overhead cost associated with each session, and so on. As another example, beyond a certain sample size, multi-site data collection becomes necessary, which increases overhead cost per participant. Our web application (WEB_APPLICATION_LINK) allows for more flexible usage.

pg 18 (Results)

Additional control analyses

Figure S27 shows the 1% optimal scan time ranges for 13 HCP and ABCD phenotypes that exhibited strong agreement with the theoretical model without serious over-shoot or under-shoot. The 1% optimal scan time ranges were similar to the main analyses. Similar results were also obtained with all 36 HCP and ABCD phenotypes after randomizing the run order, as well as a subset of 17 HCP and ABCD phenotypes that exhibited strong agreement with the theoretical model without serious over-shoot or under-shoot after randomizing run order (Figure S27).

A higher resolution cortical parcellation with 1000 parcels yielded highly overlapping optimal scan time ranges with the original analysis (Figures S28 & S29). Given the observed non-stationarity effects (Figure 5), we simulated splitting data collection into two separate sessions. This simulation is possible because the HCP collected resting-state fMRI on two different days (sessions). We found splitting data collection into two sessions slightly increase optimal scan time, but the 1% optimal scan time ranges still highly overlapped when collecting data in a single session versus two sessions (Figure S29).

Our main analyses involved a cortical parcellation with 400 regions and 19 subcortical regions, yielding 419×419 RSFC matrices. We also repeated the main analyses using 19×419 RSFC matrices (Figures S28 & S29). When overhead cost per participant was low (e.g., \$500 or \$1000), the 1% optimal scan time range was highly overlapping between subcortical-cortical FC and whole-brain FC. However, when overhead cost per participant was high (e.g., \$5000), the optimal scan time for subcortical-cortical FC was somewhat longer than for whole-brain FC, although there was still substantial overlap in the 1% optimal scan time ranges. This might arise because of lower SNR in subcortical regions, resulting in the need for longer scan time to achieve better estimate of subcortical FC.

pg 19 (Discussion)

Neuroimaging studies are always confronted with the difficult decision of how to allocate fixed resources for an optimal study design. Here, we systematically investigate the trade-off between maximising scan time and sample size in the context of predicting phenotypes from FC. Sample size and scan time per participant are broadly interchangeable up to 20-30 min of scan time, and can be explained with a logarithmic model. A more complex theoretical model is able to explain prediction accuracy for longer scan time. Model fits were excellent across multiple phenotypic domains in six datasets, suggesting strong generalizability of these findings.

pg 26 (Methods)

We also utilized resting-state fMRI from the SINGER baseline cohort. We filtered participants from an initial set of 759 participants, excluding participants who did not have at least 10 minutes of resting-fMRI data or did not have the full set of the 19 phenotypes that we investigated (Table S4). This resulted in a final set of 642 participants with demographics found in Table S5.

We utilized resting-state fMRI from the TCP dataset (Chopra et al., 2024). We filtered participants from an initial set of 241 participants, excluding participants who did not have at least 26 minutes of resting-fMRI data or did not have the full set of the 19 phenotypes that we investigated (Table S4). This resulted in a final set of 194 participants with demographics found in Table S5.

We utilized resting-state fMRI from the MDD dataset. We filtered participants from an initial set of 306 participants. We excluded participants who did not have at least 23 minutes of resting-fMRI data or did not have the full set of the 20 phenotypes that we investigated (Table S4). This resulted in a final set of 287 participants with demographics found in Table S5.

We utilized resting-state fMRI from the ADNI datasets (ADNI 2, ADNI 3 and ADNI GO). We filtered participants from an initial set of 768 participants with both fMRI and PET scans acquired within 1 year of each other. We excluded participants who did not have at least 9 minutes of resting-fMRI data or did not have the full set of the 6 phenotypes that we investigated (Table S4). This resulted in a final set of 586 participants with demographics found in Table S5.

In addition, we considered task-fMRI from the ABCD 2.0.1 release. We filtered participants from Chen's set of 5260 participants (J. Chen et al., 2023). We excluded participants who did not have all 3 task-fMRI data remaining after quality control, and did not have the full set of the 37 phenotypes that we investigated. This resulted in a final set of 2262 participants with demographics found in Table S5.

pg 27 (Methods)

For the SINGER dataset, we processed the functional data with the following steps. (1) Removal of the first 4 frames; (2) Slice time correction; (3) Motion correction and outlier detection: frames with $FD > 0.3\text{mm}$ or $DVARS > 60$ were flagged as censored frames. 1 frame before and 2 frames after these volumes were flagged as censored frames. Uncensored segments of data lasting fewer than five contiguous frames were also labeled as censored frames. Runs with over half of the frames censored were removed; (4) Correcting for susceptibility-induced spatial distortion; (5) Multi-echo denoising (DuPre et al., 2021); (6) Alignment with structural image using boundary-based registration (Greve et al., 2009); (7) Global, white matter and ventricular signals, 6 motion parameters, and their temporal derivatives were regressed from the functional data. Regression coefficients were estimated from uncensored data.; (8) Censored frames were interpolated with the Lomb-Scargle periodogram (Power et al., 2014). (9) The data underwent bandpass filtering (0.009Hz – 0.08Hz). (10) Lastly, the data was then projected onto FreeSurfer fsaverage6 surface space and smoothed using a 6 mm full-width half maximum kernel.

For the TCP dataset, the details of data processing can be found elsewhere (Chopra et al., 2024). Briefly, the functional data was processed by following the HCP minimal processing pipeline with ICA-FIX, followed by GSR. The processed data was then projected onto MNI space.

For the MDD dataset, we processed the functional data with the following steps. (1) Slice time correction; (2) Motion correction, (3) Normalization for global mean signal intensity; (4) Alignment with structural image using boundary-based registration (Greve et al., 2009); (5) Linear detrending and bandpass filtering (0.01-0.08 Hz), and (6) Global, white matter and ventricular signals, 6 motion parameters, and their temporal derivatives were regressed from the functional data. (7) Lastly, the data was then projected onto FreeSurfer fsaverage6 surface space and smoothed using a 6 mm full-width half maximum kernel.

For the ADNI dataset, we processed the functional data with the following steps. (1) Slice time correction; (2) Motion correction; (3) Alignment with structural image using boundary-based registration (Greve et al., 2009); (4) Global, white matter and ventricular signals, 6 motion parameters, and their temporal derivatives were regressed from the functional data. (5) Lastly, the data was then projected onto FreeSurfer fsaverage6 surface space and smoothed using a 6 mm full-width half maximum kernel.

We derived a 419×419 RSFC matrix for each HCP and ABCD participant using the first T minutes of scan time. The 419 regions consisted of 400 parcels from the Schaefer parcellation (Schaefer et al., 2018), and 19 subcortical regions of interest (Fischl et al., 2002). For the HCP, ABCD and TCP datasets, T was varied from 2 to the maximum scan time in intervals of 2 minutes. This resulted in 29 RSFC matrices per participant in the HCP dataset (generated from using the minimum amount of 2 minutes to the maximum

amount of 58 minutes in intervals of 2 minutes), 10 RSFC matrices per participant in the ABCD dataset (generated from using the minimum amount of 2 minutes to the maximum amount of 20 minutes in intervals of 2 minutes), and 13 RSFC matrices per participant in the TCP dataset (generated from using the minimum amount of 2 minutes to the maximum amount of 26 minutes in intervals of 2 minutes).

In the case of the MDD dataset, the total scan time was an odd number (23 minutes), so T was varied from 3 to the maximum of 23 minutes in intervals of 2 minutes, which resulted in 11 RSFC matrices per participant. For SINGER, ADNI and ABCD task-fMRI data, because the scans were relatively short (around 10 minutes), T was varied from 2 minutes the maximum scan time in intervals of 1 minute. This resulted in 9 RSFC matrices per participant in the SINGER datasets (generated from using the minimum amount of 2 minutes to the maximum amount of 10 minutes), 8 RSFC matrices per participant in the ADNI datasets (generated from using the minimum amount of 2 minutes to the maximum amount of 9 minutes), 9 RSFC matrices per participant in the ABCD n-back task (from using the minimum amount of 2 minutes to the maximum amount of 9.65 minutes), 11 RSFC matrices per participant in the ABCD SST task (from using the minimum amount of 2 minutes to the maximum amount of 11.65 minutes) and 10 RSFC matrices per participant in the ABCD MID task (from using the minimum amount of 2 minutes to the maximum amount of 10.74 minutes).

pg 30 (Methods)

The procedure for the other datasets followed the same principle as the HCP dataset. However, the ABCD (rest and task) and ADNI datasets comprised participants from multiple sites. Therefore, following our previous studies (Chen et al., 2022; Ooi et al., 2022), we combined ABCD participants across the 22 imaging sites into 10 site-clusters and combined ADNI participants across the 71 imaging sites into 20 site-clusters (Table S6). Each site-cluster has at least 227, 156 and 29 participants in ABCD (rest), ABCD (task) and ADNI datasets respectively.

Instead of the 10-fold inner-loop (nested) cross-validation procedure in the HCP dataset, we performed a leave-3-site-clusters-out inner-loop (nested) cross-validation (i.e., 7 site-clusters are used for training and 3 site-clusters are used for testing) in the ABCD (rest and task) dataset. The hyperparameter was again selected using a 10-fold CV within the training set. This nested cross-validation procedure was performed for every possible split of the site clusters, resulting in 120 replications. The prediction accuracies were averaged across all 120 replications.

We did not perform a leave-one-site-cluster-out procedure because the site-clusters are “fixed”, so the cross-validation procedure can only be repeated 10 times under a leave-one-site-cluster-out scenario (instead of 120 times). Similarly, we did not go for leave-two-site-clusters-out procedure because that will only yield a maximum of 45 repetitions of cross-validation. On the other hand, if we left more than 3 site clusters out (e.g., leave-

5-site-clusters-out), we could achieve more cross-validation repetitions, but at the cost of reducing the maximum training set size. Therefore, we opted for the leave-3-site-clusters-out procedure, consistent with our previous study (Chen et al., 2022).

To be consistent with the ABCD dataset, for the ADNI dataset, we also performed a leave-3-site-clusters-out inner-loop (nested) cross-validation procedure. This procedure was performed for every possible split of the site clusters, resulting in 1140 replications. The prediction accuracies were averaged across all 1140 replications.

We also performed 10-fold inner-loop (nested) cross-validation procedure in the TCP, MDD and SINGER datasets. Although the data from the TCP and MDD datasets were acquired from multiple sites, the number of sites was much smaller (2 and 5 respectively) than that of the ABCD and ADNI datasets. Therefore, we were unable to use the leave-some-site-out cross-validation strategy because that would reduce the training set size by too much. Therefore, we ran a 10-fold nested cross-validation strategy (similar to the HCP). However, we regress sites from the target phenotype in the training set, which were then applied to the test set. In other words, our prediction was performed on the residuals of phenotypes after site regression. Site regression was unnecessary for the SINGER dataset as the data was only collected from a single site. The rest of the prediction workflow was the same as the HCP dataset, except for the number of repetitions. Since TCP, MDD and SINGER datasets had smaller sample size than the HCP dataset, the 10-fold cross-validation was repeated 350 times. The prediction accuracies were averaged across all test folds and all repetitions.

Similar to the HCP, the analyses were repeated with different numbers of training participants, ranging from 200 to 1600 ABCD (rest) participants (in intervals of 200). Together with the full training set size of approximately 1800 participants, there were 9 different training set sizes. The whole procedure was repeated with different values of T . Since there were 10 values of T in the ABCD (rest) dataset, there were in total 10×9 values of prediction accuracies for each phenotype. In the case of ABCD (task), the sample size was smaller with maximum training set size of approximately 1600 participants, so there were only 8 different training set sizes.

The ADNI and SINGER datasets had less participants than the HCP dataset, so we decided to sample the training set size more finely. More specifically, we repeated the analyses by varying the number of training participants from the minimum sample size of 100 to the maximum sample size in intervals of 100. For SINGER, the full training set size is ~580 participants, so there were 6 different training set sizes in total (100, 200, 300, 400, 500, ~580). For ADNI, the full training set size is ~530, so there were also 6 different training set sizes in total (100, 200, 300, 400, 500, ~530).

Finally, TCP and MDD datasets were the smallest, so the training set size was sampled even more finely. More specifically, we repeated the analyses by varying the number of

training participants from the minimum sample size of 50 to the maximum sample size in intervals of 25. For TCP, the full training set size is ~175, so there 6 training set sizes in total (50, 75, 100, 125, 150, 175). For MDD, the full training set size is ~258, so there 10 training set sizes in total (50, 75, 100, 125, 150, 175, 200, 225, 250, 258).

Current best MRI practices suggest that the model hyperparameter should be optimized (Nichols et al., 2017), so in the current study, we did not consider the case where the hyperparameter was fixed. As an aside, we note that for all analyses, the best hyperparameter was selected using a 10-fold cross-validation within the training set. The best hyperparameter was then used to train the model on the full training set. Therefore, the full training set was used for hyperparameter selection and for training the model. Furthermore, we only needed to select one hyperparameter, while training the model required fitting many more parameters. Therefore, we do not expect the hyperparameter selection to be more dependent on the training set size than training the actual model itself.

We also note that our study focused on out-of-sample prediction within the same dataset, but did not explore cross-dataset prediction (Wu et al., 2023). For predictive models to be clinically useful, these models must generalize to completely new datasets. The best way to achieve this goal is by training models from multiple datasets jointly, so as to maximize the diversity of the training data (Abraham et al., 2017; P. Chen et al., 2023). However, we did not consider cross-dataset prediction in the current study because most studies are not designed with the primary aim of combining the collected data with other datasets.

A full table of prediction accuracies for every combination of sample size and scan time per participant can be found in the supplementary spreadsheet.

Table S2. Summary of prediction accuracy analyses for all datasets. The “strict accuracy threshold” column shows the number of phenotypes whose prediction accuracies (Pearson’s r) were more than 0.1 when the full dataset was used (maximum N and T). The “Adherence to theoretical model” column shows the number of phenotypes which passed the strict accuracy threshold and showed a good fit to the theoretical models after visual assessment. “Average COD of model fit” refers to the goodness-of-fit of the theoretical model averaged over the phenotypes that showed adherence to the theoretical model. We note that many phenotypes overlap between ABCD (rest), ABCD (MID), ABCD (NBACK), ABCD (SST), so in the case of the ABCD, there were in total 23 unique phenotypes that showed adherence to the theoretical model, yielding $23 + 19 + 14 + 7 + 7 + 6 = 76$ unique phenotypes used for generating Figure 6.

Dataset	Total number of phenotypes	Strict accuracy threshold (max $r > 0.1$)	Adherence to theoretical model	Average COD of model fit
ABCD (rest)	37	23 (out of 37)	17 (out of 23)	0.894
HCP	59	28 (out of 59)	19 (out of 28)	0.888

SINGER	19	15 (out of 19)	14 (out of 15)	0.926
TCP	19	10 (out of 19)	7 (out of 10)	0.818
MDD	20	11 (out of 20)	7 (out of 11)	0.844
ADNI	6	6 (out of 6)	6 (out of 6)	0.920
ABCD (MID)	37	21 (out of 37)	16 (out of 21)	0.921
ABCD (NBACK)	37	22 (out of 37)	19 (out of 22)	0.884
ABCD (SST)	37	22 (out of 37)	18 (out of 22)	0.872
Control Analyses				
ABCD (subcortical)	37	18 (out of 37)	14 (out of 18)	0.868
HCP (subcortical)	59	21 (out of 59)	13 (out of 21)	0.860
ABCD (1000 parcels)	37	24 (out of 37)	18 (out of 24)	0.890
HCP (1000 parcels)	59	30 (out of 59)	24 (out of 30)	0.858
HCP (mix days)	59	21 (out of 59)	16 (out of 21)	0.847

Table S5.2 Diagnostic distributions for each dataset

	Diagnostic distributions
HCP	Healthy Controls
ABCD	Healthy Controls
SINGER	Elderly at risk for vascular cognitive impairment
TCP	Control (N = 76), MDD (N = 31), PTSD (N = 14), GAD (N = 12), Dysthymia (N = 9), Social Anxiety Disorder (N = 8), SUD (N = 8), BPD I (N = 6), BPD II (N = 6), Other Anxiety Disorder (N = 5), Other Mood Disorder (N = 4), Schizophrenia (N = 4), Schizoaffective Disorder (N = 4), ADHD (N = 3), Eating Disorder (N = 2), OCD (N = 2)
MDD	Major Depressive Disorder (N = 287)
ADNI	Control (N = 334), Mild Cognitive Impairment (N = 184), AD dementia (N = 68)

Table S5.3 Acquisition for each dataset

	Acquisition	Atlas Space
HCP	Single-echo Multi-band Custom Skyra	fsLR

ABCD	Single-echo Multi-band on GE & Siemens scanners (more information in Table S6.1)	fsaverage
SINGER	Multi-echo multi-band Prisma Fit	fsaverage
TCP	Single-echo multi-band on two Prisma scanners	MNI152
MDD	Single-echo single-band on five Prisma scanners	fsaverage
ADNI	Single-echo multi-band (N = 78) & single-echo single-band (N = 508) on GE, Philips & Siemens scanners (more information in Table S6.2)	fsaverage

Robustness across datasets (Part 1)

Figure S24.1. Fraction of maximum achievable prediction accuracy as a function of total fMRI budget, scan cost per hour, with a **fixed overhead cost per participant of \$500 across different datasets**. The solid circles indicate location of the maximum prediction accuracy. Circles are not shown if optimal combination of sample size and scan time was beyond the edge of the graph (i.e., more than 200 minutes of scan time). The error bars indicate the range of scan time that could achieve within 1% of the maximum prediction accuracy.

Robustness across datasets (Part 2)

Figure S24.2. Fraction of maximum achievable prediction accuracy as a function of total fMRI budget, scan cost per hour, with a **fixed overhead cost per participant of \$1000 across different datasets**. The solid circles indicate location of the maximum prediction accuracy. Circles are not shown if optimal combination of sample size and scan time was beyond the edge of the graph (i.e., more than 200 minutes of scan time). The error bars indicate the range of scan time that could achieve within 1% of the maximum prediction accuracy.

Robustness across datasets (Part 3)

Figure S24.3. Fraction of maximum achievable prediction accuracy as a function of total fMRI budget, scan cost per hour, with a **fixed overhead cost per participant of \$2000 across different datasets**. The solid circles indicate location of the maximum prediction accuracy. Circles are not shown if optimal combination of sample size and scan time was beyond the edge of the graph (i.e., more than 200 minutes of scan time). The error bars indicate the range of scan time that could achieve within 1% of the maximum prediction accuracy. We note that the last row corresponds to an edge scenario where the overhead cost is very high (\$2000), while budget is very low (\$100K), which means that a small change in sample size (e.g., 1 participant) will lead to a large change in optimal scan time.

Robustness across datasets (Part 4)

Figure S24.4. Fraction of maximum achievable prediction accuracy as a function of total fMRI budget, scan cost per hour, with a **fixed overhead cost per participant of \$5000 across different datasets**. The solid circles indicate location of the maximum prediction accuracy. Circles are not shown if optimal combination of sample size and scan time was beyond the edge of the graph (i.e., more than 200 minutes of scan time). The error bars indicate the range of scan time that could achieve within 1% of the maximum prediction accuracy. We note that the last row corresponds to an edge scenario where the overhead cost is very high (\$5000), while budget is very low (\$100K), which means that a small change in sample size (e.g., 1 participant) will lead to a large change in optimal scan time.

Robustness across phenotypic domains (Part 1)

Figure S25.1. Fraction of maximum achievable prediction accuracy across **different phenotypic domains with a fixed overhead cost per participant of \$500** in the HCP, ABCD, SINGER, ADNI, TCP and MDD datasets. The solid circles indicate location of the maximum prediction accuracy. Circles are not shown if optimal combination of sample size and scan time was beyond the edge of the graph (i.e., more than 200 minutes of scan time). The error bars indicate the range of scan time that could achieve within 1% of the maximum prediction accuracy. We note that all phenotypic domains exhibited highly overlapping 1% optimal scan time ranges, with the exception of the “emotion” domain. However, there is only one phenotype in that domain, so the deviation should not be over-interpreted.

Robustness across phenotypic domains (Part 2)

Figure S25.2. Fraction of maximum achievable prediction accuracy across **different phenotypic domains with a fixed overhead cost per participant of \$1000** in the HCP, ABCD, SINGER, ADNI, TCP and MDD datasets. The solid circles indicate location of the maximum prediction accuracy. Circles are not shown if optimal combination of sample size and scan time was beyond the edge of the graph (i.e., more than 200 minutes of scan time). The error bars indicate the range of scan time that could achieve within 1% of the maximum prediction accuracy. We note that all phenotypic domains exhibited highly overlapping 1% optimal scan time ranges, with the exception of the “emotion” domain. However, there is only one phenotype in that domain, so the deviation should not be over-interpreted.

Robustness across phenotypic domains (Part 3)

Figure S25.3. Fraction of maximum achievable prediction accuracy across **different phenotypic domains with a fixed overhead cost per participant of \$2000** in the HCP, ABCD, SINGER, ADNI, TCP and MDD datasets. The solid circles indicate location of the maximum prediction accuracy. Circles are not shown if optimal combination of sample size and scan time was beyond the edge of the graph (i.e., more than 200 minutes of scan time). The error bars indicate the range of scan time that could achieve within 1% of the maximum prediction accuracy. We note that all phenotypic domains exhibited highly overlapping 1% optimal scan time ranges, with the exception of the “emotion” domain. However, there is only one phenotype in that domain, so the deviation should not be over-interpreted. We note that the last row corresponds to an edge scenario where the overhead cost is very high (\$2000), while budget is very low (\$100K), which means that a small change in sample size (e.g., 1 participant) will lead to a large change in optimal scan time.

Robustness across phenotypic domains (Part 4)

Figure S25.4. Fraction of maximum achievable prediction accuracy across **different phenotypic domains with a fixed overhead cost per participant of \$5000** in the HCP, ABCD, SINGER, ADNI, TCP and MDD datasets. The solid circles indicate location of the maximum prediction accuracy. Circles are not shown if optimal combination of sample size and scan time was beyond the edge of the graph (i.e., more than 200 minutes of scan time). The error bars indicate the range of scan time that could achieve within 1% of the maximum prediction accuracy. We note that all phenotypic domains exhibited highly overlapping 1% optimal scan time ranges, with the exception of the “emotion” domain. However, there is only one phenotype in that domain, so the deviation should not be over-interpreted. We note that the last row corresponds to an edge scenario where the overhead cost is very high (\$5000), while budget is very low (\$100K), which means that a small change in sample size (e.g., 1 participant) will lead to a large change in optimal scan time.

Robustness across resting and task states (Part 1)

Figure S26.1. Fraction of maximum achievable prediction accuracy as a function of total fMRI budget, scan cost per hour, with a **fixed overhead cost per participant of \$500 across resting and task states** in the ABCD dataset. We note that the fraction of maximum accuracy was averaged across a common set of 13 phenotypes that adhere to the theoretical model across the different conditions (resting and task states) in this figure. The solid circles indicate location of the maximum prediction accuracy. Circles are not shown if optimal combination of sample size and scan time was beyond the edge of the graph (i.e., more than 200 minutes of scan time). The error bars indicate the range of scan time that could achieve within 1% of the maximum prediction accuracy.

Robustness across resting and task states (Part 2)

Figure S26.2. Fraction of maximum achievable prediction accuracy as a function of total fMRI budget, scan cost per hour, with a **fixed overhead cost per participant of \$1000 across resting and task states** in the ABCD dataset. We note that the fraction of maximum accuracy was averaged across a common set of 13 phenotypes that adhere to the theoretical model across the different conditions (resting and task states) in this figure. The solid circles indicate location of the maximum prediction accuracy. Circles are not shown if optimal combination of sample size and scan time was beyond the edge of the graph (i.e., more than 200 minutes of scan time). The error bars indicate the range of scan time that could achieve within 1% of the maximum prediction accuracy.

Robustness across resting and task states (Part 3)

Figure S26.3. Fraction of maximum achievable prediction accuracy as a function of total fMRI budget, scan cost per hour, with a **fixed overhead cost per participant of \$5000 across resting and task states** in the ABCD dataset. We note that the fraction of maximum accuracy was averaged across a common set of 13 phenotypes that adhere to the theoretical model across the different conditions (resting and task states) in this figure. The solid circles indicate location of the maximum prediction accuracy. Circles are not shown if optimal combination of sample size and scan time was beyond the edge of the graph (i.e., more than 200 minutes of scan time). The error bars indicate the range of scan time that could achieve within 1% of the maximum prediction accuracy. We note that the last row corresponds to an edge scenario where the overhead cost is very high (\$2000), while budget is very low (\$100K), which means that a small change in sample size (e.g., 1 participant) will lead to a large change in optimal scan time.

Robustness across resting and task states (Part 4)

Figure S26.4. Fraction of maximum achievable prediction accuracy as a function of total fMRI budget, scan cost per hour, with a **fixed overhead cost per participant of \$5000 across resting and task states** in the ABCD dataset. We note that the fraction of maximum accuracy was averaged across a common set of 13 phenotypes that adhere to the theoretical model across the different conditions (resting and task states) in this figure. The solid circles indicate location of the maximum prediction accuracy. Circles are not shown if optimal combination of sample size and scan time was beyond the edge of the graph (i.e., more than 200 minutes of scan time). The error bars indicate the range of scan time that could achieve within 1% of the maximum prediction accuracy. We note that the last row corresponds to an edge scenario where the overhead cost is very high (\$5000), while budget is very low (\$100K), which means that a small change in sample size (e.g., 1 participant) will lead to a large change in optimal scan time.

Robustness across original and randomized run orders, as well as phenotypic selection criteria (Part 1)

Figure S27.1. Fraction of maximum achievable prediction accuracy as a function of total fMRI budget, scan cost per hour, with a **fixed overhead cost per participant of \$500 under stricter and randomized conditions** in the HCP and ABCD datasets. Here we consider a subset of 13 HCP and ABCD phenotypes that exhibited strong agreement with the theoretical model without serious over-shoot or under-shoot, all 36 HCP and ABCD phenotypes after randomizing the run order and a subset of 17 HCP and ABCD phenotypes that exhibited strong agreement with the theoretical model without serious over-shoot or under-shoot after randomizing run order. The solid circles indicate location of the maximum prediction accuracy. Circles are not shown if optimal combination of sample size and scan time was beyond the edge of the graph (i.e., more than 200 minutes of scan time). The error bars indicate the range of scan time that could achieve within 1% of the maximum prediction accuracy.

Robustness across original and randomized run orders, as well as phenotypic selection criteria (Part 2)

Figure S27.2. Fraction of maximum achievable prediction accuracy as a function of total fMRI budget, scan cost per hour, with a **fixed overhead cost per participant of \$1000 under stricter and randomized conditions** in the HCP and ABCD datasets. Here we consider a subset of 13 HCP and ABCD phenotypes that exhibited strong agreement with the theoretical model without serious over-shoot or under-shoot, all 36 HCP and ABCD phenotypes after randomizing the run order and a subset of 17 HCP and ABCD phenotypes that exhibited strong agreement with the theoretical model without serious over-shoot or under-shoot after randomizing run order. The solid circles indicate location of the maximum prediction accuracy. Circles are not shown if optimal combination of sample size and scan time was beyond the edge of the graph (i.e., more than 200 minutes of scan time). The error bars indicate the range of scan time that could achieve within 1% of the maximum prediction accuracy.

Robustness across original and randomized run orders, as well as phenotypic selection criteria (Part 3)

Figure S27.3. Fraction of maximum achievable prediction accuracy as a function of total fMRI budget, scan cost per hour, with a **fixed overhead cost per participant of \$2000 under stricter and randomized conditions** in the HCP and ABCD datasets. Here we consider a subset of 13 HCP and ABCD phenotypes that exhibited strong agreement with the theoretical model without serious over-shoot or under-shoot, all 36 HCP and ABCD phenotypes after randomizing the run order and a subset of 17 HCP and ABCD phenotypes that exhibited strong agreement with the theoretical model without serious over-shoot or under-shoot after randomizing run order. The solid circles indicate location of the maximum prediction accuracy. Circles are not shown if optimal combination of sample size and scan time was beyond the edge of the graph (i.e., more than 200 minutes of scan time). The error bars indicate the range of scan time that could achieve within 1% of the maximum prediction accuracy. We note that the last row corresponds to an edge scenario where the overhead cost is very high (\$2000), while budget is very low (\$100K), which means that a small change in sample size (e.g., 1 participant) will lead to a large change in optimal scan time.

Robustness across original and randomized run orders, as well as phenotypic selection criteria (Part 4)

Figure S27.4. Fraction of maximum achievable prediction accuracy as a function of total fMRI budget, scan cost per hour, with a **fixed overhead cost per participant of \$5000 under stricter and randomized conditions** in the HCP and ABCD datasets. Here we consider a subset of 13 HCP and ABCD phenotypes that exhibited strong agreement with the theoretical model without serious over-shoot or under-shoot, all 36 HCP and ABCD phenotypes after randomizing the run order and a subset of 17 HCP and ABCD phenotypes that exhibited strong agreement with the theoretical model without serious over-shoot or under-shoot after randomizing run order. The solid circles indicate location of the maximum prediction accuracy. Circles are not shown if optimal combination of sample size and scan time was beyond the edge of the graph (i.e., more than 200 minutes of scan time). The error bars indicate the range of scan time that could achieve within 1% of the maximum prediction accuracy. We note that the last row corresponds to an edge scenario where the overhead cost is very high (\$5000), while budget is very low (\$100K), which means that a small change in sample size (e.g., 1 participant) will lead to a large change in optimal scan time.

Robustness across #ROIs and types of ROIs in ABCD (Part 1)

Figure S28.1. Fraction of maximum achievable prediction accuracy as a function of total fMRI budget, scan cost per hour, with a **fixed overhead cost per participant of \$500 with different ROIs** in the ABCD dataset. 400+19 parcels involve prediction based on 419 x 419 RSFC. 19 subcortical parcels involve prediction based on 19 x 419 RSFC. 1000+19 parcels involve prediction based on 1019 x 1019 RSFC. We note that the fraction of maximum accuracy was averaged across a common set of 13 phenotypes that adhere to the theoretical model across the different conditions (ROIs) in this figure. The solid circles indicate location of the maximum prediction accuracy. Circles are not shown if optimal combination of sample size and scan time was beyond the edge of the graph (i.e., more than 200 minutes of scan time). The error bars indicate the range of scan time that could achieve within 1% of the maximum prediction accuracy.

Robustness across #ROIs and types of ROIs in ABCD (Part 2)

Figure S28.2. Fraction of maximum achievable prediction accuracy as a function of total fMRI budget, scan cost per hour, with a **fixed overhead cost per participant of \$1000 with different ROIs** in the ABCD dataset. 400+19 parcels involve prediction based on 419 x 419 RSFC. 19 subcortical parcels involve prediction based on 19 x 419 RSFC. 1000+19 parcels involve prediction based on 1019 x 1019 RSFC. We note that the fraction of maximum accuracy was averaged across a common set of 13 phenotypes that adhere to the theoretical model across the different conditions (ROIs) in this figure. The solid circles indicate location of the maximum prediction accuracy. Circles are not shown if optimal combination of sample size and scan time was beyond the edge of the graph (i.e., more than 200 minutes of scan time). The error bars indicate the range of scan time that could achieve within 1% of the maximum prediction accuracy.

Robustness across #ROIs and types of ROIs in ABCD (Part 3)

Figure S28.3. Fraction of maximum achievable prediction accuracy as a function of total fMRI budget, scan cost per hour, with a **fixed overhead cost per participant of \$2000 with different ROIs** in the ABCD dataset. We note that the fraction of maximum accuracy was averaged across a common set of 13 phenotypes that adhere to the theoretical model across the different conditions (ROIs) in this figure. 400+19 parcels involve prediction based on 419 x 419 RSFC. 19 subcortical parcels involve prediction based on 19 x 419 RSFC. 1000+19 parcels involve prediction based on 1019 x 1019 RSFC. The solid circles indicate location of the maximum prediction accuracy. Circles are not shown if optimal combination of sample size and scan time was beyond the edge of the graph (i.e., more than 200 minutes of scan time). The error bars indicate the range of scan time that could achieve within 1% of the maximum prediction accuracy. We note that the last row corresponds to an edge scenario where the overhead cost is very high (\$2000), while budget is very low (\$100K), which means that a small change in sample size (e.g., 1 participant) will lead to a large change in optimal scan time.

Robustness across #ROIs and types of ROIs in ABCD (Part 4)

Figure S28.4. Fraction of maximum achievable prediction accuracy as a function of total fMRI budget, scan cost per hour, with a **fixed overhead cost per participant of \$5000 with different ROIs** in the ABCD dataset. 400+19 parcels involve prediction based on 419 x 419 RSFC. 19 subcortical parcels involve prediction based on 19 x 419 RSFC. 1000+19 parcels involve prediction based on 1019 x 1019 RSFC. We note that the fraction of maximum accuracy was averaged across a common set of 13 phenotypes that adhere to the theoretical model across the different conditions (ROIs) in this figure. The solid circles indicate location of the maximum prediction accuracy. Circles are not shown if optimal combination of sample size and scan time was beyond the edge of the graph (i.e., more than 200 minutes of scan time). The error bars indicate the range of scan time that could achieve within 1% of the maximum prediction accuracy. We note that the last row corresponds to an edge scenario where the overhead cost is very high (\$5000), while budget is very low (\$100K), which means that a small change in sample size (e.g., 1 participant) will lead to a large change in optimal scan time.

Robustness across #ROIs, types of ROIs and #sessions in HCP (Part 1)

Figure S29.1. Fraction of maximum achievable prediction accuracy as a function of total fMRI budget, scan cost per hour, with a **fixed overhead cost per participant of \$500 under different conditions in the HCP dataset.** Here the various conditions corresponded to different ROIs, as well as the “Two Sessions” condition. 400+19 parcels involve prediction based on 419 x 419 RSFC. 19 subcortical parcels involve prediction based on 19 x 419 RSFC. 1000+19 parcels involve prediction based on 1019 x 1019 RSFC. In the “Two Sessions” condition, we simulated a situation where data was collected on two separate days (two sessions). This is achieved by splitting each scan time T min into the first $T/2$ min of the first session and the first $T/2$ min of the second session. For example, if $T = 10$ min, a 419 x 419 FC was computed based on the first 5 min of the first session and the first 5 min of the second session. We note that the fraction of maximum accuracy was averaged across a common set of 10 phenotypes that adhere to the theoretical model across the different conditions in this figure. The solid circles indicate location of the maximum prediction accuracy. Circles are not shown if optimal combination of sample size and scan time was beyond the edge of the graph (i.e., more than 200 minutes of scan time). The error bars indicate the range of scan time that could achieve within 1% of the maximum prediction accuracy.

Robustness across #ROIs, types of ROIs and #sessions in HCP (Part 2)

Figure S29.2. Fraction of maximum achievable prediction accuracy as a function of total fMRI budget, scan cost per hour, with a **fixed overhead cost per participant of \$1000** under **different conditions in the HCP dataset**. Here the various conditions corresponded to different ROIs, as well as the “Two Sessions” condition. 400+19 parcels involve prediction based on 419 x 419 RSFC. 19 subcortical parcels involve prediction based on 19 x 419 RSFC. 1000+19 parcels involve prediction based on 1019 x 1019 RSFC. In the “Two Sessions” condition, we simulated a situation where data was collected on two separate days (two sessions). This is achieved by splitting each scan time T min into the first $T/2$ min of the first session and the first $T/2$ min of the second session. For example, if $T = 10$ min, a 419 x 419 FC was computed based on the first 5 min of the first session and the first 5 min of the second session. We note that the fraction of maximum accuracy was averaged across a common set of 10 phenotypes that adhere to the theoretical model across the different conditions in this figure. The solid circles indicate location of the maximum prediction accuracy. Circles are not shown if optimal combination of sample size and scan time was beyond the edge of the graph (i.e., more than 200 minutes of scan time). The error bars indicate the range of scan time that could achieve within 1% of the maximum prediction accuracy.

Robustness across #ROIs, types of ROIs and #sessions in HCP (Part 3)

Figure S29.3. Fraction of maximum achievable prediction accuracy as a function of total fMRI budget, scan cost per hour, with a **fixed overhead cost per participant of \$2000 under different conditions in the HCP dataset**. Here the various conditions corresponded to different ROIs, as well as the “Two Sessions” condition. 400+19 parcels involve prediction based on 419 x 419 RSFC. 19 subcortical parcels involve prediction based on 19 x 419 RSFC. 1000+19 parcels involve prediction based on 1019 x 1019 RSFC. In the “Two Sessions” condition, we simulated a situation where data was collected on two separate days (two sessions). This is achieved by splitting each scan time T min into the first $T/2$ min of the first session and the first $T/2$ min of the second session. For example, if $T = 10$ min, a 419 x 419 FC was computed based on the first 5 min of the first session and the first 5 min of the second session. We note that the fraction of maximum accuracy was averaged across a common set of 10 phenotypes that adhere to the theoretical model across the different conditions in this figure. The solid circles indicate location of the maximum prediction accuracy. Circles are not shown if optimal combination of sample size and scan time was beyond the edge of the graph (i.e., more than 200 minutes of scan time). The error bars indicate the range of scan time that could achieve within 1% of the maximum prediction accuracy. We note that the last row corresponds to an edge scenario where the overhead cost is very high (\$2000), while budget is very low (\$100K), which means that a small change in sample size (e.g., 1 participant) will lead to a large change in optimal scan time

Robustness across #ROIs, types of ROIs and #sessions in HCP (Part 4)

Figure S29.4. Fraction of maximum achievable prediction accuracy as a function of total fMRI budget, scan cost per hour, with a **fixed overhead cost per participant of \$5000 under different conditions in the HCP dataset**. Here the various conditions corresponded to different ROIs, as well as the “Two Sessions” condition. 400+19 parcels involve prediction based on 419 x 419 RSFC. 19 subcortical parcels involve prediction based on 19 x 419 RSFC. 1000+19 parcels involve prediction based on 1019 x 1019 RSFC. In the “Two Sessions” condition, we simulated a situation where data was collected on two separate days (two sessions). This is achieved by splitting each scan time T min into the first $T/2$ min of the first session and the first $T/2$ min of the second session. For example, if $T = 10$ min, a 419 x 419 FC was computed based on the first 5 min of the first session and the first 5 min of the second session. We note that the fraction of maximum accuracy was averaged across a common set of 10 phenotypes that adhere to the theoretical model across the different conditions in this figure. The solid circles indicate location of the maximum prediction accuracy. Circles are not shown if optimal combination of sample size and scan time was beyond the edge of the graph (i.e., more than 200 minutes of scan time). The error bars indicate the range of scan time that could achieve within 1% of the maximum prediction accuracy. We note that the last row corresponds to an edge scenario where the overhead cost is very high (\$5000), while budget is very low (\$100K), which means that a small change in sample size (e.g., 1 participant) will lead to a large change in optimal scan time.

Referee #1:

(RIC1) This manuscript investigates the relative benefits of increasing sample size and scan duration in the prediction of behavioral phenotypes from functional connectivity MRI data. The authors used data from two large public repositories, the ABCD and HCP, to create different versions of 10-fold cross validation that systematically varied the number of participants used in each training fold, and the amount of resting-state fMRI data used from each participant. They find that increases in both scan time and participant number improve prediction levels, with the two factors initially trading off fairly equally, although eventually the benefits of scan duration plateau. The authors develop a statistical model to describe this relationship. They then use this model to create a calculator for the optimal balance between sample size and scan time for different budgets; for common budgets, arguing for larger scan times per participant than have been collected in many studies, but also the need for large sample sizes.

This manuscript adds a practical contribution to the ongoing debate on brain-wide association studies in human neuroimaging, importantly arguing for increased attention to not only sample sizes in these studies, but also the quantity of within subject-data. The manuscript was well written, the authors use techniques that they have previously validated and used extensively, and the general prediction/cross-validation approach is logical and thoughtfully carried out. However, I have some questions regarding the variability in results across datasets/measures, and the extent to which these specific prescriptions are likely to carry forward to other experimental scenarios that dampen my enthusiasm for this work as a “general purpose” solution for fMRI. I detail these issues below.

We thank the reviewer for the positive comments. We hope our revised manuscript has addressed your concerns regarding “the variability in results across datasets/measures, and the extent to which these specific prescriptions are likely to carry forward to other experimental scenarios.” Please see our extensive additional analyses in our response to E1C1.

(RIC2) My biggest concern with the current manuscript is that, while it thoroughly investigates the question of sample size and scan quantity, it only does so within a relatively constrained set of parameters – it uses two common public datasets (ABCD and HCP) with similar scan parameters (that both use e.g., higher multiband factors, faster temporal resolution, smaller voxel sizes, and single echo data that will limit SNR), it only investigates these questions at a single resolution of fMRI data (large regions), and it only investigates the consequences of participant number and data quantity on a single form of fMRI analysis (BWAS using multivariate prediction). My guess would be that SNR considerations, as well as the resolution of the data, are likely to be major factors that modify the benefit weighting between participant number and scan duration (i.e., presumably examining ever more detailed aspects of the cortex should benefit more from scan duration, rather than participant number, as participants are likely to differ more on these fine scale details; SNR is likely to improve data quality more quickly, and may lead to less of a dependence on scan duration). Thus, it’s unclear to what extent the findings from these studies are likely to extend to other, new research designs in fMRI, which as I see it is the putative goal of this paper being. It seems best suited to addressing the

potential power in already collected studies, but these can't be adjusted. The authors might address these limitations to some degree by more extensively characterizing the dependence on spatial scale, fMRI SNR, and alternative brain metrics, but the issues of data collection protocols are likely not addressable given current available consortia datasets.

We thank the reviewer for this comment, which really pushes us to greatly expand on our original analysis.

With regards to your concern about limited scan parameters and SNR considerations, our analyses now include multi-echo multi-band (SINGER) and single-echo single-band (MDD) data. The ADNI dataset also contained a mix of participants with multi-band (13.3%) and single-band (86.7%) acquisitions. With regards to the reviewer's concerns about spatial scale and alternative metrics, we have added analyses involving a higher-resolution 1000-region cortical parcellation, task-state FC and cortical-subcortical FC. We also note that the additional datasets included participants with psychiatric (MDD and TCP) and neurological (ADNI) disorders. The datasets also spanned diverse demographic groups from Asia and North America.

Furthermore, in this revision, as explained in our response to the editor (E1C1), we introduced the concept of a 1% optimal scan time range to increase flexibility in study design. More specifically, instead of providing the exact sample size and scan time to achieve maximum prediction accuracy, we now provide a range of scan time over which studies can achieve within 1% of the maximum prediction accuracy (updated Figure 6B; see our response to E1C1). Across all conditions, we found that the 1% optimal ranges of scan time were highly overlapping. Perhaps the only exception was subcortical-cortical FC. When overhead cost per participant was low (e.g., \$500 or \$1000), the 1% optimal scan time range was highly overlapping between subcortical-cortical FC and whole-brain FC. However, when overhead cost per participant was very high (e.g., \$5000), the optimal scan time for subcortical-cortical FC was somewhat higher than whole-brain FC, although we note that there was still substantial overlap in the 1% optimal scan time ranges. This might arise because of lower SNR in subcortical regions, resulting in the need for longer scan time to achieve better estimate of subcortical FC.

Given the robustness of results across six datasets and many analyses, we believe that our findings will generalize to new research designs in fMRI. For updates to the manuscript, please see our response to E1C1.

(RIC3) Relatedly, the current findings are likely driven by cortical functional connectivity results. Many BWAS, especially in the clinical domain, are fundamentally interested in subcortical interactions, which will typically exhibit substantially lower SNR, especially in high MB datasets. I think it is important for the authors to consider whether their proposed recommendations would differ depending on the brain areas under consideration.

We thank the reviewer for this suggestion. We have added an analysis that uses subcortical-cortical FC to predict phenotypes in the HCP and ABCD datasets. Please see our response to R1C2.

(RIC4) I thought it was great that the authors conducted their analyses in parallel both in the HCP and ABCD. However, it seemed to me that there were substantial differences in the dependency between sample size and scan length across the two datasets (e.g., Fig. 1B – these panels at face look similar, but use fairly different x and y axis scales across the two figures, making it unclear how truly similar the results are). Given the very specific recommendations that are being made, it seems important to make clear (and resolve) the discrepancies across these two datasets in order to move forward in practice. I think it is also very important to keep the axis scales matched across different figures to prevent the false sense of similarity.

We thank the reviewer for the compliment. We could not figure out a way to show ABCD and HCP in Figure 1B on the same scale (without compromising the utility of the presentation). However, we have added Figure S6, showing strong correspondence between the two datasets ($r = 0.97$). It is important to note that this is not a foreordained result given that the cognitive factor scores are not necessarily comparable between ACBD and HCP since each dataset contains different sets of cognitive tasks. Furthermore, even for the same tasks, children and adults might adopt different strategies to complete the tasks.

We have updated the manuscript as follows.

pg 6 (Results)

Furthermore, although the cognitive factors were not necessarily comparable across datasets, there was strong agreement in prediction accuracies between the two datasets for the same sample size and scan time per participant ($r = 0.98$; Figure S6).

Figure S6. Scatter plot of the cognition factor prediction accuracy in the ABCD (x-axis) and HCP (y-axis) datasets. Each dot represents the prediction accuracy for each dataset with the same sample size and scan time per participant (extracted from Figure 1B). Although the cognitive factor score is not comparable across datasets, we observed a strong correlation between the two datasets ($r = 0.98$).

*(RIC5) Relatedly, while the authors propose a model that does a *reasonable* job of reproducing the data, it struck me that Fig. 6A looks fairly different from Fig. 1B, especially in terms of its dependency on scan length. Perhaps this is simply apparent because of differences in color scale and axis – it would help to put this figure on the same color scale/axis as the others to confirm. But if there is a discrepancy, does this point to the model doing a better job at capturing sample size than scanner length dependencies?*

The reviewer is correct that the discrepancy between the two figures largely stem from the different x-axis and y-axis ranges. Indeed, the correlations between corresponding values from Figures 1B and 6A were high: $r = 0.99$ (HCP) and $r = 0.99$ (ABCD).

However, we have opted to show the two figures with different color schemes because Figures 1B and 6A are conceptually different in the sense that Figure 1B shows the Pearson's correlation

between predicted cognitive factor score and the true cognitive factor score, while Figure 6A shows the theoretically calculated fraction of maximum accuracy averaged across many phenotypes. Furthermore, the sample size in Figure 1B corresponds to the size of the training set, while Figure 6A (in the revision) is showing the full sample size (assuming 90% of the full dataset is used for training a model).

(RIC6) Fig. 3: Why was there such large variation in cross-validation across runs? Does this point to some issues of potential participant selection/overfitting?

We thank the reviewer for raising this point. This is not an overfitting issue. Instead, the large variability across cross-validation folds is a well-known problem for small sample sizes (Varoquaux, 2018). In order to address this concern, we have now introduced the concept of an optimal range of scan time over which studies can achieve within 1% of the maximum prediction accuracy. Using an optimal range of sample size and scan durations provides greater flexibility than a single maximum. For example, in a small sample size scenario, researchers might consider boosting sample size within the 1% optimal scan time range to minimize variability. Conversely, since the optimal scan range is asymmetric with a longer tail towards longer scan durations, in other scenarios researchers may favor longer scan time.

We have updated the manuscript with the above considerations, while not being overly prescriptive, so as to allow greater flexibility for researchers.

pg 19 (Discussion)

The 1% optimal scan time range provides greater flexibility in modifying study designs based on population- and site-specific characteristics. For example, a researcher seeking to study patients with depression from a minority population (i.e., higher overhead cost per participant) might find it more economical to increase the scan time for each participant in order to achieve the maximum possible prediction accuracy. Indeed, because the 1% optimal scan time range has a longer tail towards longer scan time, erring towards longer scan time (at the expense of sample size) increases the chance that the resulting scan time falls within the 1% optimal scan time range for the particular experiment. On the other hand, when sample size is small (in the low hundreds), there is a lot variability across cross-validation folds (Figure 3A; Varoquaux, 2018). Therefore, a case can also be made to prioritize sample size over scan time (left tail of the optimal scan time range).

(RIC7) Relatedly, I'm curious if the authors have explored the extent to which different steps in prediction are affected by sample size vs. data quantity. For example, the authors adopt a nested approach, optimizing model parameters for machine learning within each cross validation loop. My hunch is that this step may be relatively more affected by sample size (as the parameters can then be driven by a very small number of participants) ... is this the case? That is, if the optimization step is dropped, is a decreased dependency on participant number found?

In our nested cross-validation, our inner-loop makes full use of the training set by doing a 10-fold CV on the entire training set. The best hyperparameter is then used to train on the entire

training set again. Therefore, we are making use of the full training set for hyperparameter selection and the full training set for training the model.

Furthermore, we are only fitting one hyperparameter, while training the model requires fitting many more parameters. Therefore, we do not think that the hyperparameter selection will be more dependent on the training set size than training the actual model itself.

Therefore, we opted not to run the experiment suggested by the reviewer, since it is not a practice that we would recommend to the readers. We have updated the manuscript as follows.

pg 31 (Methods)

Current best MRI practices suggest that the model hyperparameter should be optimized (Nichols et al., 2017), so in the current study, we did not consider the case where the hyperparameter was fixed. As an aside, we note that for all analyses, the best hyperparameter was selected using a 10-fold cross-validation within the training set. The best hyperparameter was then used to train the model on the full training set. Therefore, the full training set was used for hyperparameter selection and for training the model. Furthermore, we only needed to select one hyperparameter, while training the model required fitting many more parameters. Therefore, we do not expect the hyperparameter selection to be more dependent on the training set size than training the actual model itself.

(RIC8) I was a bit confused about the emphasis/interpretation of the scan differences seen in this work. The authors appear to state in the text that the diminishing returns in scan duration beyond 20-30 minutes is due to state effects; however, this profile also matches well to the profile of reliability of cortical FC (Laumann et al., 2015, Neuron), and the later results on reliability seem (to me) to argue this is really the major driver of the observed relationship. On the other hand, the putative state level differences also match relatively well with the amount of data collected on a single day in the HCP (so the differences could be due to e.g., sleep/daily rhythms/etc. over longer time windows, rather than states). In general, if the scan level effects are truly due to more rapid “states”, it seems like the most likely major state effect that could be impacting these scans systematically across people is arousal (presumably other states would differ across individuals). I would recommend the authors adjust their relative emphasis on the import of reliability vs. state level effects (or justify the claims about state level effects further). I also suggest that the authors consider this potential explanation of arousal or daily differences as driving state level effects, as well as the practical conundrum that experimenters can’t typically chose to randomize their data collection protocols.

We agree with the reviewer that the diminishing returns of scan time is likely due to the saturation in FC reliability, rather than due to the state effects. Furthermore, since we did not have direct brain state measurements to interpret the mechanisms driving the effects of randomizing run order, we have modified the result section to not reference brain states, but instead uses a more neutral language of “non-stationarity in fMRI-phenotype relationship”. The reference to “brain states” has been moved to the discussion.

pg 13 (Results)

This suggests that fMRI-phenotype relationships might be non-stationary for certain phenotypes, which violates an assumption in both theoretical and logarithmic models. To put this in more colloquial terms, the assumption is that the relationship between FC and a phenotypic measure is the same (i.e., stationary) regardless of whether FC was computed based on five minutes of fMRI from the beginning, middle or end of the MRI session. To test the non-stationarity hypothesis, we note that for both HCP and ABCD datasets, the fMRI data was collected over four runs. Therefore, we randomized the fMRI run order independently for each participant and repeated the FC computation (and prediction) using the “first” T minutes of resting-state fMRI data under the randomized run order. The run randomization improved the goodness of fit of both theoretical and logarithmic models (Figures 5B, 5C, S15 and S16).

pg 20 (Discussion)

Another caveat is that the empirically informed reference is less useful for phenotypes whose prediction accuracies are strongly influenced by non-stationarity in fMRI-phenotype relationships. Arousal changes between or during resting-state scans are well-established (Tagliazucchi et al., 2014; Wang et al., 2016; Bijsterbosch et al., 2017; Laumann et al., 2017; Orban et al., 2020), so we expect the fMRI to be non-stationary especially for longer scans. However, since run randomization affected some phenotypes more than others (Figure 5), this suggests that there is an interaction between fMRI non-stationarity and phenotypes, i.e., there appears to be a non-stationary relationship between fMRI and phenotypes.

However, we should not over-emphasize the effect of non-stationarity in fMRI-phenotype relationship. The primary effect of increasing scan time is increasing FC reliability, while the non-stationarity of fMRI-phenotype relationship is a secondary effect. Indeed, while randomizing run order improves the fit of the theoretical and logarithmic models, the 1% optimal scan time range was similar to the main analyses (Figure S27). Furthermore, explicit state manipulation, such as asking participants to perform a task (Figure S23) or splitting the data collection into two separate days (Figure S26) also yielded highly overlapping optimal scan time range with the main analyses. Nevertheless, it is possible that some other (undiscovered) state manipulation could modify the 1% optimal scan time range significantly.

(RIC9) While some measures show diminishing returns for scan duration after 20 minutes, others shown in the supplement continue to improve throughout the testable scan duration, even doubling. In general, there was a lot of variability in the relationships between scan duration and participant number across measures. Understanding what might cause this variation seems important if practical suggestions are likely to be made from this work. I would appreciate the authors digging into this issue in a bit more detail, and providing more nuance in their text to characterize this variation.

We thank the reviewer for this suggestion. Figure S25 shows the 1% optimal scan time ranges for different phenotypic domains. Figure S25 can be found in our response to E1C1. We found that all phenotypic domains exhibited highly overlapping 1% optimal scan time ranges, with the exception of the “emotion” domain. However, there is only one phenotype in the emotion domain, so the deviation should not be over-interpreted.

However, we agree with the reviewer that there is significant variation across phenotypes within a phenotypic domain. Therefore, in the web application, we allow researchers to explore different phenotypic domains. We have updated the manuscript as follows.

pg 19 (Discussion)

We also emphasize that although the trade-off between scan time and sample size are similar across phenotypic domains, there exists variation within phenotypic domains. Therefore, our web application also allows users to select different phenotypes.

(RIC10) Why does Fig. 3B show spatial orientation rather than the cognition factor score shown elsewhere? I think it would be more fair to carry the same measure forward, rather to swap to a new measure (out of all of the measures, this shows perhaps the least increase with increasing scan amount, so it doesn't feel representative of the full group)

We thank the reviewer for this suggestion. Our original intention was to provide an illustration of the theoretical model working for another phenotype. Given the reviewer’s suggestion, we have replaced “Spatial Orientation” with the cognition factor score. We have updated the manuscript as follows.

Figure 3. As scan time increases, sample size eventually becomes more important than scan time. (A) Prediction accuracy of the HCP cognition factor score when total scan duration is fixed at 6000 minutes, while varying scan time per participant from 10 to 60 minutes. Each violin plot shows the distribution of prediction accuracies across 50 random cross-validation splits. * indicates that the distributions of prediction accuracies were significantly different between adjacent pairs of sample sizes (and scan time per participant) after false discovery rate (FDR) $q <$

0.05 correction. (B) Scatter plot of prediction accuracy against total scan duration for the cognitive factor score in the HCP dataset. The curves were obtained by fitting a theoretical model to the prediction accuracies of the cognitive factor score. The theoretical model explains why sample size is more important than scan time (see main text).

(RIC11) Similar to above, can Fig. 2A results be shown on the same axes for better comparability? In general, I think it's important that equivalent measures in the ABCD and HCP be shown on the same axes. Similarly, can the reliability figures in S18 be shown on the same axes.

As mentioned in our response to RIC4, it is not easy to show HCP and ABCD results on the same scale (Figure 2A). Instead we have provided a supplementary figure showing strong correspondence between the two datasets in terms of prediction accuracy (see our response to RIC4).

The same issue applies to univariate and multivariate reliability. Therefore, we have opted to keep the two plots on different scales, but have now added supplementary figures showing strong correspondence between the two datasets in terms of reliability.

Figure S31. Scatter plot of the cognition factor univariate reliability (ICC) in the ABCD (x-axis) and HCP (y-axis) datasets. Each dot represents the prediction accuracy for each dataset with the same sample size and scan time per participant (extracted from Figure S30B). Although the cognitive factor score is not comparable across datasets, we observed a strong correlation between the two datasets ($r = 0.99$).

Figure S42. Scatter plot of the cognition factor univariate reliability (ICC) in the ABCD (x-axis) and HCP (y-axis) datasets. Each dot represents the prediction accuracy for each dataset with the same sample size and scan time per participant (extracted from Figure S30B). Although the cognitive factor score is not comparable across datasets, we observed a strong correlation between the two datasets ($r = 0.99$).

(RIC12) There are measures that cannot be made without more extensive data per participant (and vice versa). That is, the current work only characterizes sample size/scan duration dependency in a very specific set of (albeit popular) analyses. I think it would be useful for the discussion to grapple with these more complex issues and to provide some context of how these results are relevant (or not) to these other scenarios.

We thank the reviewer for this suggestion. We agree that for certain studies, an extensive amount of data per participant is necessary and unavoidable. Conversely, there might be other studies where minimum scan time is needed. We have updated our discussion as follows.

Finally, we note that beyond economic considerations, the diversity of the data sample and the generalizability of predictive models to subpopulations are also important factors when designing a study (Benkarim et al., 2022; Greene et al., 2022; Li et al., 2022; Kopal et al., 2023; Gell et al., 2024). There might also be studies where extensive scan time per participant is unavoidable. For example, when studying sleep stages, it is not easy to predict how long a participant would need to enter a particular sleep stage. Conversely, some phenomena of interest might be inherently short-lived. For example, if the goal is to study a fast acting drug (e.g., nitrous oxide), then it might not make sense to collect long fMRI scans. Furthermore, not all studies are interested in cross-sectional relationships between brain and non-brain-imaging phenotypes. For example, in the case of personalized brain stimulation (Cash et al., 2021; Lynch et al., 2022) or neurosurgical planning (Boutet et al., 2021), significant quantity of resting-state fMRI data might be necessary for accurate individual-level network estimation (Laumann et al., 2015; Braga et al., 2017; Gordon et al., 2017).

(RIC13) I was surprised that regular FD was used for the HCP data, given evidence (similar to in the ABCD) that a filtered version of FD might be more appropriate.

We agree with the reviewer that a filtered FD might be better. However, this study collates datasets across multiple labs and institutions. Even within the same lab, a dataset might have been preprocessed by different individuals many years apart. As such, the preprocessing was not consistent across datasets. Another variation is that some datasets were projected to fsaverage space, while other datasets were projected to MNI152 and fsLR space. We suggest that this is a strength of the study because the consistency of our results across datasets and analyses (Figures S24 to S29) indicates the robustness of our findings. We have added this point to the manuscript.

pg 29 (Methods)

We note that the above preprocessed data was collated across multiple labs, and even within the same lab, datasets were processed by different individuals many years apart. This led to significant preprocessing heterogeneity across datasets. For example, raw FD was used in the HCP dataset because it was processed many years ago, while the more recently processed ABCD dataset utilized a filtered version of FD, which has been shown to more effective. Another variation is that some datasets were projected to fsaverage space, while other datasets were projected to MNI152 and fsLR space. We considered this heterogeneity a strength of our study. Indeed, the consistency of our results across datasets and analyses (Figures S24 to S26) indicate that the robustness of our findings.

(RIC14) In the derivations shown in the supplement, I didn't understand the S8 extension – could some more text be added here? What does the limitation of $t = 0.5$ indicate? What is t ?

We apologize for being unclear. First, there was a typo in the text. The “ $f(y)$ ” in the original text should be “ $f(t)$ ”. The corrected text is as follows:

pg 2 (Supplement)

Note that it will be useful to approximate this with a 2nd order Taylor series approximation for $f(t) = (1 - t^2)^2$ about $t = 0$, $1 - 2t^2$

$$V(\hat{\rho}_j) \approx \frac{1}{N} \left(1 - 2\rho_j^2 R(X_{Tj})R(Y) \right) \quad (\text{S8})$$

which we find to be fairly accurate up through $t = \rho_j \sqrt{R(X_{Tj})R(Y)} = 0.5, \dots$

Second, we note that this approximation was used in the reliability analysis (and not the prediction accuracy analysis). In this context, $\rho_j \sqrt{R(X_{Tj})R(Y)}$ is the true correlation between FC edge j and a phenotype for a fMRI acquisition of length T .

Obviously, we do have access to the true correlation between FC edge j and a phenotype, but we can compute the actual correlation between FC and a phenotype to check how good is this approximation. The phenotype with the strongest correlation with FC is the cognitive factor score. In the case of the HCP dataset, across all edges, the largest absolute correlation between FC and the cognitive factor was 0.27, while in the case of the ABCD dataset, across all edges, the largest absolute correlation between FC and the cognitive factor was 0.22. Therefore, we believe that this is a good approximation in practice.

We have updated the supplement to be clearer.

pg 2 (Supplement)

where as a reminder, $\rho_j \sqrt{R(X_{Tj})R(Y)}$ is the true correlation between FC edge j and a phenotype for a fMRI acquisition of length T . We will use this approximation for the reliability analysis (Supplementary Methods S1.3). Obviously, we do not know the true correlation between FC edge j and a phenotype. However, we can compute the actual correlation between FC and a phenotype to check the quality of the approximation. The phenotype with the strongest correlation with FC is the cognitive factor score. In the case of the HCP dataset, across all edges, the largest absolute correlation between FC and the cognitive factor was 0.27, while in the case of the ABCD dataset, across all edges, the largest absolute correlation between FC and the cognitive factor was 0.22. As the strongest correlation is much smaller than 0.5, we believe that our approximation is good.

(RIC15) In association with Figure 4, the authors say that the model does poorly in some cases due to “intrinsic predictability” of measures. What does that mean?

We apologize for being unclear. We simply meant that the prediction was poor. We have updated the sentence in the manuscript to be clearer.

pg 13 (Results)

These findings suggest that the imperfect fit of the theoretical and logarithmic models for some phenotypes may be due to their poor predictability, rather than true variation in prediction accuracy with respect to sample size and scan time.

(RIC16) Fig. 5: are C and D for all measures, or just anger/aggression?

We apologize for being unclear. Figures 5C and 5D use all phenotypes with the looser prediction accuracy threshold of Pearson's r being positive in at least 90% of all combinations of sample size N and scan time T . We have updated the manuscript to be clearer.

pg 14 (Figure 5 captions)

(C) Box plots showing goodness of fit to logarithmic and theoretical models before and after randomizing fMRI run order for 33 ABCD phenotypes. * indicates that goodness-of-fits were significantly different (after FDR correction with $q < 0.05$). (D) Same as panel C, but for 42 HCP phenotypes. For all panels, model fit was performed using the maximum scan time per participant. For panels C and D, we considered all phenotypes whose prediction accuracies (Pearson's r) were positive in at least 90% of all combinations of N and T . For each boxplot, the horizontal line indicates the median across 33 ABCD phenotypes or 42 HCP phenotypes. The bottom and top edges of the box indicate the 25th and 75th percentiles, respectively. Outliers are defined as data points beyond 1.5 times the interquartile range. The whiskers extend to the most extreme data points not considered outliers.

Referee #2:

(R2C1) This study investigates the impact of sample size and fMRI scan duration on phenotypic predictions using resting-state fMRI (rs-fMRI) data from two large datasets, ABCD and HCP. It reveals that while sample size and scan duration are largely interchangeable in enhancing prediction accuracy, the benefits of extended scan duration diminish beyond 30 minutes, making sample size the more critical factor. Additionally, the study introduces a theoretical model capable of estimating prediction accuracy based on these variables, accompanied by a web application designed to guide future research planning.

The research underscores the importance of considering scan duration in study design and power calculations, a point well-acknowledged in previous studies or discussions on reliability or replicability (Birn et al., 2013; Nee, 2019; Gordon et al., 2017) but less so in the context of brain-wide association studies (BWAS).

However, the study could benefit from a more detailed exploration of why scan time versus sample size is important for phenotypic prediction. Although the authors concluded these factors are broadly interchangeable, this assertion may only hold superficially, as their beneficial effects likely stem from different mechanisms. For example, longer scan durations might enhance the reliability of brain measures, whereas larger sample sizes could improve model robustness by increasing data variability and thus reducing the risk of overfitting.

We thank the reviewer for the careful read of our manuscript. The reviewer is concerned that our results might only hold superficially. In the revision, we have added many additional analyses, demonstrating that our theoretical formula generalizes to task-FC, clinical populations, other racial groups, other age groups, acquisition protocols (single band and multi-echo multi-band). Please see our response to E1C1. As such, we believe that our results are empirically general.

We apologize for not being clear about the mechanism of sample size N and scan time T . We agree with the reviewer that N and T contribute to prediction accuracy with different underlying mechanisms. This differing mechanism is why there are separate $1/N$ and $1/NT$ terms in our theoretical derivation of prediction accuracy. In addition, N and T are in the $1/NT$ term for different reasons, as the reviewer correctly points out! By estimating the relative influence of $1/N$ and $1/NT$ terms, we find that N and T are interchangeable up to around 20 minutes (on average), but as T increases, the $1/N$ term becomes much more important than the $1/NT$ term.

We also note that the theoretical derivation actually provides insights into the mechanism of N and T (which we were not previously clear in the original submission). We note that phenotypic prediction can be theoretically decomposed into two components: one component relating to an average prediction (common to all participants) and a second component relating to a participant's deviation from this average prediction. The uncertainty (variance) of the first component scales as $1/N$, like a conventional standard error of the mean.

The uncertainty (variance) of the second component can be shown to scale with $1/NT$. To provide some intuition, phenotypic prediction is equal to regression coefficients \times FC (assuming linear regression). The uncertainty of the regression coefficient estimates scales with $1/N$, and

the uncertainty of the FC estimates scale with $1/T$. Thus, the uncertainty of the second component scales with $1/N \times 1/T$ or $1/NT$. We note that this explanation is a simplification, as the exact results depend on the specific FC values, but we hope it is helpful for illustrating why prediction accuracy is a function of $1/N$ and $1/NT$.

Our theoretical derivation does make several simplifying assumptions. For example, we assume that the uncertainty of FC scales with $1/T$, which is only true if the fMRI time series is stationary (i.e., the statistical properties of fMRI stay the same throughout the entire fMRI scan). Furthermore, the theoretical derivations do not tell us the relative importance of the $1/N$ and $1/NT$ terms. Therefore, in our study, we fitted the three parameters of the theoretical model to actual prediction accuracies of many phenotypes and datasets.

The goal is to determine (1) whether our theoretical model (despite the simplifying assumptions) can still explain the empirical results, and (2) to determine the relative importance of $1/N$ and $1/NT$, so that we can inform future BWAS study design. The extensive results in our previous manuscript and revision suggest that our simplifying assumptions do not break the model, i.e., the model is able to explain the observed phenomena under a wide range of conditions. Indeed, when T is small, the $1/NT$ term dominates the $1/N$ term, so N and T are almost interchangeable. As T increases, the $1/NT$ term becomes small, so the $1/N$ term dominates the prediction accuracy.

Finally, we note that our simplifying assumption that the uncertainty of FC scales with $1/T$ is exactly the reviewer's suggestion that "longer scan durations might enhance the reliability of brain measures".

We have updated the manuscript as follows:

pg 10 (Results)

Even though sample size and scan time are broadly interchangeable, there is a diminishing return of scan time per participant relative to sample size (Figure 3A). To gain insights into this phenomenon, we derived a closed-form mathematical relationship relating prediction accuracy (Pearson's correlation) with scan time per participant T and sample size N (see Methods). To provide an intuition for the theoretical derivations, we note that phenotypic prediction can be theoretically decomposed into two components: one component relating to an average prediction (common to all participants) and a second component relating to a participant's deviation from this average prediction.

The uncertainty (variance) of the first component scales as $1/N$, like a conventional standard error of the mean. For the second component, we note that the prediction can be written as regression coefficients \times FC (for linear regression). The uncertainty (variance) of the regression coefficient estimates scales with $1/N$. The uncertainty (variance) of the FC estimates scales with $1/T$ (i.e. reliability improves with T). Thus, the uncertainty of the second component scales with $1/NT$. Overall, our theoretical derivation suggests that prediction accuracy can be expressed as a function of $1/N$ and $1/NT$ with three free parameters.

There were several simplifying assumptions in the theoretical derivations. Furthermore, the theoretical derivations did not tell us the relative importance of the $1/N$ and $1/NT$ terms. Therefore, we fitted the theoretical model to actual prediction accuracies in the HCP and ABCD datasets (Figure 1B). The goal was to determine (1) whether our theoretical model (despite the simplifying assumptions) would still explain the empirical results, and (2) to determine the relative importance of $1/N$ and $1/NT$.

We found an excellent fit with actual prediction accuracies for the 19 HCP and 17 ABCD phenotypes (Figures 3B, S12 & S13): $R^2 = 0.89$ and 0.90 respectively (Table S1B). This suggests that our theoretical model was able to explain the observed phenomena despite the simplifying assumptions. When T was small, the $1/NT$ term dominated the $1/N$ term, which explained the almost 1-to-1 interchangeability between scan time and sample size for shorter scan time. The existence of the $1/N$ term ensured that sample size was still slightly more important than scan time even for small T . FC reliability eventually saturated with increasing T . Therefore, the $1/N$ term eventually dominated the $1/NT$ term, so sample size became much more important than scan time.

We note that for shorter scan time, the logarithmic and theoretical models performed equally well with equivalent goodness of fit (R^2) across the 17 ABCD phenotypes ($p = 0.57$). For longer scan time, the theoretical model exhibited better fit than the logarithmic model ($p = 0.002$ across the 19 HCP phenotypes; Figure S14). Furthermore, prediction accuracy under the logarithmic model will exceed a correlation of one for sufficiently large N and T , which is obviously wrong. Therefore, we will use the theoretical model to derive a reference for future study design (in a later section).

(R2C2) Their model, primarily descriptive, appears suited only to the specific context of rsfMRI-based BWAS prediction for cognitive phenotypes. While there is a growing body of literature that highlights the advantages of task-based data for phenotypic prediction (Zhao et al., 2023; Greene et al., 2018), the generalizability of the study's findings and models to task fMRI-based BWAS remains uncertain.

We agree with the reviewer that the logarithmic model is descriptive. Indeed, we originally came up with the logarithmic model by visual inspection. However, we hope that the intuition we provide to our response to R2C1 has demonstrated that our theoretical model is *not* merely descriptive, but provides insights into how sample size N and scan time T might influence prediction accuracy through different mechanisms. Indeed, based on the theoretical derivation, the theoretical model should also be applicable to task-FC, although it is unclear whether the trade-off between sample size and scan time is the same for task-FC compared with RSFC.

Therefore, as suggested by the reviewer, we have included task-FC from the ABCD study in our revision. Consistent with previous work (Greene et al., 2018; Chen et al., 2022; Zhao et al., 2023), n-back FC was better than RSFC in predicting certain phenotypic measures, but the 1%

optimum scan time ranges were highly overlapping across resting-state and the three tasks (Figure S26). This speaks to the generalizability of our theoretical model.

We note that Figure S26 can be found in our response to E1C1, but for your convenience, we have included relevant text from the result section.

pg 17 (Results)

Different phenotypic domains also yielded highly overlapping 1% optimum scan time ranges (Figure S25). Furthermore, replicating previous studies (Greene et al., 2018; Chen et al., 2022; Zhao et al., 2023), n-back task-FC yielded better prediction performance for cognitive measures compared with RSFC in the ABCD dataset. However, although the optimal scan time was slightly higher for resting-state and the SST-task, the 1% optimum scan time ranges were highly overlapping across resting-state and the three tasks (Figure S26).

(R2C3) Similarly, it is ambiguous whether the model is applicable to clinical data, although the authors imply that their results have clinical implications (e.g., for children with ADHD, as discussed in the Discussion section). A deeper mechanistic understanding of how scan time and sample size enhance phenotypic prediction could facilitate the application of these findings to broader contexts.

We thank the reviewer for the excellent suggestion. We have now added three clinical datasets. The MDD dataset contains participants with major depressive disorder. The ADNI dataset contains a mix of participants that have AD dementia, have mild cognitive impairment (MCI) or are cognitively normal. The TCP dataset is a transdiagnostic psychiatric dataset containing a mix of healthy control and participants with psychiatry disorders. We found that the 1% optimal scan time range was highly overlapping between the clinical datasets (MDD, ADNI and TCP) and the original analyses (ABCD and HCP). Again, this speaks to the generalizability of our theoretical model. Please see our response to E1C1 for more details.

(R2C4) The analyses and interpretations of the state effects were intriguing; however, these remain speculative due to the lack of direct measures of states in the data. More direct evidence of state-related measures is required to substantiate these interpretations, thus necessitating caution.

We agree with the reviewer. Since we did not have direct brain state measurements for the run randomization analysis, we have modified the corresponding result section to not reference brain states, but instead uses a more neutral language of “non-stationarity in fMRI-phenotype relationship”. The reference to “brain states” has been moved to the discussion.

Furthermore, based on the reviewer’s comment (R2C5), we have added another analysis in which we considered the situation in which resting-state fMRI was collected in two separate sessions. This analysis was possible because HCP collected data on two separate days. We found splitting data collection into two sessions slightly increase optimal scan time, but the 1% optimal scan time ranges still highly overlap between collecting data in a single session versus two sessions (Figure S29).

In addition, we considered FC computed during a task (as suggested by the reviewer's comment R2C2), which is a manipulation of brain state, and again found that the 1% optimal scan time range was highly similar between rest and tasks (Figure S26).

Figures S26 and S29 can be found in our response to E1C1. However, for the reviewer's convenience, relevant text from the results and discussion is copied here.

pg 18 (Results)

Given the observed non-stationarity effects (Figure 5), we simulated splitting data collection into two separate sessions. This simulation is possible because the HCP collected resting-state fMRI on two different days (sessions). We found splitting data collection into two sessions slightly increase optimal scan time, but the 1% optimal scan time ranges still highly overlapped when collecting data in a single session versus two sessions (Figure S29).

pg 20 (Discussion)

Another caveat is that the empirically informed reference is less useful for phenotypes whose prediction accuracies are strongly influenced by non-stationarity in fMRI-phenotype relationships. Arousal changes between or during resting-state scans are well-established (Tagliazucchi et al., 2014; Wang et al., 2016; Bijsterbosch et al., 2017; Laumann et al., 2017; Orban et al., 2020), so we expect the fMRI to be non-stationary especially for longer scans. However, since run randomization affected some phenotypes more than others (Figure 5), this suggests that there is an interaction between fMRI non-stationarity and phenotypes, i.e., there appears to be a non-stationary relationship between fMRI and phenotypes.

However, we should not over-emphasize the effect of non-stationarity in fMRI-phenotype relationship. The primary effect of increasing scan time is increasing FC reliability, while the non-stationarity of fMRI-phenotype relationship is a secondary effect. Indeed, while randomizing run order improves the fit of the theoretical and logarithmic models, the 1% optimal scan time range was similar to the main analyses (Figure S27). Furthermore, explicit state manipulation, such as asking participants to perform a task (Figure S23) or splitting the data collection into two separate days (Figure S26) also yielded highly overlapping optimal scan time range with the main analyses. Nevertheless, it is possible that some other (undiscovered) state manipulation could modify the 1% optimal scan time range significantly.

(R2C5) However, their interpretation of state effects indeed raises an important point—even with increasing data for an individual, the benefits of such data could stem from multiple factors. For instance, longer scan times might enhance the reliability of measures, but they could also introduce greater within-individual variability (i.e., state-related variability) in the training data. In this context, "longer scan time" encompasses various meanings and can be implemented in several ways. For example, to target reliability, a single-session long scan (e.g., a 30-minute resting scan in one session) may be effective, while to maximize within-individual variability, the scan might be segmented into multiple short scan sessions (e.g., three 10-minute resting scans across different days). Alternatively, simple state manipulation procedures could be used to increase within-individual variability. Therefore, it is conceivable that extensive scan times may not be necessary if sufficient within-individual variability can be induced. This suggests that a broad assertion such as "many studies would benefit from longer scan time per participant" could be misleading without a more detailed and nuanced understanding of the specific advantages conferred by longer scans.

We thank the reviewer for this comment. Please see our response to R2C4.

(R2C6) Their main findings derive from the averaged prediction accuracies across 10-fold cross-validation and 50 repetitions with random participant splits. While average is a key metric of prediction performance, variance represents another significant aspect. Figure 3A clearly shows the differential impacts of sample size and scan time on both the variance and the mean of prediction accuracy—specifically, a smaller sample size results in higher variance in test outcomes. Considering that clinical applications or independent testing of models in prospective studies typically do not involve multiple repetitions, this variance becomes an essential factor to consider.

We thank the reviewer for this excellent point! The reviewer is correct that smaller sample sizes lead to higher variability in cross-validation folds (Varoquaux, 2018). In the revised manuscript, we have now introduced the concept of an optimal range of scan time over which studies can achieve within 1% of the maximum prediction accuracy. Therefore, in the small sample size scenario, researchers might consider going for a larger sample size (at the expense of scan time) within the 1% optimal scan time range to minimize variability (left side of the 1% optimal scan time range).

On the other hand, we note that an opposing consideration is that the 1% optimal scan time range is asymmetric with a longer tail towards longer scan time. Therefore, erring towards longer scan time (at the expense of sample size) might increase the chance that the resulting scan time falls within the 1% optimal scan time range for the particular experiment.

We have updated the manuscript with the above considerations, while not being overly prescriptive, so as to allow greater flexibility for researchers.

pg 19 (Discussion)

The 1% optimal scan time range provides greater flexibility in modifying study designs based on population- and site-specific characteristics. For example, a researcher seeking to study patients with depression from a minority population (i.e., higher overhead cost

per participant) might find it more economical to increase the scan time for each participant in order to achieve the maximum possible prediction accuracy. Indeed, because the 1% optimal scan time range has a longer tail towards longer scan time, erring towards longer scan time (at the expense of sample size) increases the chance that the resulting scan time falls within the 1% optimal scan time range for the particular experiment. On the other hand, when sample size is small (in the low hundreds), there is a lot variability across cross-validation folds (Figure 3A; Varoquaux, 2018). Therefore, a case can also be made to prioritize sample size over scan time (left tail of the optimal scan time range).

(R2C7) I wonder if their theoretical model works better even after considering the increased number of free parameters.

We have now added a statistical test of the COD difference between the logarithmic and theoretical models in the HCP and ABCD datasets. As expected, the COD of the theoretical model was better than the logarithmic model in the HCP dataset ($p = 0.002$), while the two models are equivalent in the ABCD dataset ($p = 0.57$).

It is true that the theoretical model has more parameters than the logarithmic model – 3 versus 2 parameters. However, we believe that this additional parameter is critical given that for many phenotypes, there is a clear separation in prediction accuracies for different sample sizes N (regardless of scan time T). This is most obvious in the HCP dataset, which has the longest scan times, but we also observe this effect for some phenotypes in datasets with shorter scan time (Figures S17 to S20).

Finally, we note that the logarithmic model allows for a prediction accuracy (Pearson's correlation) greater than one for sufficiently large sample size and scan time, which is obviously wrong. The theoretical model does not have this problem. Therefore, it makes sense to use the theoretical model (rather than the logarithmic model) to generate a reference for future study design.

We have updated the manuscript as follows.

pg 11 (Results)

We note that for shorter scan time, the logarithmic and theoretical models performed equally well with equivalent goodness of fit (R^2) across the 17 ABCD phenotypes ($p = 0.57$). For longer scan time, the theoretical model exhibited better fit than the logarithmic model ($p = 0.002$ across the 19 HCP phenotypes; Figure S14). Furthermore, prediction accuracy under the logarithmic model will exceed a correlation of one for sufficiently large N and T , which is obviously wrong. Therefore, we will use the theoretical model to derive a reference for future study design (in a later section).

Figure S14. Visual comparison of logarithmic and theoretical models fitted to three HCP phenotypes using the full 58 minutes of data. Because the logarithm model treated the sample size N and scan time T as being interchangeable, it was not able to explain the diminishing returns of scan time T relative to sample size N for larger value of T .

(R2C8) The abstract says prediction accuracy “might benefit from much longer scan durations (>50 min),” but it appears somewhat contradictory to the observed diminishing returns on prediction accuracy when extending beyond 30 minutes. The study indicates that sample size and scan duration are interchangeable for up to 30 minutes in the HCP dataset. This discrepancy suggests a need for clarity or revision.

We apologize for being unclear. We have updated the abstract as follows. Hopefully, this will not confuse the reader anymore.

pg 3 (Abstract)

A pervasive dilemma in neuroimaging is whether to prioritize sample size or scan time given fixed resources. Here, we systematically investigate this trade-off in the context of brain-wide association studies (BWAS) using functional magnetic resonance imaging (fMRI). We find that total scan duration (sample size \times scan time per participant) robustly explains individual-level phenotypic prediction accuracy via a logarithmic model, suggesting that sample size and scan time are broadly interchangeable up to 20-30 min of data. However, the returns of scan time diminish relative to sample size, which we explain with principled theoretical derivations. When accounting for fixed overhead costs associated with each participant (e.g., recruitment, non-imaging measures), prediction accuracy in many small-scale and some large-scale BWAS might benefit from longer scan time than typically assumed. These results generalize across phenotypic domains, scanners, acquisition protocols, racial groups, mental disorders, age groups, as well as resting-state and task-state functional connectivity. Overall, our study emphasizes the importance of scan time, which is ignored in standard power calculations. Standard power calculations maximize sample size, at the expense of scan time, which can result in sub-optimal prediction accuracies and inefficient use of resources. Our empirically informed reference is available for future study design: WEB_APPLICATION_LINK

(R2C9) For the ABCD data, the researchers employed a leave-3-site-clusters-out nested cross-validation approach, i.e., 7 vs. 3 splits, whereas for the HCP data, they used a 10-fold nested cross-validation strategy, i.e., 9 vs. 1 splits. Clarification about this analysis decision would be helpful.

We apologize for being unclear. We have now updated this section of the manuscript to explain the inconsistency.

pg 30 (Methods)

Instead of the 10-fold inner-loop (nested) cross-validation procedure in the HCP dataset, we performed a leave-3-site-clusters-out inner-loop (nested) cross-validation (i.e., 7 site-clusters are used for training and 3 site-clusters are used for testing) in the ABCD (rest and task) dataset. The hyperparameter was again selected using a 10-fold CV within the training set. This nested cross-validation procedure was performed for every possible split of the site clusters, resulting in 120 replications. The prediction accuracies were averaged across all 120 replications.

We did not perform a leave-one-site-cluster-out procedure because the site-clusters are “fixed”, so the cross-validation procedure can only be repeated 10 times under a leave-one-site-cluster-out scenario (instead of 120 times). Similarly, we did not go for leave-two-site-clusters-out procedure because that will only yield a maximum of 45 repetitions of cross-validation. On the other hand, if we left more than 3 site clusters out (e.g., leave-5-site-clusters-out), we could achieve more cross-validation repetitions, but at the cost of reducing the maximum training set size. Therefore, we opted for the leave-3-site-clusters-out procedure, consistent with our previous study (Chen et al., 2022).

Referee #3:

(R3C1) In this study, Ooi et al. examined the issues of sample size and scan duration selection in brain-phenotype prediction modeling and association analysis. The topic is important and practical, and the conclusion will be very helpful for many studies in relevant fields. Most of the analyses are appropriate and sound. The manuscript is well written and very informative. However, I have some concerns that should be adequately addressed before it is considered for publication. I have described my points below and hope that the authors will find them helpful in improving this manuscript.

We thank the reviewer for the careful review of our work.

(R3C2) My main concerns are from the randomization analysis regarding brain “states”. Fig 5CD visualizes the results from the maximum scan time. I am sorry if I misunderstood something, but functional connectivity by correlation does not depend on the order of the time points. Therefore, it is unclear why the original and randomized results differ in the figure panels.

The authors mentioned “state” effects during the resting scan and used randomization to address this issue. However, brain state within the resting scan was never directly defined, examined, and demonstrated in this study. The authors did not provide any practical or theoretical evidence for different states within the resting state fMRI. I would like to ask for a clear definition of states and direct evidence supporting different brain states within the resting period.

Together, I found the randomization analysis and results needs clarification.

We apologize for being unclear. The reviewer is correct that the FC of two time series is the same regardless of the ordering of time points within the time series. However, in our main analysis (Figures 1 to 4 and 6), we vary the length of the scan time T by considering the first T minutes of the data. For example, if T = 5 minutes, then we are computing FC based on the first 5 minutes of fMRI. However, one can imagine that the computed FC might be different if we consider the last 5 minutes of fMRI, instead of the first 5 minutes.

Let’s consider an example ABCD participant with the original run order (run 1, run 2, run 3, run 4). Each run is 5 minutes long. In the original analysis (Figures 1 to 4 and 6), if T is 5 minutes, then we use all the data from run 1 to compute FC. If T is 10 minutes, then we are using run 1 and run 2 to compute FC. If T is 15 minutes, then we are using runs 1, 2 and 3 to compute FC. Finally, if T is 20 minutes, we are using all 4 runs to compute FC.

After performing run randomization, let’s say this specific participant’s run order has become run 3, run 2, run 4, run 1. In this situation, if T is 5 minutes, then we use all the data from run 3 to compute FC. If T is 10 minutes, then we are using run 3 and run 2 to compute FC. If T is 15 minutes, then we are using runs 3, 2 and 4 to compute FC. Finally, if T is 20 minutes, we are using all 4 runs to compute FC. We hope this explanation has made the analysis clearer.

Furthermore, the reviewer is correct that we did not have direct state measurements. Therefore, we have modified the corresponding result section to not reference brain states, but instead uses a more neutral language of “non-stationarity in fMRI-phenotype relationship”. The reference to “brain states” has been moved to the discussion.

We have updated the manuscript as follows.

pg 13 (Results)

This suggests that fMRI-phenotype relationships might be non-stationary for certain phenotypes, which violates an assumption in both theoretical and logarithmic models. To put this in more colloquial terms, the assumption is that the relationship between FC and a phenotypic measure is the same (i.e., stationary) regardless of whether FC was computed based on five minutes of fMRI from the beginning, middle or end of the MRI session. To test the non-stationarity hypothesis, we note that for both HCP and ABCD datasets, the fMRI data was collected over four runs. Therefore, we randomized the fMRI run order independently for each participant and repeated the FC computation (and prediction) using the “first” T minutes of resting-state fMRI data under the randomized run order. The run randomization improved the goodness of fit of both theoretical and logarithmic models (Figures 5B, 5C, S15 and S16).

pg 20 (Discussion)

Another caveat is that the empirically informed reference is less useful for phenotypes whose prediction accuracies are strongly influenced by non-stationarity in fMRI-phenotype relationships. Arousal changes between or during resting-state scans are well-established (Tagliazucchi et al., 2014; Wang et al., 2016; Bijsterbosch et al., 2017; Laumann et al., 2017; Orban et al., 2020), so we expect the fMRI to be non-stationary especially for longer scans. However, since run randomization affected some phenotypes more than others (Figure 5), this suggests that there is an interaction between fMRI non-stationarity and phenotypes, i.e., there appears to be a non-stationary relationship between fMRI and phenotypes.

pg 34 (Methods)

To elaborate further, let us consider an ABCD participant with the original run order (run 1, run 2, run 3, run 4). Each run was 5 minutes long. In the original analysis, if scan time T was 5 minutes, then we used all the data from run 1 to compute FC. If scan time T was 10 minutes, then we used run 1 and run 2 to compute FC. If scan time T was 15 minutes, then we used runs 1, 2 and 3 to compute FC. Finally, if scan time T was 20 minutes, we used all 4 runs to compute FC.

On the other hand, after run randomization, for the purpose of this exposition, let us assume this specific participant’s run order had become run 3, run 2, run 4, run 1. In this situation, if scan time T was 5 minutes, then we used all data from run 3 to compute FC. If scan time T was 10 minutes, then we used run 3 and run 2 to compute FC. If scan time

T was 15 minutes, then we used runs 3, 2 and 4 to compute FC. Finally, if T was 20 minutes, we used all 4 runs to compute FC.

(R3C3) One important finding of this study is the fit of logarithmic and theoretical models explaining the relationship between prediction accuracy and total scan time. The manuscript argues that the theoretical models fit better. However, the authors did not perform a statistical analysis comparing the fit of two models. Although two measures are numerically very different, the authors would want to add statistical results.

We have now added a statistical test of the COD difference between the logarithmic and theoretical models in the HCP and ABCD datasets. As expected, the COD of the theoretical model was better than the logarithmic model in the HCP dataset ($p = 0.002$), while the two models are equivalent in the ABCD dataset ($p = 0.57$).

It is also worth noting the logarithmic model allows for a prediction accuracy (Pearson's correlation) greater than one for sufficiently large sample size and scan time, which is obviously wrong. The theoretical model does not have this problem. Therefore, it makes sense to use the theoretical model (rather than the logarithmic model) to generate a reference for future study design.

We have updated the manuscript as follows.

pg 11 (Results)

We note that for shorter scan time, the logarithmic and theoretical models performed equally well with equivalent goodness of fit (R^2) across the 17 ABCD phenotypes ($p = 0.57$). For longer scan time, the theoretical model exhibited better fit than the logarithmic model ($p = 0.002$ across the 19 HCP phenotypes; Figure S14). Furthermore, prediction accuracy under the logarithmic model will exceed a correlation of one for sufficiently large N and T, which is obviously wrong. Therefore, we will use the theoretical model to derive a reference for future study design (in a later section).

Figure S14. Visual comparison of logarithmic and theoretical models fitted to three HCP phenotypes using the full 58 minutes of data. Because the logarithm model treated the sample size N and scan time T as being interchangeable, it was not able to explain the diminishing returns of scan time T relative to sample size N for larger value of T .

(R3C4) One of the target phenotypes is the cognition factor. If the factor was defined and estimated by the HCP and ABCD samples used in this manuscript, this is data leakage that inflates the prediction accuracy. If the factor was defined by an independent set of small samples, I would like to ask how stable and reliable the factor estimation is.

The HCP and ABCD factors were from a previous study and were previously derived out-of-sample in that study (Ooi et al., 2022). However, in the current study, because we re-selected the participants, they are not out-of-sample.

To address the reviewer's comments, below we re-computed cognitive factor scores from 230 HCP and 7415 ABCD participants, who did not overlap with the main analyses. The correlation between the manuscript's cognitive factor score and the new cognitive factor score is 0.95 and 1 respectively in the HCP and ABCD datasets.

Furthermore, the prediction accuracies are almost identical between the original and new factor scores (see figure below). However, given the reviewer's comment, we have replaced our old factor scores with the new factor scores in all analyses and figures.

This figure juxtaposes the prediction accuracies obtained with the original factor score (in the original submission) with the new factor score (in the revised manuscript). The results were almost identical.

(R3C5) The authors distinguish between BWA reliability and prediction accuracy. Predictive modeling requires external validation beyond internal CV for better demonstration. Although the authors replicate similar patterns of internal CV results using two datasets, HCP and ABCD, it does not necessarily mean the reliability and validity of the prediction results. Therefore, I would like to see how well the prediction models would work in external independent datasets.

We thank the reviewer for raising this point. The goal of our study is to inform the optimization of sample size and scan time for prediction accuracy under a range of different conditions. In our original submission, we have shown strong agreement between HCP and ABCD datasets. Indeed, for the same sample size and scan time, the prediction accuracies were highly similar across the two datasets ($r = 0.98$). Furthermore, in the revision, we have added 4 additional datasets, demonstrating highly overlapping optimal scan time range across all six datasets, as well as across resting-state and task-state FC (please see our detailed response to E1C1). As such, we believe that we have sufficiently achieved the goal of our study.

pg 6 (Results)

Furthermore, although the cognitive factors were not necessarily comparable across datasets, there was strong agreement in prediction accuracies between the two datasets for the same sample size and scan time per participant ($r = 0.98$; Figure S6).

Figure S6. Scatter plot of the cognition factor prediction accuracy in the ABCD (x-axis) and HCP (y-axis) datasets. Each dot represents the prediction accuracy for each dataset with the same sample size and scan time per participant (extracted from Figure 1B). Although the cognitive factor score is not comparable across datasets, we observed a strong correlation between the two datasets ($r = 0.98$).

The reviewer suggests that we apply the HCP model to ABCD (and vice versa). Cross-dataset prediction is indeed an interesting and complex problem. We believe the best way to improve cross-dataset prediction is to train models from multiple datasets jointly in order to maximize the diversity of the training data (Abraham et al., 2017; P. Chen et al., 2023), rather than train a model in one dataset and apply it to another dataset.

However, most studies and grants are not designed with the primary aim of combining the proposed collected data with other datasets. Instead, our study seeks to inform crucial decisions of study design within the scope of most current studies and grants. Therefore, we believe the reviewer's suggestion is outside the scope of the current study. We have updated the manuscript as follows.

pg 32 (Methods)

We also note that our study focused on out-of-sample prediction within the same dataset, but did not explore cross-dataset prediction (Wu et al., 2023). For predictive models to be clinically useful, these models must generalize to completely new datasets. The best way to achieve this goal is by training models from multiple datasets jointly, so as to maximize the diversity of the training data (Abraham et al., 2017; P. Chen et al., 2023). However, we did not consider cross-dataset prediction in the current study because most studies are not designed with the primary aim of combining the collected data with other datasets.

(R3C6) The authors found that KRR and LRR lead to similar conclusions across phenotype domains. I would like to see a discussion of this finding. What does it tell us about the underlying brain-phenotype associations? Are the associations always linear? Or does it mean that we need a lot more data to uncover true brain-phenotype associations?

In our previous studies, we have shown that several deep neural networks and kernel ridge regression lead to similar prediction accuracies when using FC to predict phenotypic traits (He et al., 2020). Furthermore, kernel ridge regression (KRR) and linear ridge regression (LRR) also lead to similar prediction accuracies (Chen et al., 2022). There was also a recent study that suggests that resting-state brain dynamics (as measured by fMRI and iEEG) can be best explained with linear autoregressive models (Nozari et al., 2024). Therefore, due to the challenges of in-vivo recordings, linear models might be sufficiently powerful to explain macroscopic brain measurements.

However, we emphasize that the current study is not trying to make a similar (potentially provocative) claim. Instead, we would only like to claim (and we believe our results support this claim) that trade-off between scan time and sample size appears to be similar for both KRR and LRR, as well as across phenotypic domains, scanners, acquisition protocols, racial groups, mental disorders, age groups, as well as resting-state and task-state functional connectivity (see our response to E1C1).

We have updated our manuscript as follows.

pg 33 (Methods)

Here we have chosen to use kernel ridge regression and linear regression because previous studies have shown that they have comparable prediction performance, and also exhibited similar prediction accuracies as several deep neural networks (He et al., 2020; Chen et al., 2022). Indeed, a recent study suggested that linear dynamical models provide better fit to resting-state brain dynamics (as measured by fMRI and intracranial electroencephalogram) than nonlinear models, suggesting that due to the challenges of in-vivo recordings, linear models might be sufficiently powerful to explain macroscopic brain measurements. However, we note that in the current study, we are not making a similar claim. Instead, our results suggest that the trade-off between scan time and sample size are similar for different regression models, and phenotypic domains, scanners,

acquisition protocols, racial groups, mental disorders, age groups, as well as resting-state and task-state functional connectivity.

(R3C7) ABCD has different imaging sites. I wonder if the site effect, which would add more inter-individual variance, compensates for the “state effect”, if it exists, which was stronger in HCP.

This is a very interesting point raised by the reviewer! The run randomization analysis does seem to increase COD more for the HCP dataset, compared with the ABCD dataset (Figures 5C and 5D), so this non-stationarity effect might indeed be a bigger factor in the HCP than ABCD dataset. As suggested by the reviewer, one possible reason could be the multi-site data collection in ABCD, which might increase inter-individual variation, thus “compensating” for this non-stationarity effect. Unfortunately, there are other plausible reasons for differences between HCP and ABCD, e.g., the HCP dataset involved much longer resting-state scans (58 min) compared with ABCD (20 minutes).

Therefore, we cannot really be sure what is really the cause of the larger non-stationarity effect in HCP, compared with ABCD. In addition, the reviewer rightfully pointed out (R3C2) that we did not have direct state measurements during the resting-state. We also agree with reviewer 1 (R1C8) that the main effect of increasing scan time is increasing FC reliability, which in turn improves prediction accuracy. While non-stationarity effects exist, they are secondary factors, as can be seen by the 1% optimal scan time range largely overlapping after randomizing run order. Therefore, in the revised manuscript, instead of elaborating on potential causes of greater non-stationarity effects in the HCP dataset, we have instead sought to better contextualize the non-stationarity effects as secondary to the primary effect of increasing reliability with scan time.

pg 20 (Discussion)

However, we should not over-emphasize the effect of non-stationarity in fMRI-phenotype relationship. The primary effect of increasing scan time is increasing FC reliability, while the non-stationarity of fMRI-phenotype relationship is a secondary effect. Indeed, while randomizing run order improves the fit of the theoretical and logarithmic models, the 1% optimal scan time range was similar to the main analyses (Figure S27). Furthermore, explicit state manipulation, such as asking participants to perform a task (Figure S23) or splitting the data collection into two separate days (Figure S26) also yielded highly overlapping optimal scan time range with the main analyses. Nevertheless, it is possible that some other (undiscovered) state manipulation could modify the 1% optimal scan time range significantly.

(R3C8) Fig 5 and 6 need more explanation about the graph; error bar, shaded area, etc.

We apologize for being unclear. We have updated Figure 5 captions as follows.

pg 14 (Figure 5 captions)

(C) Box plots showing goodness of fit to logarithmic and theoretical models before and after randomizing fMRI run order for 33 ABCD phenotypes. * indicates that goodness-of-fits were significantly different (after FDR correction with $q < 0.05$). (D) Same as

panel C, but for 42 HCP phenotypes. For all panels, model fit was performed using the maximum scan time per participant. For panels C and D, we considered all phenotypes whose prediction accuracies (Pearson's r) were positive in at least 90% of all combinations of N and T . For each boxplot, the horizontal line indicates the median across 33 ABCD phenotypes or 42 HCP phenotypes. The bottom and top edges of the box indicate the 25th and 75th percentiles, respectively. Outliers are defined as data points beyond 1.5 times the interquartile range. The whiskers extend to the most extreme data points not considered outliers.

For Figure 6, the original shaded area represented standard error across phenotypes. However, in the current manuscript, we decided to add a 1% optimal scan time range instead. Variability across phenotypes can be explored on the online web app, where users can select different phenotypes and/or phenotypic domains.

pg 17 (Figure 6 captions)

The error bars indicate the optimal range of scan time that can achieve within 1% of the maximum prediction accuracy.

(R3C9) In the first line and paragraph, the manuscript starts by mentioning brain structure and function. But this study (scan duration) is only about functional data. It might be helpful to explicitly clarify this first.

We thank the reviewer for this suggestion. We have modified the first paragraph of the introduction to only reference brain function.

pg 4 (Introduction)

A fundamental question in systems neuroscience is how individual differences in brain function are related to common variation in phenotypic traits, such as cognitive ability or physical health. Following recent work (Marek et al., 2022), we define brain wide association studies (BWAS) as studies of the associations between phenotypic traits and common inter-individual variability of the human brain.

(R3C10) The authors reported that the censoring did not substantially change their conclusion. I would like to ask for discussion about this observation.

We thank the reviewer for raising this point. Our interpretation is similar to what we have mentioned before (see our response to R3C7), which is that the primary effect of increasing scan time T is to increase FC reliability. Other factors (e.g., changing brain state by asking participants to perform a task, censoring, collecting resting-state fMRI on two separate sessions) can have a secondary (smaller) effect than the primary effect.

Another important point is that our analysis utilized a subset of participants whose data survived censoring, so this is a relatively "clean" set of participants. In the web application, we have provided an option for researchers to provide an estimate of the percentage of participants, whose data might be lost due to quality control or study drop out in general. This dropout will be factored into the study design calculations.

We have updated our manuscript as follows.

pg 19 (Discussion)

As a third example, our analysis was performed on participants whose data survived quality control. Therefore, we have also provided an option on the web application to allow researchers to specify their estimate of the percentage of participants, whose data might be lost due to poor data quality (or general drop out).

References

- Abraham, A., Milham, M. P., Di Martino, A., Craddock, R. C., Samaras, D., Thirion, B., & Varoquaux, G. (2017). Deriving reproducible biomarkers from multi-site resting-state data: An Autism-based example. *Neuroimage*, *147*, 736-745. <https://doi.org/https://doi.org/10.1016/j.neuroimage.2016.10.045>
- Benkarim, O., Paquola, C., Park, B.-y., Kebets, V., Hong, S.-J., Vos de Wael, R., Zhang, S., Yeo, B. T. T., Eickenberg, M., Ge, T., Poline, J.-B., Bernhardt, B. C., & Bzdok, D. (2022). Population heterogeneity in clinical cohorts affects the predictive accuracy of brain imaging. *PLOS Biology*, *20*(4), e3001627. <https://doi.org/10.1371/journal.pbio.3001627>
- Bijsterbosch, J., Harrison, S., Duff, E., Alfaro-Almagro, F., Woolrich, M., & Smith, S. (2017). Investigations into within- and between-subject resting-state amplitude variations. *Neuroimage*, *159*, 57-69. <https://doi.org/10.1016/j.neuroimage.2017.07.014>
- Boutet, A., Madhavan, R., Elias, G. J. B., Joel, S. E., Gramer, R., Ranjan, M., Paramanandam, V., Xu, D., Germann, J., Loh, A., Kalia, S. K., Hodaie, M., Li, B., Prasad, S., Coblenz, A., Munhoz, R. P., Ashe, J., Kucharczyk, W., Fasano, A., & Lozano, A. M. (2021). Predicting optimal deep brain stimulation parameters for Parkinson's disease using functional MRI and machine learning. *Nature Communications*, *12*(1), 3043. <https://doi.org/10.1038/s41467-021-23311-9>
- Braga, R. M., & Buckner, R. L. (2017). Parallel Interdigitated Distributed Networks within the Individual Estimated by Intrinsic Functional Connectivity. *Neuron*, *95*(2), 457-471.e455. <https://doi.org/10.1016/j.neuron.2017.06.038>
- Cash, R. F. H., Weigand, A., Zalesky, A., Siddiqi, S. H., Downar, J., Fitzgerald, P. B., & Fox, M. D. (2021). Using Brain Imaging to Improve Spatial Targeting of Transcranial Magnetic Stimulation for Depression. *Biol Psychiatry*, *90*(10), 689-700. <https://doi.org/10.1016/j.biopsych.2020.05.033>
- Chen, J., Ooi, L. Q. R., Tan, T. W. K., Zhang, S., Li, J., Asplund, C. L., Eickhoff, S. B., Bzdok, D., Holmes, A. J., & Yeo, B. T. T. (2023). Relationship between prediction accuracy and feature importance reliability: An empirical and theoretical study. *Neuroimage*, *274*, 120115. <https://doi.org/https://doi.org/10.1016/j.neuroimage.2023.120115>
- Chen, J., Tam, A., Kebets, V., Orban, C., Ooi, L. Q. R., Asplund, C. L., Marek, S., Dosenbach, N. U. F., Eickhoff, S. B., Bzdok, D., Holmes, A. J., & Yeo, B. T. T. (2022). Shared and unique brain network features predict cognitive, personality, and mental health scores in the ABCD study. *Nature Communications*, *13*(1), 2217. <https://doi.org/10.1038/s41467-022-29766-8>
- Chen, P., An, L., Wulan, N., Zhang, C., Zhang, S., Ooi, L. Q. R., Kong, R., Chen, J., Wu, J., Chopra, S., Bzdok, D., Eickhoff, S. B., Holmes, A. J., & Yeo, B. T. T. (2023). Multilayer meta-matching: translating phenotypic prediction models from multiple datasets to small data. *bioRxiv*. <https://doi.org/10.1101/2023.12.05.569848>

Chopra, S., Cocuzza, C. V., Lawhead, C., Ricard, J. A., Labache, L., Patrick, L. M., Kumar, P., Rubenstein, A., Moses, J., Chen, L., Blankenbaker, C., Gillis, B., Germine, L. T., Harpaz-Rote, I., Yeo, B. T. T., Baker, J. T., & Holmes, A. J. (2024). The Transdiagnostic Connectome Project: a richly phenotyped open dataset for advancing the study of brain-behavior relationships in psychiatry. *medRxiv*, 2024.2006.2018.24309054. <https://doi.org/10.1101/2024.06.18.24309054>

DuPre, E., Salo, T., Ahmed, Z., Bandettini, P., Bottenhorn, K., Caballero, C., Dowdle, L., Gonzalez-Castillo, J., Heunis, S., Kundu, P., Laird, A., Markello, R., Markiewicz, C., Moia, S., Staden, I., Teves, J., Uruñuela, E., Vaziri-Pashkam, M., Whitaker, K., & Handwerker, D. (2021). TE-dependent analysis of multi-echo fMRI with tedana. *Journal of Open Source Software*, 6, 3669. <https://doi.org/10.21105/joss.03669>

Fischl, B., Salat, D. H., Busa, E., Albert, M., Dieterich, M., Haselgrove, C., Van Der Kouwe, A., Killiany, R., Kennedy, D., Klaveness, S., Montillo, A., Makris, N., Rosen, B., & Dale, A. M. (2002). Whole Brain Segmentation. *Neuron*, 33(3), 341-355. [https://doi.org/10.1016/s0896-6273\(02\)00569-x](https://doi.org/10.1016/s0896-6273(02)00569-x)

Gell, M., Noble, S., Laumann, T. O., Nelson, S. M., & Tervo-Clemmens, B. (2024). Psychiatric neuroimaging designs for individualised, cohort, and population studies. *Neuropsychopharmacology*. <https://doi.org/10.1038/s41386-024-01918-y>

Gordon, E. M., Laumann, T. O., Gilmore, A. W., Newbold, D. J., Greene, D. J., Berg, J. J., Ortega, M., Hoyt-Drazen, C., Gratton, C., Sun, H., Hampton, J. M., Coalson, R. S., Nguyen, A. L., McDermott, K. B., Shimony, J. S., Snyder, A. Z., Schlaggar, B. L., Petersen, S. E., Nelson, S. M., & Dosenbach, N. U. F. (2017). Precision Functional Mapping of Individual Human Brains. *Neuron*, 95(4), 791-807.e797. <https://doi.org/10.1016/j.neuron.2017.07.011>

Greene, A. S., Gao, S., Scheinost, D., & Constable, R. T. (2018). Task-induced brain state manipulation improves prediction of individual traits. *Nature Communications*, 9(1). <https://doi.org/10.1038/s41467-018-04920-3>

Greene, A. S., Shen, X., Noble, S., Horien, C., Hahn, C. A., Arora, J., Tokoglu, F., Spann, M. N., Carrión, C. I., Barron, D. S., Sanacora, G., Srihari, V. H., Woods, S. W., Scheinost, D., & Constable, R. T. (2022). Brain-phenotype models fail for individuals who defy sample stereotypes. *Nature*, 609(7925), 109-118. <https://doi.org/10.1038/s41586-022-05118-w>

Greve, D. N., & Fischl, B. (2009). Accurate and robust brain image alignment using boundary-based registration. *Neuroimage*, 48(1), 63-72. <https://doi.org/https://doi.org/10.1016/j.neuroimage.2009.06.060>

He, T., Kong, R., Holmes, A. J., Nguyen, M., Sabuncu, M. R., Eickhoff, S. B., Bzdok, D., Feng, J., & Yeo, B. T. T. (2020). Deep neural networks and kernel regression achieve comparable accuracies for functional connectivity prediction of behavior and demographics. *Neuroimage*, 206, 116276. <https://doi.org/https://doi.org/10.1016/j.neuroimage.2019.116276>

Kopal, J., Uddin, L. Q., & Bzdok, D. (2023). The end game: respecting major sources of population diversity. *Nature Methods*. <https://doi.org/10.1038/s41592-023-01812-3>

Laumann, Timothy O., Gordon, Evan M., Adeyemo, B., Snyder, Abraham Z., Joo, Sung J., Chen, M.-Y., Gilmore, Adrian W., McDermott, Kathleen B., Nelson, Steven M., Dosenbach, Nico U. F., Schlaggar, Bradley L., Mumford, Jeanette A., Poldrack, Russell A., & Petersen, Steven E. (2015). Functional System and Areal Organization of a Highly Sampled Individual Human Brain. *Neuron*, 87(3), 657-670.
<https://doi.org/https://doi.org/10.1016/j.neuron.2015.06.037>

Laumann, T. O., Snyder, A. Z., Mitra, A., Gordon, E. M., Gratton, C., Adeyemo, B., Gilmore, A. W., Nelson, S. M., Berg, J. J., Greene, D. J., McCarthy, J. E., Tagliazucchi, E., Laufs, H., Schlaggar, B. L., Dosenbach, N. U. F., & Petersen, S. E. (2017). On the Stability of BOLD fMRI Correlations. *Cerebral Cortex*, 27(10), 4719-4732. <https://doi.org/10.1093/cercor/bhw265>

Li, J., Bzdok, D., Chen, J., Tam, A., Ooi, L. Q. R., Holmes, A. J., Ge, T., Patil, K. R., Jabbi, M., Eickhoff, S. B., Yeo, B. T. T., & Genon, S. (2022). Cross-ethnicity/race generalization failure of behavioral prediction from resting-state functional connectivity. *Sci Adv*, 8(11), eabj1812.
<https://doi.org/10.1126/sciadv.abj1812>

Lynch, C. J., Elbau, I. G., Ng, T. H., Wolk, D., Zhu, S., Ayaz, A., Power, J. D., Zebley, B., Gunning, F. M., & Liston, C. (2022). Automated optimization of TMS coil placement for personalized functional network engagement. *Neuron*, 110(20), 3263-3277.e3264.
<https://doi.org/10.1016/j.neuron.2022.08.012>

Marek, S., Tervo-Clemmens, B., Calabro, F. J., Montez, D. F., Kay, B. P., Hatoum, A. S., Donohue, M. R., Foran, W., Miller, R. L., Hendrickson, T. J., Malone, S. M., Kandala, S., Feczko, E., Miranda-Dominguez, O., Graham, A. M., Earl, E. A., Perrone, A. J., Cordova, M., Doyle, O., Moore, L. A., Conan, G. M., Uriarte, J., Snider, K., Lynch, B. J., Wilgenbusch, J. C., Pengo, T., Tam, A., Chen, J., Newbold, D. J., Zheng, A., Seider, N. A., Van, A. N., Metoki, A., Chauvin, R. J., Laumann, T. O., Greene, D. J., Petersen, S. E., Garavan, H., Thompson, W. K., Nichols, T. E., Yeo, B. T. T., Barch, D. M., Luna, B., Fair, D. A., & Dosenbach, N. U. F. (2022). Reproducible brain-wide association studies require thousands of individuals. *Nature*, 603(7902), 654-660. <https://doi.org/10.1038/s41586-022-04492-9>

Nichols, T. E., Das, S., Eickhoff, S. B., Evans, A. C., Glatard, T., Hanke, M., Kriegeskorte, N., Milham, M. P., Poldrack, R. A., Poline, J.-B., Proal, E., Thirion, B., Van Essen, D. C., White, T., & Yeo, B. T. T. (2017). Best practices in data analysis and sharing in neuroimaging using MRI. *Nature Neuroscience*, 20(3), 299-303. <https://doi.org/10.1038/nn.4500>

Nozari, E., Bertolero, M. A., Stiso, J., Caciagli, L., Cornblath, E. J., He, X., Mahadevan, A. S., Pappas, G. J., & Bassett, D. S. (2024). Macroscopic resting-state brain dynamics are best described by linear models. *Nature Biomedical Engineering*, 8(1), 68-84.
<https://doi.org/10.1038/s41551-023-01117-y>

Ooi, L. Q. R., Chen, J., Zhang, S., Kong, R., Tam, A., Li, J., Dhamala, E., Zhou, J. H., Holmes, A. J., & Yeo, B. T. T. (2022). Comparison of individualized behavioral predictions across anatomical, diffusion and functional connectivity MRI. *Neuroimage*, 263, 119636.
<https://doi.org/https://doi.org/10.1016/j.neuroimage.2022.119636>

- Orban, C., Kong, R., Li, J., Chee, M. W. L., & Yeo, B. T. T. (2020). Time of day is associated with paradoxical reductions in global signal fluctuation and functional connectivity. *PLoS Biol*, *18*(2), e3000602. <https://doi.org/10.1371/journal.pbio.3000602>
- Power, J. D., Mitra, A., Laumann, T. O., Snyder, A. Z., Schlaggar, B. L., & Petersen, S. E. (2014). Methods to detect, characterize, and remove motion artifact in resting state fMRI. *Neuroimage*, *84*, 320-341. <https://doi.org/10.1016/j.neuroimage.2013.08.048>
- Schaefer, A., Kong, R., Gordon, E. M., Laumann, T. O., Zuo, X.-N., Holmes, A. J., Eickhoff, S. B., & Yeo, B. T. T. (2018). Local-Global Parcellation of the Human Cerebral Cortex from Intrinsic Functional Connectivity MRI. *Cerebral Cortex*, *28*(9), 3095-3114. <https://doi.org/10.1093/cercor/bhx179>
- Tagliazucchi, E., & Laufs, H. (2014). Decoding Wakefulness Levels from Typical fMRI Resting-State Data Reveals Reliable Drifts between Wakefulness and Sleep. *Neuron*, *82*(3), 695-708. <https://doi.org/https://doi.org/10.1016/j.neuron.2014.03.020>
- Varoquaux, G. (2018). Cross-validation failure: Small sample sizes lead to large error bars. *Neuroimage*, *180*(Pt A), 68-77. <https://doi.org/10.1016/j.neuroimage.2017.06.061>
- Wang, C., Ong, J. L., Patanaik, A., Zhou, J., & Chee, M. W. L. (2016). Spontaneous eyelid closures link vigilance fluctuation with fMRI dynamic connectivity states. *Proceedings of the National Academy of Sciences*, *113*(34), 9653-9658. <https://doi.org/10.1073/pnas.1523980113>
- Wu, J., Li, J., Eickhoff, S. B., Scheinost, D., & Genon, S. (2023). The challenges and prospects of brain-based prediction of behaviour. *Nature Human Behaviour*, *7*(8), 1255-1264. <https://doi.org/10.1038/s41562-023-01670-1>
- Zhao, W., Makowski, C., Hagler, D. J., Garavan, H. P., Thompson, W. K., Greene, D. J., Jernigan, T. L., & Dale, A. M. (2023). Task fMRI paradigms may capture more behaviorally relevant information than resting-state functional connectivity. *Neuroimage*, *270*, 119946. <https://doi.org/10.1016/j.neuroimage.2023.119946>

Response to reviewers 2023-06-10250B “MRI economics: Balancing sample size and scan duration in brain wide association studies”

We thank the reviewers and editor for their insightful feedback. We are pleased by the reviewers’ positive comments. There were several residual concerns, which we have thoroughly addressed in the re-revised manuscript. These revisions have significantly improved the manuscript. Our responses are in blue. Reviewer and editor comments are in italics. Changes to the manuscript are quoted verbatim in normal black font.

Editor:

(EIC1) As I believe we discussed when we met at OHBM in Montreal, we see the impact of your paper as a practical one: It has the potential to provide actionable insights that could increase the robustness and efficiency of BWAS studies. We therefore share RI's concerns that the tradeoffs and implications are not sufficiently explicit and therefore cannot provide (a) take away message(s) that could impact the community in a sufficiently substantial way to warrant further consideration in Nature. As you have already done so many additional analyses for this revision, we do not feel that we can, in good faith, invite another revision and therefore cannot consider this paper further. If, however, you feel that this is something you can address, we could potentially consider an appeal on our decision.

We agree with you wholeheartedly that highlighting actionable insights could strengthen the manuscript. Keeping in mind your suggestions during our discussion at OHBM, we have gone through our results to distill the following three key messages:

- (1) Many BWAS consortia collect ≤ 10 min fMRI per participant, which is highly cost inefficient
- (2) To maximize efficiency, optimum scan time is ≥ 20 min in the vast majority of BWAS.
- (3) 30-min scans are the most cost-effective (on average across nine resting-state and task-state fMRI datasets), yielding 22% cost savings over 10-min scans.

Further elaborations of these key messages and manuscript changes are found below.

Referee #1:

(RIC1) I appreciate the added work the authors have put into this revision. I’m still of the opinion that this work is interesting, and I found that the analyses on the new datasets are an important addition. However, I’m still unconvinced about the overall message and its impact, as explained further below: In general, I found the new dependence on 1% error bars to largely muddy the underlying issues at play in this work, leading to a lack of clarity on the findings. While I understand the desire to provide them, in practice, they lead to very large discrepancies in the “amount of data needed” (e.g., in Figure S24.1, anywhere from 10 to 120 minutes in the ADNI dataset). This is even worse in several other comparisons. If this is truly the best guess provided by the current approach, this severely weakens the impact of the current manuscript, as it provides little practical guidance on the optimal amount of data to collect for future

researchers. Moreover, these large error bars allow the authors to ignore the differences that cause certain measures/datasets to be acceptable with relatively less (10 minutes) vs. more (30 minutes) data per person. Understanding the differences leading to these variations in the “minimum acceptable data” per participant seems critical to extrapolating conclusions and next directions from the current work, and to making it more useful for the broader community.

Thank you for your insightful comments. We absolutely agree with you that the broad 1% error bars limited practical guidance for future studies. We have removed the 1% error bars and moved the figure to Extended Data Figure 2. We have created a new figure (below), which provides more direct guidance for researchers.

To explain the new figure, recall that we have previously fitted the theoretical model to 76 phenotypes across nine datasets comprising six resting-fMRI datasets and three ABCD task-fMRI datasets. For each phenotype, the fitted model was normalized by its maximum achievable accuracy (estimated by the theoretical model), yielding a fraction of maximum achievable prediction accuracy for every combination of sample size and scan time per participant. The fraction of maximum achievable prediction accuracy was then averaged across all phenotypes yielding Figure 6a below (same as Figure 6a in the previous submission).

Figure 6a (below) allows us to optimize scan time to minimize study cost based on a fixed accuracy target. For example, suppose we want to achieve 90% of maximum achievable accuracy, we can find all pairs of sample size and scan time per participant along the black contour line corresponding to 0.9 in Figure 6a. For every pair of sample size and scan time, we can then compute the study cost given a particular scan cost per hour (e.g., \$500) and a particular overhead cost per participant (e.g., \$500). The optimal scan time (and sample size) leading to the lowest study cost across the nine datasets can then be obtained.

Here, we considered 3 possible accuracy targets (80%, 90% or 95% of maximum accuracy), 2 possible overhead costs (\$500 or \$1000 per participant) and 2 possible scan costs per hour (\$500 or \$1000). In total there were $3 \times 2 \times 2 = 12$ conditions. Since there were nine datasets, this resulted in $12 \times 9 = 108$ scenarios. In the vast majority (85%) of these 108 scenarios, the optimal scan time was at least ≥ 20 min (Figure 6b).

However, for future study design, the optimal scan time (Figure 6b) is not known in advance, so we also aimed to derive a *fixed* scan time that is the most cost-effective in most situations. Fig. 6c shows the normalized cost inefficiency of various fixed scan times relative to the optimal scan time for each of 108 scenarios. Many consortia BNAS collect 10-min fMRI scans, which is highly cost inefficient. On average across resting and task states, 30-min scans were the most cost-effective, yielding 22% cost savings over 10-min scans (Figure 6d).

These findings were distilled across six resting-fMRI and three task-fMRI datasets, ensuring they reflect the variability across datasets that you have highlighted. Please also see our response to R1C2, which discusses further analyses to explore this variability in more depth. We have updated the manuscript as follows.

Fig. 6 | 30-min scans yield significant cost savings over 10-min scans across nine datasets. a. Fraction of maximum prediction accuracy as a function of sample size and scan time per participant averaged across 76 phenotypes from nine datasets (six resting-fMRI datasets and three ABCD task-fMRI datasets). Here sample size was calculated assuming 90% of participants were used for training the model, and 10% of participants were used for model evaluation (i.e., 10-fold cross-validation). **b.** Histogram of optimal scan time (with regards to costs) across 108 scenarios. Given 3 possible accuracy targets (80%, 90% or 95% of maximum achievable accuracy), 2 possible overhead costs (\$500 or \$1000 per participant) and 2 possible scan costs per hour (\$500 or \$1000), there are $3 \times 2 \times 2 = 12$ conditions. In total, we have $9 \text{ datasets} \times 12 \text{ conditions} = 108$ scenarios. For 85% of the scenarios, the optimal scan time (in terms of cost) was at least 20 min (red dashed line). **c.** Normalized cost inefficiency (across the 108 scenarios) as a function of a fixed scan time per participant, relative to the optimal scan time in (b). We note that in practice, the optimal scan time in (b) is not known in advance, so this plot seeks to derive a **fixed** optimal scan time generalizable to most situations. Each boxplot contains 108 data points (corresponding to the 108 scenarios). For visualization, the boxplots are normalized by subtracting the cost inefficiency of the best possible fixed scan time (30 min in this case), so that the normalized cost inefficiency of the best possible fixed scan time is centered at zero. **d.** Cost savings relative to 10 min of scan time per participant. The greatest cost saving (22%) was achieved at 30 min.

pg 3 (Abstract)

A pervasive dilemma in brain-wide association studies (BWAS) is whether to prioritize functional MRI (fMRI) scan time or sample size. We derive a theoretical model showing

that individual-level phenotypic prediction accuracy increases with sample size and total scan duration (sample size \times scan time per participant). The model explains empirical prediction accuracies extremely well across 76 phenotypes from nine resting-fMRI and task-fMRI datasets ($R^2 = 0.89$), spanning a wide range of scanners, acquisitions, racial groups, disorders and ages. For scans ≤ 20 mins, prediction accuracy increases linearly with the logarithm of total scan duration, suggesting interchangeability of sample size and scan time. However, sample size is ultimately more important than scan time in determining prediction accuracy. Nevertheless, when accounting for overhead costs associated with each participant (e.g., recruitment costs), to boost prediction accuracy, longer scans can yield substantial cost savings over larger sample size. To achieve high prediction performance, 10-min scans are highly cost inefficient. In most scenarios, the optimal scan time is ≥ 20 mins. On average, 30-min scans are the most cost-effective, yielding 22% cost savings over 10-min scans. Overshooting is cheaper than undershooting the optimal scan time, so we recommend aiming for ≥ 30 mins. Compared with resting-state whole-brain BWAS, the most cost-effective scan time is shorter for task-fMRI and longer for subcortical-cortical BWAS. Standard power calculations maximize sample size at the expense of scan time. Our study demonstrates that optimizing both sample size and scan time can boost prediction power while cutting costs. Our empirically informed reference is available for future study planning: WEB_APPLICATION_LINK

pg 15 (Results)

30-min scans are considerably cheaper than 10-min scans

Beyond optimizing scan time to maximize prediction accuracy within a fixed scan budget (previous section), the model fits shown in Fig. 6a can also be used to optimize scan time to minimize study cost to achieve a fixed accuracy target. For example, suppose we want to achieve 90% of maximum achievable accuracy, we can find all pairs of sample size and scan time per participant along the black contour line corresponding to 0.9 in Fig. 6a. For every pair of sample size and scan time, we can then compute the study cost given a particular scan cost per hour (e.g., \$500) and a particular overhead cost per participant (e.g., \$1000). The optimal scan time (and sample size) with the lowest study cost can then be obtained.

Here, we considered 3 possible accuracy targets (80%, 90% or 95% of maximum accuracy), 2 possible overhead costs (\$500 or \$1000 per participant) and 2 possible scan costs per hour (\$500 or \$1000). In total there were $3 \times 2 \times 2 = 12$ conditions. Since there were nine datasets, this resulted in $12 \times 9 = 108$ scenarios. In the vast majority (85%) of these 108 scenarios, the optimal scan time was at least ≥ 20 min (Fig. 6b).

However, during study design, the optimal scan time (Fig. 6b) is not known in advance, therefore we also seek to derive a *fixed* scan time that is the most cost-effective in most situations. Fig. 6c shows the normalized cost inefficiency of various fixed scan times relative to the optimal scan time for each of 108 scenarios. Many consortia BWAS collect 10-min fMRI scans, which is highly cost inefficient. On average across resting and task states, 30-min scans were the most cost-effective, yielding 22% cost savings over 10-min

scans (Fig. 6d). We again note the asymmetry in the cost curves, so it is cheaper to overshoot than undershoot the most cost-effective scan time.

pg 34 (Methods)

Optimizing sample size and scan time to achieve a fixed prediction accuracy

To generate Fig. 6b, 6c, 7 and 8a, suppose we want to achieve 90% of maximum achievable accuracy, we can find all pairs of sample size and scan time per participant along the 0.9 black contour line in Fig. 6a. For every pair of sample size and scan time, we can then compute the study cost given a particular scan cost per hour (e.g., \$500) and a particular overhead cost per participant (e.g., \$1000). The optimal scan time (and sample size) with the lowest study cost can then be obtained.

(RIC2) With this in mind, while the authors have done a lot for this response in terms of adding new datasets, many of the specific questions and suggestions I (and the other reviewers made) made were not very well addressed (e.g., the specific dependency on SNR, resolution, association with test-retest reliability, hyperparameter selection). While new datasets were added, it wasn't clear to me that there was a substantial investigation of how scan parameters affected the results, aside from the description that the error bars overlapped (and, given the size of the error bars as noted above, this seems like a not very useful conclusion). When differences were found, they were relatively skirted over (e.g., the differences between subcortical and cortical connectivity, which were never returned to in the discussion section).

We thank you for re-raising these important points. In this revision, we have added new analyses and figures to specifically explore the effects of fMRI SNR (Figure 8b), resolution (Figure 8c, Extended Data Figures 3d & 3e), phenotypic test-retest reliability (Extended Data Figure 3c), subcortical connectivity (Figure 8a) and other factors. Key results (e.g., resting-state versus task-state, as well as whole-brain versus subcortical-cortical BWAS) are now prominently discussed in the study, including in the abstract.

To contextualize these additional analyses and results, recall that in our previous response (RIC1), we have shown that 30-min scans are the most cost-effective on average across nine resting-state and task-state datasets. Looking at individual resting-state datasets (Figure 7a) and task states (Figure 7b), we did observe substantial variability in the most cost-effective scan time. However, despite this variability, the most cost-effective scan time was at least 20 minutes in all datasets (Figure 7). In the ABCD dataset, there were notable differences between brain states, as the most cost-effective scan time was significantly shorter in all task-FC than resting-FC. Among the three tasks, the most cost-effective scan time was the shortest for the n-back task (25 min). However, even for the n-back task, 30-min scans led to only 0.9% higher cost (relative to 25 min), compared with 16.1% higher cost for 10-min scans. Indeed, all cost inefficiency curves exhibit an asymmetric pattern, consistent with the grand average we reported in Figures 6c and 6d. This asymmetry suggests that for future studies dealing with unknown unknowns, overshooting (rather than undershooting) a general target of 30 minutes might be preferable for minimizing overall study costs.

We have updated the manuscript as follows.

pg 3 (Abstract)

Compared with resting-state whole-brain BWAS, the most cost-effective scan time is shorter for task-fMRI and longer for subcortical-cortical BWAS.

pg 17 (Results)

Most cost-effective scan time for task-fMRI is shorter than for resting-fMRI

Across the six resting-state datasets (Fig. 7a), the most cost-effective scan time was the longest for ABCD (60 min) and shortest for the TCP and ADNI datasets (20 min). However, a scan time of 30 minutes was still relatively cost-effective for all datasets, because of a flat cost curve near the optimum and the asymmetry of the cost curve. For example, even for the TCP dataset, which had the shortest most cost-effective scan time of 20 min, over-scanning with 30-min scans led to only 3.7% higher cost relative to 20 min, compared with 7.3% higher cost for under-scanning 10-min scans.

Fig. 7 | Variation in the most cost-effective scan time across resting-state and task-state fMRI. **a.** Cost inefficiency as a function of scan time per participant for the six resting-state datasets. This plot provides the same information as Fig. 6c but shown for each dataset separately. Numbers in brackets indicate number of phenotypes. **b.** Cost inefficiency for ABCD resting-state and task-state fMRI. **c.** Cost inefficiency if data was collected in two separate sessions (based on the HCP dataset). Similar to Fig. 6c, for visualization, each curve is normalized by subtracting the cost inefficiency of the best possible fixed scan time (of each curve), so that the normalized cost inefficiency of the best possible fixed scan time is centered at zero.

Previous studies have shown that task-FC leads to better prediction performance for cognitive measures (Greene et al., 2018; Chen et al., 2022). Here, we extend previous results, finding that the most cost-effective scan time was significantly shorter in ABCD task-fMRI than ABCD resting-fMRI (Fig. 7b). Among the three tasks, the most cost-effective scan time was the shortest for the n-back task (25 min). However, even for the n-back task, 30-min scans led to only 0.9% higher cost (relative to 25 min), compared with 16.1% higher cost for 10-min scans.

These task results suggest that the most cost-effective scan time is sensitive to brain state manipulation. Task-based fMRI may preferentially engage cognitive and physiological mechanisms that are closely tied to the expression of specific phenotypes (e.g., processing speed), thereby enhancing the specificity of functional connectivity estimates for phenotypic prediction. Tasks may also facilitate shorter, more efficient scan durations by aligning brain states across individuals in a controlled manner, thereby reducing spurious non-stationary influences that could otherwise obscure reliable modeling of inter-individual differences. This alignment might be better achieved in tasks that present stimuli and conditions with identical timing across participants – whether using event-related or block designs.

Non-stationarity may also be potentially increased by distributing resting state fMRI runs across multiple sessions. Since the HCP dataset was collected on two different days (sessions), we were also able to directly compare the effect of a two-session versus a one-session design, controlling for total scan time. The most cost-effective scan time for the two-session design was only slightly longer than the original HCP analysis: 40 min vs 30 min (Fig. 7c).

Overall, these results suggest that state manipulation can influence the most cost-effective scan time, and that a relatively large state manipulation (e.g., task) can significantly influence cost-effectiveness.

In addition to exploring resting-state versus task-state, we have also followed your suggestion to explore phenotypic variability. The most cost-effective scan time of 30 minutes was generalizable across phenotypic domains (Extended Data Figure 3a). There was also not an obvious relationship between optimal scan time and phenotypic prediction accuracy (Extended Data Fig. 3b), phenotypic test-retest reliability (Extended Data Fig. 3c), temporal resolution (Extended Data Fig. 3d), voxel resolution (Extended Data Fig. 3e) or scan sequences (Extended Data Fig. 3f). We have updated the manuscript as follows

pg 16 (Results)

Variation across phenotypic domains and scan parameters

There were clear variations across phenotypes. For example, there were phenotypes that could be predicted very well and demonstrated prediction gains up to the maximum amount of data per participant (e.g., age in the ADNI dataset; Supplementary Fig. 17). However, there were also other phenotypes, which were predicted less well (e.g., BMI in the SINGER dataset; Supplementary Fig. 14), but showed prediction gains up to the maximum amount of data per participant. Since single phenotypes are not easily interpreted, we grouped the phenotypes into seven phenotypic domains to study phenotypic variation in more details.

For five of the seven phenotypic domains, the most cost-effective scan times ranged from 25 min to 40 min (Extended Data Fig. 3a). The most cost-effective scan time for emotion domain was very long, but this outlier was driven by a single phenotypic measure, so should not be over-interpreted. For the positron emission tomography (PET) phenotypic domain, our original scenarios assumed overhead costs of \$500 or \$1000 per participant,

which was unrealistic. Assuming a more realistic overhead PET cost per participant (\$5000 or \$10,000) yielded 50 min as the most cost-effective scan time.

Although there was a strong relationship between phenotypic prediction accuracy and goodness-of-fit to the theoretical model (Fig. 4), we did not find an obvious relationship between phenotypic prediction accuracy and optimal scan time (Extended Data Fig. 3b). Recent studies have also demonstrated that phenotypic reliability is important for BWAS power (Nikolaidis et al., 2022; Gell et al., 2023). In our theoretical model, phenotypic reliability directly impacts overall prediction accuracy but does not directly contribute to the trade-off between sample size and scan time. Indeed, there was not an obvious relationship between phenotypic test-retest reliability and optimal scan time (Extended Data Fig. 3c).

There was also not an obvious relationship of optimal scan time with temporal resolution, voxel resolution or scan sequence (Extended Data Figs. 3d to 3f). We emphasize that we are not claiming that scan parameters do not matter, but that other variations between datasets (e.g., phenotypes, populations) might exert a greater impact than common variation in scan parameters.

Consistent with the previous sections, we note that for the vast majority of phenotypes and scan parameters, the optimal scan time was at least 20 min, and on average the most cost-effective scan time was 30 min (Fig. 6c).

Extended Data Fig. 3 | Variation in cost inefficiency and optimal scan time across phenotypic domains and common scan parameters. **a.** Cost inefficiency as a function of scan time for various phenotypic domains across nine resting-fMRI and task-fMRI datasets. For the purpose of this plot, the positron emission tomography (PET) curve was based on a more realistic overhead cost of \$5000 or \$10000 per participant (instead of \$500 or \$1000) that was used for other phenotypic measures. Arrows indicate scan times with the lowest budgets. **b.** Optimal scan time as a function of phenotypic prediction accuracies. This analysis was obtained by sorting the maximum prediction accuracies (based on resting-state FC) of 19 HCP and 17 ABCD phenotypes into three bins. **c.** Optimal scan time as a function of phenotypic test-retest reliability. This analysis was obtained by considering 41 participants from the HCP, where the same phenotypic measures were collected twice (several months apart), allowing us to compute the test-retest reliability of the HCP phenotypes. We sorted the phenotypic test-retest reliability of the 19 HCP phenotypes into three bins. **d.** Optimal scan time as a function of repetition time (TR). **e.** Optimal scan time as a function of voxel size. **f.** Optimal scan time as a function of MRI acquisition. SE-SB: single-echo single-band; SE-MB: single-echo multi-band; ME-MB: multi-echo multi-band. We note that panels (b) to (f) only considered resting-state FC. Furthermore, the ADNI dataset was excluded from the analysis in panels (d), (e) and (f) because it included both single-band and multi-band data with different TRs and voxel sizes. Similar to Fig. 6c, for visualization, the curves in panel (a) are normalized by subtracting the cost inefficiency of the

best possible fixed scan time (of each curve), so that the best possible fixed scan time is centered at zero.

Using our new framework of minimizing scan costs to meet an accuracy target (Figure 6), we have re-analyzed the subcortical-cortical BWAS, and we found that the most cost-effective scan time for subcortical-cortical FC was roughly double that of whole-brain FC (Figure 8a). Additional simulations suggest that this might result from lower SNR (Figure 8b) and/or fewer FC features used for prediction (Figure 8c). We have updated the manuscript as follows.

pg 17 (Results)

Most cost-effective scan time for subcortical BWAS is longer than whole-brain BWAS

Our main analyses involved a cortical parcellation with 400 regions and 19 subcortical regions, yielding 419×419 RSFC matrices. We also repeated the analyses using 19×419 subcortical-cortical RSFC matrices. The most cost-effective scan time for subcortical-cortical RSFC was about double that of whole-brain RSFC (Fig. 8a). This might arise because of lower signal-to-noise ratio (SNR) in subcortical regions, resulting in the need for longer scan time to achieve better estimate of subcortical FC.

Fig. 8 | Most cost-effective scan time for subcortical BWAS is longer than whole-brain BWAS. **a.** Cost inefficiency as a function of scan time per participant with subcortical-cortical FC versus whole-brain FC. For visualization, similar to Fig. 6c, the curves are normalized by subtracting the cost inefficiency of the best possible fixed scan time (of each curve), so that the normalized cost inefficiency of the best possible fixed scan time is centered at zero. Numbers in brackets indicate number of phenotypes in each condition. **b.** Optimal scan time for predicting the cognitive factor score as a function of simulated Gaussian noise with standard deviation σ . **c.** Optimal scan time for predicting the cognitive factor score as a function of the resolution of cortical parcellation.

To explore the effects of fMRI SNR, for each parcel time course, we z-normalized the fMRI time course, so the resulting standard deviation of the time course was equal to one. We then added zero mean Gaussian noise with standard deviation σ . Even doubling the noise ($\sigma = 1$) had very little impact on the optimal scan time (Fig. 8b). As a sanity check,

we added a large quantity of noise ($\sigma = 3$), which lead to a much longer optimal scan time. (Fig. 8b).

Intuitively, this is not surprising because lower SNR means that a longer scan time is necessary to get an accurate estimate of individual-level FC. What is interesting is that a large SNR change is necessary to make a noticeable difference in optimal scan time, which might explain the robustness of optimal scan times across the common scan parameters we explored in the previous section (Extended Data Fig. 3). Therefore, even with small to moderate technological improvements in SNR, the most cost-effective scan time is unlikely to deviate from our estimate. Major increase in SNR could however shorten the most cost-effective scan time from the current estimates.

We also varied the resolution of the cortical parcellation with 200, 400, 600, 800 or 1000 parcels for predicting the cognitive factor scores in the HCP and ABCD datasets. There was a weak trend in which higher parcellation resolution led to slightly lower optimal scan time, although there was a big drop in optimal scan time from 200 parcels to 400 parcels in the ABCD dataset (Fig. 8c). Given that subcortical-cortical FC have less edges (features) than whole-brain FC, this could be another reason why subcortical-cortical FC requires longer optimal scan time.

Overall, our results suggest that 10-min scans are highly inefficient and 30-min scans are on average the most cost-effective. These results apply to the vast majority of individual-level BWAS in the existing literature (Figure 6). The new Figure 7, Figure 8 and Extended Data Figure 3 also provide further insights into potential deviation from the take home messages, thus providing extrapolations of the take home messages to other situations, including task-FC, multi-session data collection, subcortical-cortical FC, fMRI SNR and parcellation resolutions. We are grateful for your suggestions to undertake these additional analyses as we believe they bolster confidence that the take-home messages are broadly applicable across a range of scenarios that might differ across studies.

We have updated the manuscript as follows.

pg 18 (Results)

Longer scans improve prediction accuracy while reducing costs

To summarize, 30-min scans are on average the most cost-effective across resting-state and task-state whole-brain BWAS (Fig. 6c). The cost curves are also asymmetric, so it is cheaper to overshoot than undershoot the optimum. Therefore, even when the most-effective scan time is shorter than 30 min (e.g., n-back task or TCP dataset), 30-min scans only incurred a small penalty, relative to clairvoyantly knowing the true optimal scan time in advance. Furthermore, for subcortical BWAS, the most cost-effective scans are much longer than 30 mins.

Our results present a compelling case for moving beyond traditional power analyses, whose only inputs are sample size, to inform BWAS design. Such power analyses can only point towards maximizing sample size, so scan time becomes implicitly minimized under budget constraints. Our findings show that we can achieve higher prediction

performance by increasing both sample size and scan time, while generating substantial cost-savings compared to only increasing sample size alone.

Our results complement recent advocacy for larger sample sizes to increase BWAS reproducibility (Marek et al., 2022). Consistent with previous studies (Varoquaux, 2018), when sample size is small, there is a high degree of variability across cross-validation folds (Fig. 3a). Furthermore, large sample sizes are still necessary for high prediction accuracy. To achieve 80% of maximum prediction accuracy with 30-min scans, a sample size of ~900 is necessary (Fig. 6a), which is much larger than typical BWAS (Marek et al., 2022). To achieve 90% of maximum prediction accuracy with 30-min scans, a sample size of ~2500 is necessary (Fig. 6a).

In addition to increasing sample size and scan time, BWAS effect sizes can also be enhanced through innovative study designs. Recent work showed that U-shaped population sampling can enhance the strength of associations between functional connectivity and phenotypic measures (Kang et al., 2024). However, more complex screening procedures will increase overhead cost per participant, which might lengthen the optimal scan time.

The current analysis was focused on high target accuracies (80%, 90% or 95%) and relatively low overhead costs (\$500 or \$1000). Lower target accuracies (in smaller-scale studies) and higher overhead costs (e.g., PET, multi-site data collection) will lead to longer cost-effective scan time (Extended Data Fig. 2). In practice, scans are also more likely to be spuriously shortened (e.g., due to participant discomfort) than to be spuriously extended. Therefore, we recommend scan time to be ≥ 30 min.

Finally, with regards to the reviewer's point about fixing the hyperparameter, it is widely accepted that hyperparameter tuning is very important in machine learning. For example, Ferrer and Hutter (2019) noted that "it is also widely acknowledged that tuned hyperparameters improve over the default setting provided by common machine learning libraries." In another influential paper, Yang and Shami (2020) stated that "[t]o fit a machine learning model into different problems, its hyper-parameters must be tuned. Selecting the best hyperparameter configuration for machine learning models has a direct impact on the model's performance." Therefore, in keeping with established best practices, we believe that the use of hyperparameter tuning is appropriate.

References

Feurer M, Hutter F. Hyperparameter optimization. Automated machine learning: Methods, systems, challenges. 2019:3-3.

Yang L, Shami A. On hyperparameter optimization of machine learning algorithms: Theory and practice. Neurocomputing. 2020 Nov 20;415:295-316.

(RIC3) I was also surprised, given the focus of the article, that with the addition of many new datasets, very little attention was given to how much data was present per person in the new datasets. This seems like information that should be prominently included in the primary text figures. Were any able to sufficiently test the maximal scan time per participant to validate the

findings from the model shown in Fig. 6? If no, this seems like an important limitation that should be noted in the paper, and weakens the conclusions that can be made.

We thank you for raising this point. We have included a new Table 1 showing the sample size and scan time per participant for the 6 resting-state datasets and 3 ABCD task datasets. Further scan details are found in Supplementary Table 3.

Table 1 | Sample size and amount of scan time per participant in each dataset.

Dataset	Sample Size (N)	Scan Time (T)	Remarks
HCP-REST	792	57m 36s	Two sessions; two runs per session
ABCD-REST	2565	20m	Four fMRI runs in a single session
SINGER-REST	642	9m 56s	One fMRI run in a single session
TCP-REST	194	26m 2s	Four fMRI runs in a single session
MDD-REST	287	23m 12s	Four fMRI runs in a single session
ADNI-REST	586	9m	One fMRI run in a single session
ABCD-MID	2262	10m 44s	Two fMRI runs in a single session
ABCD-NBACK	2262	9m 39s	Two fMRI runs in a single session
ABCD-SST	2262	12m 39s	Two fMRI runs in a single session

With regards to the reviewer’s comments about whether there is sufficient scan time per participant to validate our original Figure 6, we note that the study’s take home messages have been updated (see our response to E1C1 and R1C1).

Recall that the first take-home message is that 10 minutes of BWAS is never cost efficient, while the second take-home message is that the optimum scan time is ≥ 20 min in most BWAS scenarios. Four of the nine datasets have at least 20 min of fMRI per participant, so we believe that we have sufficient data to support the first two take home messages without any caveat.

With respect to the third take-home message, which is the recommendation that studies should aim for at least 30 min of scan time, only one dataset (HCP) has more than 30 min of scan time per participant. As such, the revised manuscript will take note of this caveat.

However, we also want to emphasize that we were unable to find another dataset with at least 30 minutes of resting-state fMRI with relatively large sample sizes to be used for a BWAS. Therefore, the caveat about the third take-home message also underscores the importance of our study, which is to highlight the fact that most BWAS studies can achieve significant budgetary savings with longer scan time per participant. We have updated the manuscript as follows.

pg. 19 (Results)

Overall, 10-min scans are rarely cost-effective, and optimum scan time is ≥ 20 mins in most BWAS (Fig. 6c). Among the datasets we analyzed, four included scans ≥ 20 mins, providing robust evidence to support this conclusion across multiple datasets. On the other hand, we could only identify one dataset (HCP) with scans exceeding 30 minutes and a sufficiently large sample size for inclusion in our study (Table 1). This limitation underscores the importance of our findings, emphasizing the need for BWAS to prioritize longer scans.

(RIC4) I still think it's notable that many of the measures continue to show more gains with more data per participant, for the full extent of data collected (Supp. Figs), especially for high prediction measures like Age. At the least, this point seems like one that merits additional discussion within the manuscript. Could it be that other limitations (e.g., the accuracy of behavioral metrics) limit the potential per participant gains seen with other measures?

We agree with you that there are phenotypes, which can be predicted very well (e.g., age in the ADNI dataset; Figure S20), and continue to show prediction gains up to the maximum amount of data per participant. However, we note that there are also other phenotypes, which are predicted less well (e.g., BMI in the SINGER dataset; Figure S17), but continue to show prediction gains up to the maximum amount of data per participant. We now mention the examples you raise in the manuscript. See our response to R1C2.

To study this more systematically, we also plotted optimal scan time as a function of phenotypic prediction accuracy and phenotypic test-retest reliability (see Extended Data Fig. 3 in our response to R1C2). We found no obvious relationship between optimal scan time and phenotypic prediction accuracy or test-retest reliability. We emphasize that we are not saying that there is no variation across phenotypes, but the factors underlying this variation are likely more complex than those we have considered. Furthermore, as mentioned in our response to R1C2, this variation does not undermine the take home messages.

(RIC5) The x-axis scale chosen for these model-based scan figures makes it very difficult to identify the small amounts of scan time (<10 min.) used in many large consortia datasets (UK Biobank, ADNI). How then can researchers practically use the information from this paper to interpret past results? This seems like a missed opportunity to clarify the current literature. Ultimately, as I read through the manuscript, I asked myself: what is the goal of this manuscript? If the goal is to critique past consortia studies for not collecting sufficient per participant data, then this point is not made very clearly, and not shown in the figures. If it is to provide direction for future consortia studies, this guidance seems relatively weak, given the large error bars and lack of engagement with specific parameters likely to affect future BWAS. At best, it provides murky findings. At worst, I think it has the risk of promoting a take-away message that is not sufficiently nuanced (i.e., "participant number and data quantity trade off"), which can lead to the impression that these two quantities don't matter individually and have the same influence. But, as R2 pointed out (and is also embedded in the modeling framework), different mechanisms cause these parameters to influence BWAS, which will likely lead their outcome to differ based on the specific brain areas, resolution, and dataset characteristics of the project in question. While the model may be able to capture these differences, the authors do not do much to engage with them within the primary text or explain them to readers.

Once again, we thank you for your insightful comments, which were critical in driving us to derive a key set of take-home messages. We hope that our responses above have clarified practical guidance for the field. We also hope that the extensive analyses presented in our response to R1C2 provide a more in-depth assessment of the extent to which sample size and scan duration are interchangeable across different study design parameters and constraints. We believe that the extensive analyses, and our careful consideration of exceptions and edge cases

throughout the new revised manuscript, have provided a nuanced discussion of these factors for study design that avoid over-simplification. For example, how specific brain areas, resolutions and dataset characteristics influence optimal scan times in the context of scanning costs, will be useful for future (unknown) technological innovations that can improve scan characteristics (e.g., SNR) or phenotypic characterizations (e.g., more predictable or reliable phenotypes). At the same time, all of these analyses support the take-home message – extending scan time to 30 min can lead to significant budgetary savings for future BWAS studies.

Referee #2:

(R2C1) The revision has successfully addressed all my comments. I appreciate their thoughtful and thorough responses. In particular, I commend the authors for their efforts in developing a detailed theoretical interpretation of a potential mechanism in R2C1, along with the corresponding revisions in the Results section. Additionally, the inclusion of further tests using task-based data, clinical data, and two-session resting-state data significantly strengthens the study. Overall, I believe the authors have done an excellent job with these revisions. Congratulations!

We thank you for the positive comments. We also appreciate your suggestions, which really help to improve the study.

Referee #3:

(R3C1) In this revision, the authors have added a number of new analyses, including statistics and generalizability tests to task fMRI data, clinical data, different phenotypes, and different MR acquisition and processing protocols. The authors have provided substantial results, addressing the previous review comments from myself and other reviewers. This revision has greatly improved the manuscript, and I commend the authors for their responsiveness and extensive work.

We thank you for the positive comments. Your suggestions have been extremely useful to improving our study.

(R3C2) I have one specific follow-up question regarding the task fMRI analyses. The revision discusses "non-stationarity" in rest fMRI data. I wonder whether the differences between task and rest fMRI data (Fig. S26) can be at least partially explained by differences in stationarity between fMRI types (i.e. mental/cognitive states, randomness, etc.), and if so, to what extent. I guess that the experimental designs (block vs. event and their similarity across participants) may influence such an observation. It would be helpful to further discuss the experimental choices (sample size and scan length) with differences between fMRI data types.

You have raised a very interesting point! While the optimal total scan length for FC computed from resting state and task fMRI were both greater than 20 minutes, tasks seemed to require less scanning length than resting state scans. One explanation for this could be due to fMRI tasks being effective at synchronizing brain states across participants, thereby minimizing the presence of spurious non-stationary influences which may degrade modelling of inter-individual

differences. In addition, task-based fMRI may preferentially engage cognitive and physiological mechanisms related to the expression of various phenotypes (e.g. processing speed), thereby enhancing the specificity of FC estimates for phenotypic prediction compared with FC computed from resting state scans.

pg 16 (Results)

Previous studies have shown that task-FC leads to better prediction performance for cognitive measures (Greene et al., 2018; Chen et al., 2022). Here, we extend previous results, finding that the most cost-effective scan time was significantly shorter in ABCD task-fMRI than ABCD resting-fMRI (Fig. 7b). Among the three tasks, the most cost-effective scan time was the shortest for the n-back task (25 min). However, even for the n-back task, 30-min scans led to only 0.9% higher cost (relative to 25 min), compared with 16.1% higher cost for 10-min scans.

These task results suggest that the most cost-effective scan time is sensitive to brain state manipulation. Task-based fMRI may preferentially engage cognitive and physiological mechanisms that are closely tied to the expression of specific phenotypes (e.g., processing speed), thereby enhancing the specificity of functional connectivity estimates for phenotypic prediction. Tasks may also facilitate shorter, more efficient scan durations by aligning brain states across individuals in a controlled manner, thereby reducing spurious non-stationary influences that could otherwise obscure reliable modeling of inter-individual differences. This alignment might be better achieved in tasks that present stimuli and conditions with identical timing across participants – whether using event-related or block designs.

Non-stationarity may also be potentially increased by distributing resting state fMRI runs across multiple sessions. Since the HCP dataset was collected on two different days (sessions), we were also able to directly compare the effect of a two-session versus a one-session design, controlling for total scan time. The most cost-effective scan time for the two-session design was only slightly longer than the original HCP analysis: 40 min vs 30 min (Fig. 7c).

Overall, these results suggest that state manipulation can influence the most cost-effective scan time, and that a relatively large state manipulation (e.g., task) can significantly influence cost-effectiveness.

References

Chen, J., Tam, A., Kebets, V., Orban, C., Ooi, L. Q. R., Asplund, C. L., Marek, S., Dosenbach, N. U. F., Eickhoff, S. B., Bzdok, D., Holmes, A. J., & Yeo, B. T. T. (2022). Shared and unique brain network features predict cognitive, personality, and mental health scores in the ABCD study. *Nature Communications*, 13(1), 2217. <https://doi.org/10.1038/s41467-022-29766-8>

Gell, M., Eickhoff, S. B., Omidvarnia, A., Küppers, V., Patil, K. R., Satterthwaite, T. D., Müller, V. I., & Langner, R. (2023). The Burden of Reliability: How Measurement Noise Limits Brain-Behaviour Predictions. *bioRxiv*, 2023.2002.2009.527898. <https://doi.org/10.1101/2023.02.09.527898>

Greene, A. S., Gao, S., Scheinost, D., & Constable, R. T. (2018). Task-induced brain state manipulation improves prediction of individual traits. *Nature Communications*, 9(1). <https://doi.org/10.1038/s41467-018-04920-3>

Kang, K., Seidlitz, J., Bethlehem, R. A. I., Xiong, J., Jones, M. T., Mehta, K., Keller, A. S., Tao, R., Randolph, A., Larsen, B., Tervo-Clemmens, B., Feczko, E., Dominguez, O. M., Nelson, S. M., Alexander-Bloch, A. F., Fair, D. A., Schildcrout, J., Fair, D. A., Satterthwaite, T. D., Alexander-Bloch, A., Vandekar, S., & Lifespan Brain Chart, C. (2024). Study design features increase replicability in brain-wide association studies. *Nature*, 636(8043), 719-727. <https://doi.org/10.1038/s41586-024-08260-9>

Marek, S., Tervo-Clemmens, B., Calabro, F. J., Montez, D. F., Kay, B. P., Hatoum, A. S., Donohue, M. R., Foran, W., Miller, R. L., Hendrickson, T. J., Malone, S. M., Kandala, S., Feczko, E., Miranda-Dominguez, O., Graham, A. M., Earl, E. A., Perrone, A. J., Cordova, M., Doyle, O., Moore, L. A., Conan, G. M., Uriarte, J., Snider, K., Lynch, B. J., Wilgenbusch, J. C., Pengo, T., Tam, A., Chen, J., Newbold, D. J., Zheng, A., Seider, N. A., Van, A. N., Metoki, A., Chauvin, R. J., Laumann, T. O., Greene, D. J., Petersen, S. E., Garavan, H., Thompson, W. K., Nichols, T. E., Yeo, B. T. T., Barch, D. M., Luna, B., Fair, D. A., & Dosenbach, N. U. F. (2022). Reproducible brain-wide association studies require thousands of individuals. *Nature*, 603(7902), 654-660. <https://doi.org/10.1038/s41586-022-04492-9>

Nikolaidis, A., Chen, A. A., He, X., Shinohara, R., Vogelstein, J., Milham, M., & Shou, H. (2022). Suboptimal phenotypic reliability impedes reproducible human neuroscience. *bioRxiv*, 2022.2007.2022.501193. <https://doi.org/10.1101/2022.07.22.501193>

Varoquaux, G. (2018). Cross-validation failure: Small sample sizes lead to large error bars. *Neuroimage*, 180(Pt A), 68-77. <https://doi.org/10.1016/j.neuroimage.2017.06.061>

Editorial Requests

(ER1) Please add references to the abstract and reduce it to 230 words or less (currently there are 255 words). Please also be sure to put the link to the working web application in the abstract.

Yes, we have done so.

(ER2) Please add the full statistical details that any of your claims rest on to the results (or add a reference to where these can be found, e.g. a table in the SI). However, it is important that every statement based on statistical analyses either includes these or references them.

We have added Supplementary Table 1, which summarizes all descriptive statistics and statistical tests supporting claims in this study. In cases where a single statistic could be reported, we have done so in the text. For more complex statistical arguments, we referenced Supplementary Table 1 in the text.

(ER3) Is it possible to do a statistical test to show that scan time per participant and sample size are interchangeable, ie something that shows that the two are equivalent? If so, please add such a test (and if not, just make sure that you reference the data that allow you to make this claim).

Thank you for highlighting this point. We have discussed with Thomas Nichols and we do not think that it is possible to do an explicit statistical test on scan time and sample size being interchangeable. However, we have performed a statistical test showing that prediction accuracy has a strong linear relationship with logarithm of total scan duration ($p = 0.001$). Since total scan duration = sample size \times scan time per participant, we can interpret this statistical test as providing evidence that sample size and scan time are broadly interchangeable (when scan time is ≤ 20 mins).

The manuscript has been updated with the following sentence “The logarithm of total scan duration explained prediction accuracy very well ($r = 0.95$; $p = 0.001$). This suggests that sample size and scan time are broadly interchangeable, in the sense that larger sample size can compensate for smaller scan time and vice versa. The exact degree of interchangeability will be characterized in the next section.”

Details of this statistical test can be found in the new Supplementary Table 1.

(ER4) On a related note, please always quantify rather than qualify and add statistics (e.g. when you say “diminishing returns weren’t observed” (line 189), please add the information about the non-significant statistics (or reference the table that shows these). Please also be sure to always add null statistics (e.g. equivalence tests or Bayes factors) if you want to interpret any null results / any null results are important to understand the implications of the paper.

We have carefully gone through the study, and when possible, we have added statistical tests and/or confidence intervals. When not possible, we have added quantification and explanation of why the quantification support our interpretations. All details can be found in the new Supplementary Table 1.

With regards to “diminishing returns were not observed”, we have again consulted Thomas Nichols. We do not think it is possible to have a statistical test showing that “diminishing returns weren’t observed.” Furthermore, we now think that the statement might also confuse the reader. The original statement was based on visual inspection of Fig. 2a, but the reader might interpret the statement as contradicting the next section where we do show a diminishing returns of scan time relative to sample size, even in ABCD (Fig. 3a, Extended Data Fig. 3 and also in the theoretical model in Fig. 3b).

As such, we have rephrased various statements in this section, where we provided an example of diminishing returns of scan time (relative to sample size), and we weakened the statement where we discussed the same effect in ABCD. More detailed examples are provided in the new Supplementary Table 1.

pg 5 (Sample size & scan time interchangeability)

In the HCP dataset, we also observed diminishing returns of scan time relative to sample size, especially beyond 30 minutes (Fig. 2a; Supplementary Table 1). For example, starting from an accuracy of 0.33 with 200 participants \times 14 min scans, a 3.5 \times larger sample ($N = 700$) lifted accuracy to 0.45, whereas a 4.1 \times longer scan ($T = 58$ min) raised it only to 0.40...

... Diminishing returns of scan time (relative to sample size) was observed for many HCP phenotypes, especially beyond 20 minutes (Supplementary Table 1). This phenomenon was less pronounced for the ABCD phenotypes, potentially because maximum scan time was only 20 minutes (Supplementary Table 1).

(ER5) Please ensure that the references (1) numbered (rather than written out in the text and (2) that this numbering is continuous, meaning that you start with reference 1 in the abstract and continue counting up through the main text; if you end with 60, the first methods reference should be 61 (and proceed continuously from there).

Thanks. We have done so.

(ER6) Please reference relevant subsections of the methods from the results whenever necessary.

Thanks, we have done so.

(ER7) Please remove the main figures from the article file and re-supply them individually in an acceptable format such as EPS, AI, PS, PDF, PPT, PSD or XLS (for graphs) with editable vector files.

Thanks. We have done so.

(ER8) Please ensure that the text size in all figures is at least 5 pt Arial.

Yes, we confirm.

(ER9) Please indicate discontinuities with double slashes on any relevant graphs (e.g. Fig 1b).

Thank you for the clarification in your email. We have now updated Fig 1b and Extended Data Fig. 5c to start from zero. Discontinuities are shown with double slashes.

(ER10) Please reduce subheadings to 40 characters (with spaces) or less.

We have managed to reduce all results and methods subheadings to 40 characters or less, except subheading “Sample size and scan time interchangeability” (44 characters) and “Variation across phenotypes and scan parameters” (47 characters). Hope that is ok, though we are open to suggestions.

(ER11) Please consider moving some (maybe 2-3) of the main text figures/tables to Extended Data to reduce the length of the paper.

Thanks, we have done so.

(ER12) Please move some of the display items from the Supplement (SI) to the Extended Data (ED). You can have 10 ED items, and we think it would be good if you could use them all.

Thanks. We now have 10 extended data figures/tables.

(ER13) Please remove the Extended data figures from the article file and re-supply them individually in EPS, JPEG or TIF format.

Done!

(ER14) Please provide a supplementary information guide.

Thank you. We have added a table of contents for the supplementary material.

(ER15) For any Supplementary Figures, please check and confirm that:

* If data is presented as bar charts, individual data points are shown using overlaid dot plots.

There is no bar chart in the study.

* The n number (i.e. the sample size used to derive statistics) is provided and defined as a precise value (not a range).

Thanks! We have now added this information in the figure captions and in the new Supplementary Table 1.

* Any chart axis, error bars, scale bars, symbols and colour scales are defined.

Thanks. We confirm this has been done.

* Any statistical tests used for data analysis are specified and exact p-values are provided either on the figures themselves, in the legend or in the Source Data file.

Thanks. We have done so.

(ER16) Please provide separate 'Data availability' and 'Code availability' statements in the manuscript.

Thanks. We have done so.

(ER17) Please update the link(s) in these statements and ensure they are in working order.

The links have been updated.

(ER18) Please include information on both the ethical approval for the datasets you use and whether or not you needed ethical approval (or had a waiver) for the current study in the methods section of the main manuscript.

We have added more information about ethics approval.

Reviewer 3 comments

(R3C1) It seems to me that the authors have adequately addressed my last comment and R1's concerns in this re-revised submission.

We thank the reviewer for reviewing our paper again. We appreciate advice from all the reviewers, which have greatly improved the paper.

(R3C2) However, I'd like to raise an additional concern regarding the authors' third key message. They suggest that 30-minute scans are cost-effective even for task-related data, as demonstrated by their analysis. My concern is that this conclusion about cost-effectiveness for task fMRI is not directly supported by the empirical data, since all task fMRI datasets used in this study are shorter than 15 minutes (Table 1).

This limitation should be clearly discussed in the manuscript. I am sorry if the authors have already clarified this somewhere in the manuscript that I may have missed.

Thank you for the suggestion. We have updated the manuscript to include this limitation.

pg 13 (Longer scans are more cost-effective in the Results section)

Overall, 10-min scans are rarely cost-effective, and optimum scan time is ≥ 20 min in most BWAS (Fig. 4c). Among the datasets we analyzed, four included scans ≥ 20 min, providing robust evidence to support this conclusion across multiple datasets. In contrast, we could only identify one dataset (HCP) with scans exceeding 30 minutes and a sufficiently large sample size for inclusion in our study. Similarly, although the ABCD task-fMRI scans are among the longest in existing large-scale datasets, the longest scan

duration is less than 13 min. This limitation underscores the importance of our findings, emphasizing the need for BWAS to prioritize longer scans.